# Astrocyte–neuron subproteomes and obsessive–compulsive disorder mechanisms

Joselyn S. Soto[1], Yasaman Jami-Alahmadi[2], Jakelyn Chacon[1], Stefanie L. Moye[1], Blanca Diaz-Castro[1,4], James A. Wohlschlegel[2] & Baljit S. Khakh[1,3 ✉]

Astrocytes and neurons extensively interact in the brain. Identifying astrocyte and neuron proteomes is essential for elucidating the protein networks that dictate their respective contributions to physiology and disease. Here we used cell-specific and subcompartment-specific proximity-dependent biotinylation[1] to study the proteomes of striatal astrocytes and neurons in vivo. We evaluated cytosolic and plasma membrane compartments for astrocytes and neurons to discover how these cells differ at the protein level in their signalling machinery. We also assessed subcellular compartments of astrocytes, including end feet and fine processes, to reveal their subproteomes and the molecular basis of essential astrocyte signalling and homeostatic functions. Notably, SAPAP3 (encoded by *Dlgap3*), which is associated with obsessive–compulsive disorder (OCD) and repetitive behaviours[2–8], was detected at high levels in striatal astrocytes and was enriched within specific astrocyte subcompartments where it regulated actin cytoskeleton organization. Furthermore, genetic rescue experiments combined with behavioural analyses and molecular assessments in a mouse model of OCD[4] lacking SAPAP3 revealed distinct contributions of astrocytic and neuronal SAPAP3 to repetitive and anxiety-related OCD-like phenotypes. Our data define how astrocytes and neurons differ at the protein level and in their major signalling pathways. Moreover, they reveal how astrocyte subproteomes vary between physiological subcompartments and how both astrocyte and neuronal SAPAP3 mechanisms contribute to OCD phenotypes in mice. Our data indicate that therapeutic strategies that target both astrocytes and neurons may be useful to explore in OCD and potentially other brain disorders.

Astrocytes are the predominant type of glia in the central nervous system and have coevolved with neurons[9]. Astrocytes are vital components of the brain[10] and like neurons, they display morphologies and properties that differ among brain regions[11–14]. Both astrocytes and neurons are extensively implicated in brain diseases[15], including psychiatric disorders. However, little is known about shared or separate astrocytic and neuronal molecular mechanisms and their respective contributions within brain regions relevant to defined psychiatric diseases or phenotypes in mice.

Neurons and astrocytes interact anatomically and physiologically, including within the striatum[16,17]. In the settings of physiology and disease, most studies have compared astrocytes and neurons using neuropathological methods, physiology, cellular markers or RNA expression analyses. Regarding RNA, although invaluable, the relationship between RNA expression levels and protein levels[18] is highly complex; therefore, it is crucial to identify specific protein-based mechanisms for neurons and astrocytes[19]. Furthermore, to understand the basic biology of astrocytes and neurons, it is necessary to capture protein identities and their differences within morphologically intact cells. Cell dissociation and fluorescence activated cell sorting

(FACS) procedures shear most astrocyte and neuronal processes and are particularly damaging to astrocytes that normally respond to tissue stress[20–22], vitiating the use of these methods for proteomics. As a result, the proteomes of astrocytes and neurons have not been directly measured, compared or utilized to understand their contributions to relevant phenotypes in physiology or psychiatric disease in any species.

## Approach

The striatum is the largest nucleus of the basal ganglia, a group of subcortical nuclei involved in movement, actions and diverse neuropsychiatric conditions[23–25]. The striatum contains extensive contacts between astrocytes and neurons, 95% of which are DARPP32-positive medium spiny neurons (MSNs)[16]. As astrocytes lose their complex morphology following dissociation (Extended Data Fig. 1a–d), we characterized the composition of cell-type-specific proteomes (astrocytes and neurons) and compartments (cytosolic and plasma membrane (PM)) using genetically targeted biotin ligase (BioID2; Extended Data Fig. 2a,b) delivered in vivo within the striatum using adeno-associated viruses

[1]Department of Physiology, University of California, Los Angeles, CA, USA. [2]Department of Biological Chemistry, David Geffen School of Medicine, University of California, Los Angeles, CA, USA. [3]Department of Neurobiology, University of California, Los Angeles, CA, USA. [4]Present address: UK Dementia Research Institute and Centre for Discovery Brain Sciences, University of Edinburgh, Edinburgh, UK. ✉e-mail: bkhakh@mednet.ucla.edu

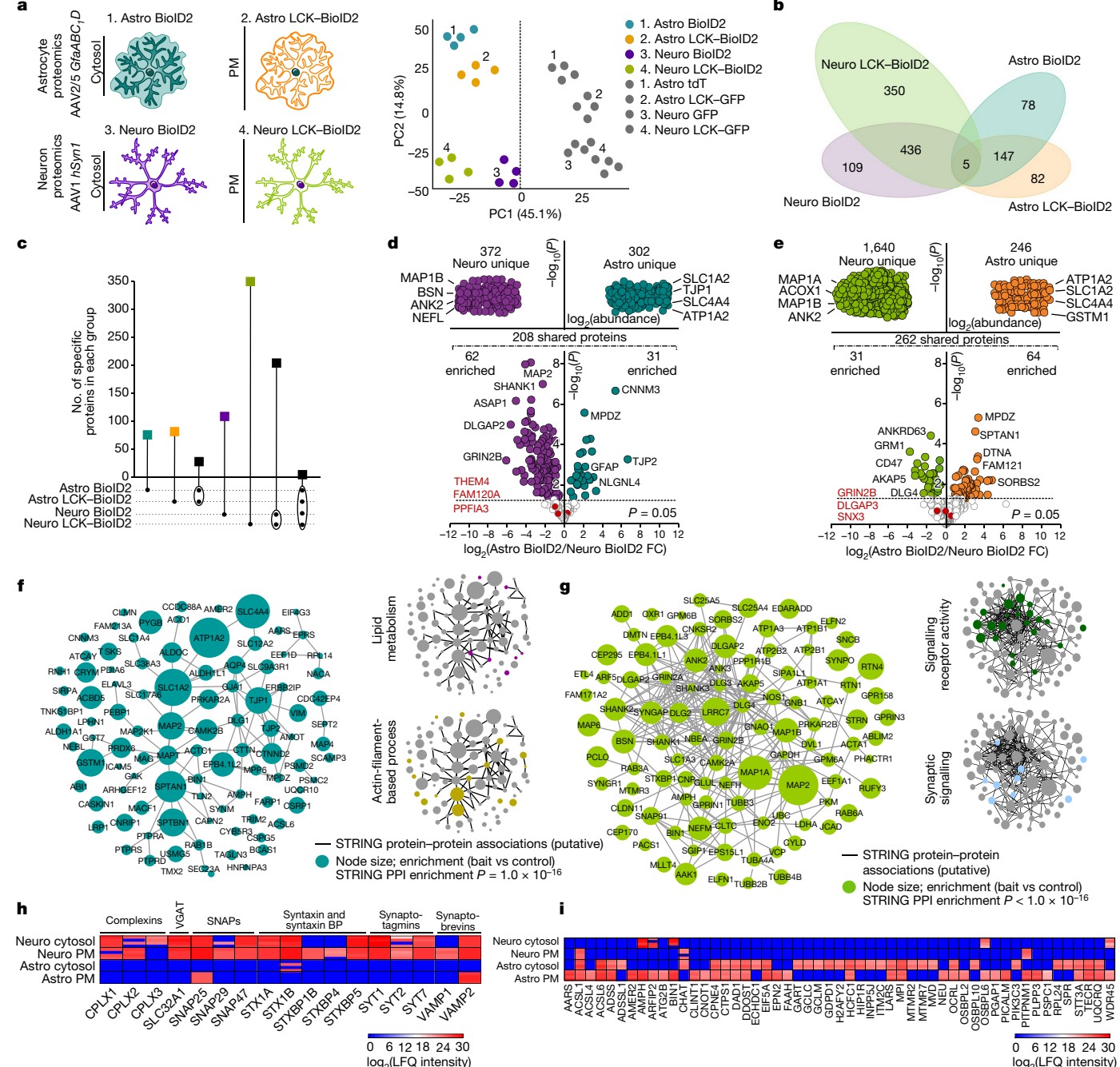

**Fig. 1 | Striatal astrocyte and neuron proteomes. a**, Left, cartoons representing genetically targeted BioID2 in the cytosol and PM of astrocytes and neurons. Right, PCA plot of all proteins detected by mass spectrometry (*n* = 4 technical runs per construct with 8 mice each). **b**, Clustergram depicting the specific number of proteins detected in each BioID2 construct experiment of astrocytes and neurons. All proteins hereafter represent those that were significant (log$_2$(fold change (FC)) > 1 and FDR < 0.05 versus GFP controls). **c**, UpSet plot of BioID2-identified proteins. **d**, LFQ comparison of proteins detected by cytosolic Astro BioID2 and Neuro BioID2. Top, proteins specific to Neuro BioID2 or Astro BioID2 when compared with each other. The four most abundant proteins are named. Bottom, comparison of proteins that were shared in both cytosolic Astro BioID2 and Neuro BioID2. The five highest

enriched proteins (log$_2$(FC) > 2) are indicated. The top three proteins that showed no enrichment in either cell are depicted in red. **e**, As in **d** but for PM Astro BioID2 and Neuro BioID2. **f**, Left, STRING analysis map of the top 100 proteins identified with Astro BioID2 and Astro LCK–BioID2. Node size represents the enrichment of each protein versus the GFP control. Edges represent putative interactions from STRING. Right, small clustergrams show categories for biological process. PPI, protein–protein interaction. **g**. As in **f**, but for Neuro BioID2 and Neuro LCK–BioID2. **h**, Expression levels (LFQ intensity) of Ca$^{2+}$-dependent vesicle release proteins identified by each BioID2 construct. BPs, binding proteins. **i**, Expression levels of proteins related to lipid metabolism identified in each BioID2 construct.

(AAVs; Extended Data Fig. 2b,c). This method does not use cell dissociation or FACS. BioID2 biotinylates proteins at free lysine residues in the presence of biotin[1,26]. After characterizing cytosolic BioID2 and PM-targeted LCK–BioID2 constructs in HEK-293 cells (Extended Data

Fig. 3), we selectively delivered BioID2 or LCK–BioID2 to astrocytes or neurons using a truncated *GFAP* promoter (*GfaABC₁D*) or human *SYN1* promoter[27] and AAVs with preferred astrocyte (Astro) or neuron (Neuro) tropism, respectively (Fig. 1a and Extended Data Fig. 2c–h).

Astro BioID2, Astro LCK–BioID2 and the proteins they biotinylated were detected only in S100β-positive bushy astrocytes, including within end feet (Extended Data Fig. 4a–d). Conversely, Neuro BioID2, Neuro LCK–BioID2 and their biotinylated proteins were detected within DARPP32-positive neuronal somata and the neuropil, which reflected their axonal and dendritic expression, respectively (Extended Data Fig. 4e–h). Western blot analyses confirmed biotinylation (Extended Data Figs. 2e–h and 4; $P < 0.01$ in each case), which enabled protein identification by liquid chromatography–tandem mass spectrometry (LC–MS/MS).

## Cytosolic and PM proteomes

When we compared the number of proteins detected in each cell-specific and compartment-specific BioID2 experiment to their respective AAV fluorescent protein controls, we found around 500–1,800 proteins within Astro BioID2, Astro LCK–BioID2, Neuro BioID2 and Neuro LCK–BioID2 experiments (Supplementary Table 4). Principal component analysis (PCA) separated controls from BioID2 groups, astrocytes from neurons and cytosol and PM for both astrocytes and neurons (Fig. 1a). Clustergram and UpSet analyses identified distinct proteins in Astro BioID2, Astro LCK–BioID2, Neuro BioID2 and Neuro LCK–BioID2 groups, proteins shared between the cytosol and PM of astrocytes and neurons, and proteins common to all four groups (Fig. 1b,c). For example, 82 unique proteins were identified in the astrocyte PM group (LCK–BioID2; Fig. 1b,c). Astrocyte and neuron proteomes contained cell-enriched markers[28], but not those for other cells (Extended Data Fig. 5a). Cell-specific proteomes demonstrated differences between cytosolic and PM compartments of astrocytes and neurons (Fig. 1a–c).

Using label-free quantification (LFQ) intensity and a false discovery rate (FDR) of <0.05 for cytosolic BioID2, we identified 208 proteins that were shared between astrocytes and neurons, and 302 and 372 that were specific to astrocytes and neurons, respectively (Fig. 1d). Similarly, we identified 262 proteins that were shared between PM compartments of astrocytes and neurons, and 246 and 1,640 that were specific to the PM of astrocytes and neurons, respectively (Fig. 1e). The examples in Fig. 1d,e include known and new proteins; the full list in Supplementary Table 1 includes genes that encode synaptic proteins for neurons (for example, *GRIA4*, *HOMER*, *Dlg4*, *Shank1* and *Ank1*) and genes that encode membrane and cytoskeletal proteins in astrocytes (for example, *Ezr*, *Slc1a2*, *Atp1a2*, *Kcnj10* and *Rdx*). The major signalling pathways identified were lipid metabolism, cell–cell signalling and actin-filament-based processes for astrocytes. By contrast, ion binding, receptor and synaptic signalling were the main signalling pathways identified for neurons (Extended Data Fig. 5). Core biosynthetic pathways were shared between astrocytes and neurons (Extended Data Fig. 5). We determined the putative association network and node size for the 100 most abundant proteins detected in astrocytes and the network population by proteins related to lipid metabolism and to actin-filament-based processes (Fig. 1f). Similarly, Fig. 1g plots the putative association network and node size for the 100 most abundant proteins detected in neurons and networks related to signalling receptor activity and synaptic signalling. We also evaluated the putative association network for all astrocyte and neuron proteins (Extended Data Fig. 6). Fundamental differences between astrocytes and neurons were identified; for example, proteins related to $Ca^{2+}$-dependent vesicular γ-aminobutyric acid release were abundant in neurons but largely absent in astrocytes[28] (Fig. 1h and Extended Data Fig. 5f). Conversely, proteins related to lipid metabolism were abundant in astrocytes but were fewer in neurons (Fig. 1i and Extended Data Fig. 5g). Although many of the top genes enriched in astrocytes and neurons identified by RNA sequencing (RNA-seq)[27,28] were detected in the proteomes (Extended Data Fig. 7a,b), the relationship between protein abundance and RNA expression was weak (Extended Data Fig. 7c–j). This result indicates that RNA levels do not accurately reflect protein abundance[18]. This is

not a critique of RNA-seq, but reflects meaningful biology related to differences in transcript and protein turnover, as known for other cells[18].

We validated the expression of *Crym* (which encodes μ-crystallin), *Mapt* (which encodes microtubule-associated protein tau) and *Tjp1* (which encodes the tight junction protein ZO-1) by RNAscope in astrocytes positive for both S100β and *Aldh1l1* tdTomato alongside assessment of their expression in different cell types using single cell RNA-seq (scRNA-seq) data[29] (Extended Data Fig. 8). These analyses confirmed that some of the identified proteins were enriched in astrocytes (μ-crystallin) and others were concomitantly expressed in other cell types (*Mapt* and *Tjp1*). Similar evaluations will be necessary on a case-by-case basis for other proteins (Fig. 1).

## Astrocyte subproteomes

Astrocytes comprise a cell body and subcompartments such as the PM, branches, blood-vessel-associated end feet and finer processes[16] (Extended Data Fig. 1). It is widely held that important physiology occurs in these specialized structures, making it important to understand the proteins in these spaces. Extending work with HEK-293 cells[1], we explored the subproteomes of five astrocyte subcompartments in vivo defined by the presence of known molecules (Fig. 2a). We therefore generated the following AAVs: (1) AQP4–BioID2 to assess astrocyte end feet (AQP4 is the water channel enriched in end feet[30]); (2) EZR–BioID2 to evaluate astrocyte processes (EZR is a structural protein within fine processes[31]); (3) GLT1–BioID2 to evaluate sites of extracellular glutamate uptake (also known as SLC1A2, GLT1 is the major astrocyte glutamate transporter[32]); (4) KIR4.1–BioID2 to assess sites of extracellular $K^+$ homeostasis (KIR4.1 is a main astrocyte $K^+$ channel[33]); and (5) CX43–BioID2 to assess astrocyte–astrocyte contacts (CX43 is the main connexin underlying astrocyte coupling[34]). The control for each was the identical targeting molecule but with green fluorescent protein (GFP) replacing BioID2. Each AAV resulted in BioID2-HA expression levels similar to the endogenous target protein (Extended Data Fig. 9). Furthermore, the distribution patterns of the biotinylated proteins, as assessed by immunohistochemistry (IHC), depended on the construct (Extended Data Fig. 9), which indicated that biotinylated proteins were proximal to the cognate BioID2 construct. This was a desired and anticipated feature[1] because biotinylation displays proximity dependence over tens of nanometres. Western blot analyses for all target BioID2 groups showed biotinylation (Extended Data Fig. 10; $P < 0.05$ in each case). PCA of the proteomics data separated controls from the target BioID2 groups, several from each other (Fig. 2b), and clustergram analyses identified specific proteins in each subcompartment, ranging from 51 in the CX43–BioID2 compartment to 247 in AQP4–BioID2. There were 26 proteins shared across all astrocyte subcompartments (Fig. 2c). We detected astrocyte markers[28] in the proteomics data, but not those for other major cell types (Extended Data Fig. 10f). The shared proteins and subproteomes are provided in Supplementary Table 2. Using our in vivo methods, 3,274 astrocyte subcompartment proteins were identified, whereas for astrocytes isolated by FACS[28], only 1,378 were detected. This result underscores the fact that FACS-isolated cells lose their bushy processes and associated proteomes (Extended Data Fig. 1).

## Astrocyte subproteome cards

We compared proteins shared between any single BioID2-targeted subcompartment and astrocyte cytosolic BioID2 (Fig. 2) to find subcompartment-enriched proteins. Volcano plots were generated to compare Astro BioID2 with LCK–BioID2 (Fig. 2d), Astro BioID2 with EZR–BioID2 (Fig. 2e) and Astro BioID2 with AQP4–BioID2 (Fig. 2f). Interaction maps were made for the top 50 proteins identified with EZR–BioID2 and AQP4–BioID2 (Fig. 2g,h). Astrocyte subcompartments differed in their proteins and their predicted biological

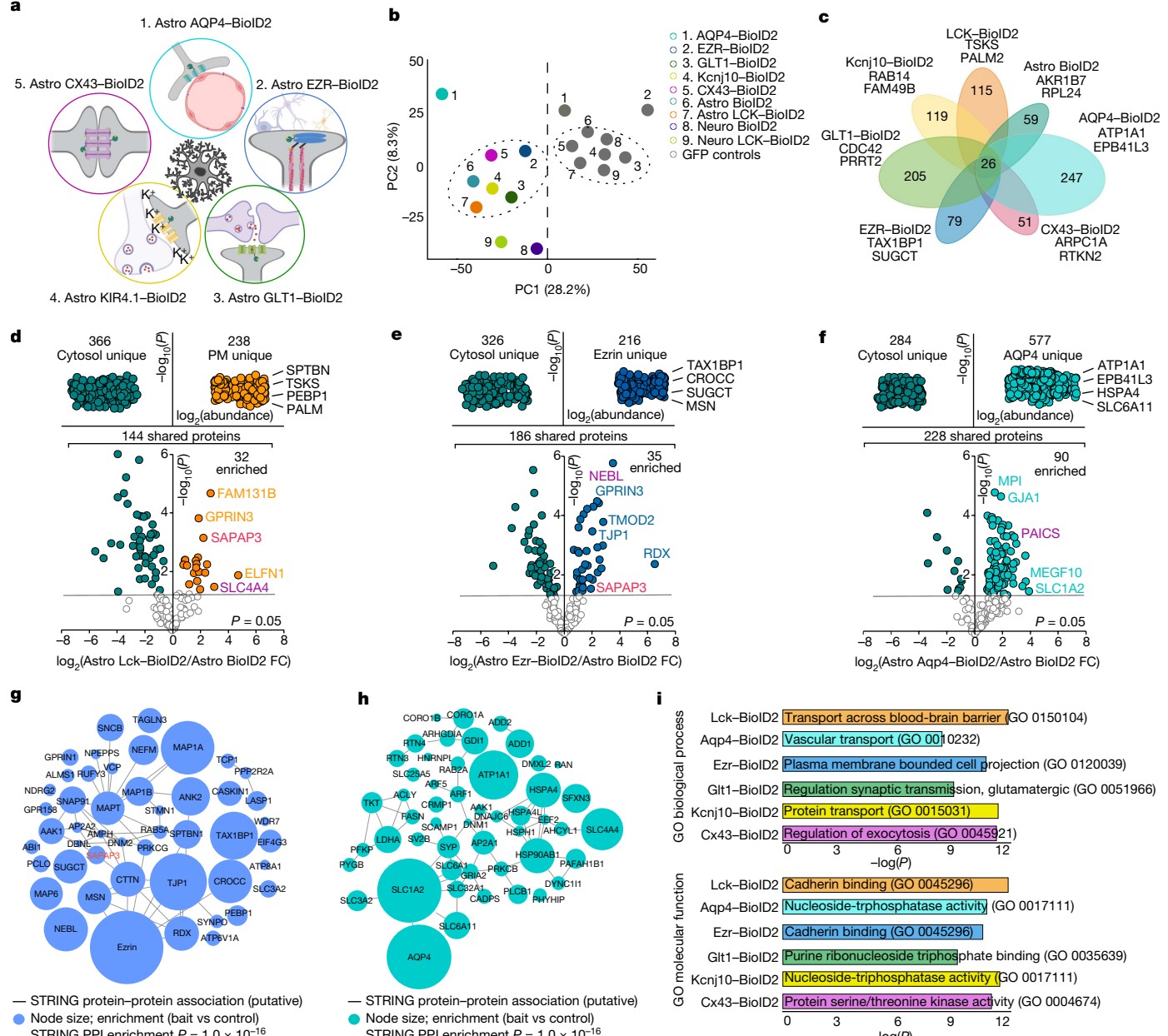

**Fig. 2 | Striatal astrocyte subcompartment proteomes. a**, Cartoon illustrating the five astrocyte subcompartments targeted genetically with BioID2. **b**, PCA plot of the average of all proteins identified in each BioID2 construct. Points represent the mean for each construct ($n$ = 4 technical runs consisting of 8 mice each per construct). **c**, Clustergram depicting the unique number of proteins detected in each subcompartment-specific BioID2 experiment. All proteins hereafter represent those that were significant ($\log_2$FC > 1 and FDR < 0.05 *versus* GFP controls). The top two most abundant proteins for each subcompartment are named. **d**, Label-free based quantification comparison of proteins detected in the cytosolic Astro BioID2 and PM Astro LCK–BioID2. Top, specific LCK–BioID2 proteins compared to cytosol. The top four most abundant proteins for LCK–BioID2 are indicated. Bottom, volcano plot comparing proteins that were shared in both cytosolic BioID2 and LCK–

BioID2. The five highest enriched proteins for LCK–BioID2 ($\log_2$(FC) > 2) are indicated. Magenta label shows protein that was validated by IHC in Extended Data Fig. 11. Red label shows that SAPAP3 is enriched in the astrocyte PM. **e,f**, As in **d** but for cytosolic Astro BioID2 and Astro EZR–BioID2 (**e**) and cytosolic Astro BioID2 and Astro AQP4–BioID2 (**f**). **g**, STRING analysis map of the top 50 (by LFQ abundance) biotinylated proteins identified in astrocyte fine processes with Astro EZR–BioID2. Node size represents the enrichment of each protein versus the GFP control. Edges represent putative interactions from the STRING database. **h**, As in **g** but for proteins identified in the astrocyte end foot with Astro AQP4–BioID2. **i**, Bars show the most significant Enrichr gene ontology (GO) term for the unique and enriched proteins found in each astrocyte subcompartment. Top, the GO term for biological process. Bottom, the GO term for molecular function.

functions (Fig. 2i). For each subcompartment, we provide astrocyte subproteome cards reporting the following information: (1) unique and enriched proteins; (2) relationships between protein abundance and RNA expression; (3) validation of candidate proteins within the targeted subcompartment; and (4) protein–protein-association maps for the unique and enriched proteins, along with major signalling pathways

(Supplementary Video 1 and Extended Data Figs. 11–16). Although known interactions, such as between Kir4.1 and AQP4, EZR and radixin, AQP4 and GLT1, CX43 and TJP1, and GLT1 and hepatic and glial cell adhesion molecule (HepaCAM), were confirmed, many hundreds of new putative interactions were discovered across subcompartments (Extended Data Figs. 11–16 and Supplementary Table 2). The astrocyte

subproteomes permit new types of experiments to explore astrocytic contributions to brain function. The data revealed proteins that were previously unexplored in astrocytes. One of these proteins was enriched in the striatum[35]: SAPAP3 (encoded by *Dlgap3*; Figs. 1e and 2d,e and Extended Data Figs. 11 and 13).

## Neuron and astrocyte SAPAP3 expression

SAPAP3, which is expressed in MSNs and is associated with OCD in humans and with repetitive behaviours[2–8,35], was detected at similarly high levels in striatal astrocyte and neuron PM compartments (Fig. 3a). SAPAP3 was also found in astrocyte subcompartments assessed using EZR–BioID2, but not those assessed using AQP4–BioID2, GLT1–BioID2, KIR4.1-BioID2 or CX43–BioID2 (Fig. 3a). This result implied that astrocytic SAPAP3 mostly exists in the cytosol and near the PM of the fine processes of astrocytes. The proteomics findings were supported by neuron-specific and astrocyte-specific RNA-seq data, which showed similar *Dlgap3* expression levels (Fig. 3b; FDR < 0.05). Accordingly, scRNA-seq[36] analyses of *Dlgap3* showed similar expression in neurons and astrocytes (Fig. 3c). To further validate the data from astrocytes, we performed RNAscope fluorescence in situ hybridization. Abundant *Dlgap3* mRNA in genetically labelled tdTomato astrocytes was detected, which was not observed in *Dlgap3* knockout (SAPAP3 KO) mice (Fig. 3d,f; $P < 0.01$). We also detected abundant SAPAP3 protein in genetically labelled tdTomato astrocytes, whereas immunostaining was significantly reduced in SAPAP3 KO mice (Fig. 3e,g; $P < 0.01$). Together, the data from proteomics, cell-specific RNA-seq, scRNA-seq, RNAscope and IHC in wild-type (WT) and SAPAP3 KO mice provide strong evidence that astrocytes express SAPAP3 (Figs. 1e, 2d,e and 3a–e).

## Neuron and astrocyte SAPAP3 mechanisms

Although SAPAP3 can interact with the postsynaptic density at glutamatergic synapses onto MSNs[4], there are no SAPAP3 interactome data in either astrocytes or neurons. To shed light on the mechanisms engaged by SAPAP3 (Fig. 4), we performed proteomics using SAPAP3–BioID2 constructs for astrocytes and neurons (Fig. 4a,b and Extended Data Figs. 17 and 18). We identified 49 SAPAP3 interactors in astrocytes, 306 in neurons and 109 shared ones (Extended Data Fig. 18). The top astrocyte SAPAP3 interactors were *Slc1a3*, *Slc1a2*, *Slc4a4*, *Dstn* and *Arpc2*, which reflect functions related to synaptic glutamate uptake and homeostasis and the actin cytoskeleton, which recalls the finding that SAPAP3 was identified as an EZR and PM interactor (Fig. 2d,e). The top neuron SAPAP3 interactors were *Grin2b*, *Shank3*, *Dlg3*, *Cnp* and *Syngap1*, which represent proteins in the postsynaptic density of glutamatergic synapses. Differential expression analysis showed that SAPAP3 fell on the *y* axis of the volcano plot. This result indicated that SAPAP3 exhibits similar abundance in astrocytes and neurons (Extended Data Fig. 18). SAPAP3-interaction maps for astrocytes and neurons (Fig. 4a,b) highlighted molecular pathways within astrocytes related to glutamate regulation through transporters, G protein signalling, protein localization and the actin cytoskeleton (Fig. 4a). Proteomics data defined putative SAPAP3 cell-specific interactions and molecular mechanisms in astrocytes and neurons that are shared (for example, related to glutamatergic signalling) and distinct (for example, actin cytoskeleton; Fig. 4a,b).

## Astrocytic SAPAP3 molecular mechanisms

The major interactions of SAPAP3 within astrocytes related to glutamate uptake and the actin cytoskeleton (Fig. 4a). We therefore sought to validate key protein–protein interactions between SAPAP3 and GLT1 and of SAPAP3 with EZR. As SAPAP3, EZR and GLT1 are expressed in other cells as well as in astrocytes[29], co-immunoprecipitation (co-IP) of endogenous proteins would not inform whether they associate in astrocytes.

Thus, we first used recombinant proteins expressed in striatal astrocytes in vivo for co-IP. HA-tagged SAPAP3 co-immunoprecipitated with EZR–GFP and with GLT1–GFP, and conversely, EZR–GFP and GLT1–GFP co-immunoprecipitated with HA–SAPAP3 (Extended Data Fig. 18d). Second, we used proximity ligation assays (PLAs) to explore associations between endogenous proteins (Fig. 4c). The results showed clear associations between SAPAP3 and GLT1 and between SAPAP3 and EZR in tdTomato-positive astrocytes (Fig. 4d,e). The PLA signals were absent in SAPAP3 KO mice (Fig. 4c,d).

As SAPAP3 interacted with several astrocytic membrane proteins (Fig. 4a), we determined whether the astrocyte PM proteome was altered in SAPAP3 KO mice relative to WT controls. Overall, 182 proteins were downregulated and 275 proteins were upregulated in SAPAP3 KO mice, including EZR and GLT1 (Fig. 4f.i). An analysis of the altered proteins in SAPAP3 KO mice identified 'actin cytoskeleton organization' as the major dysregulated pathway (Fig 4f, top), which complements the SAPAP3 interactor results (pink nodes in Fig. 4a) and the SAPAP3–EZR interactions (Figs. 2e and 4d and Extended Data Fig. 18d). To explore this finding, we used LifeAct GFP as an actin cytoskeleton reporter. A strong reduction in intensity of labelling of the actin cytoskeleton within astrocytes from SAPAP3 KO mice (Fig. 4g) was observed, and this was greatest at the edges of astrocyte territories where fine processes abut synapses[28] (Fig. 4i). Furthermore, on the basis of the LifeAct GFP images, astrocyte territories were reduced in area (Fig. 4i), which was independently confirmed using tdTomato (Fig. 4h,j). Taken together, these data provide strong evidence for the presence of SAPAP3 within astrocytes (Figs. 1–3) and for its molecular interactors and pathways that include the actin cytoskeleton (Fig. 4a,c,d). Moreover, the results also demonstrate the astrocytic molecular (Fig. 4f) and cellular (Fig. 4g–j) consequences of SAPAP3 deletion. We next explored the relevance of astrocytic SAPAP3 in relation to OCD phenotypes.

## Rescue of OCD-like phenotypes in mice

SAPAP3, a cytosolic scaffold protein involved in human OCD[3,5–7], is expressed in neurons and within select subcompartments of astrocytes (Figs. 1–4). SAPAP3 KO mice display OCD-like phenotypes of anxiety and repetitive self-grooming that results in facial lesions[4]. SAPAP3 KO mice are relevant models to use because SAPAP3 genetic variations are associated with some forms of human OCD[3,5–7] and SAPAP3 is highly expressed in the striatum of humans and mice[4,35]. In light of the proteomics data showing similar SAPAP3 abundance in astrocytes and neurons, we developed AAVs to deliver SAPAP3 specifically to astrocytes or neurons in SAPAP3 KO mice to determine whether expression within either cell type at postnatal day 28 (P28) could rescue OCD-like phenotypes at P180 (ref. 4) (Fig. 5a). AAVs were delivered bilaterally and broadly[4] within the striatum, which resulted in SAPAP3 expression selectively within astrocytes or neurons for Astro GFP–SAPAP3 and Neuro GFP–SAPAP3, respectively (Fig. 5b,c and Extended Data Fig. 18e–g).

We reproduced OCD-like behaviours[4] in WT and SAPAP3 KO mice (Extended Data Fig. 19). We then assessed the same behaviours to determine whether SAPAP3 rescue in astrocytes or neurons could produce ameliorative effects. Repetitive self-grooming in SAPAP3 KO mice and other OCD models results in facial lesions[4]. We measured the area of facial lesions, the number of lesions, the number of self-grooming bouts and the total time spent self-grooming as measures of repetitive behaviour. Astrocytic expression of GFP–SAPAP3 significantly ameliorated all of these parameters in a manner similar to neuronal expression of GFP–SAPAP3 (Fig. 5d,f,h). We next measured anxiety-like behaviour on the basis of ambulation in the elevated plus maze (EPM) test (Fig. 5e,g,i) and in the open-field test (Extended Data Fig. 19). We quantified the time spent in the centre of the open field, the speed in the centre and the time spent in the open arms of the EPM (Fig. 5i). We also measured total ambulation as the total distance travelled

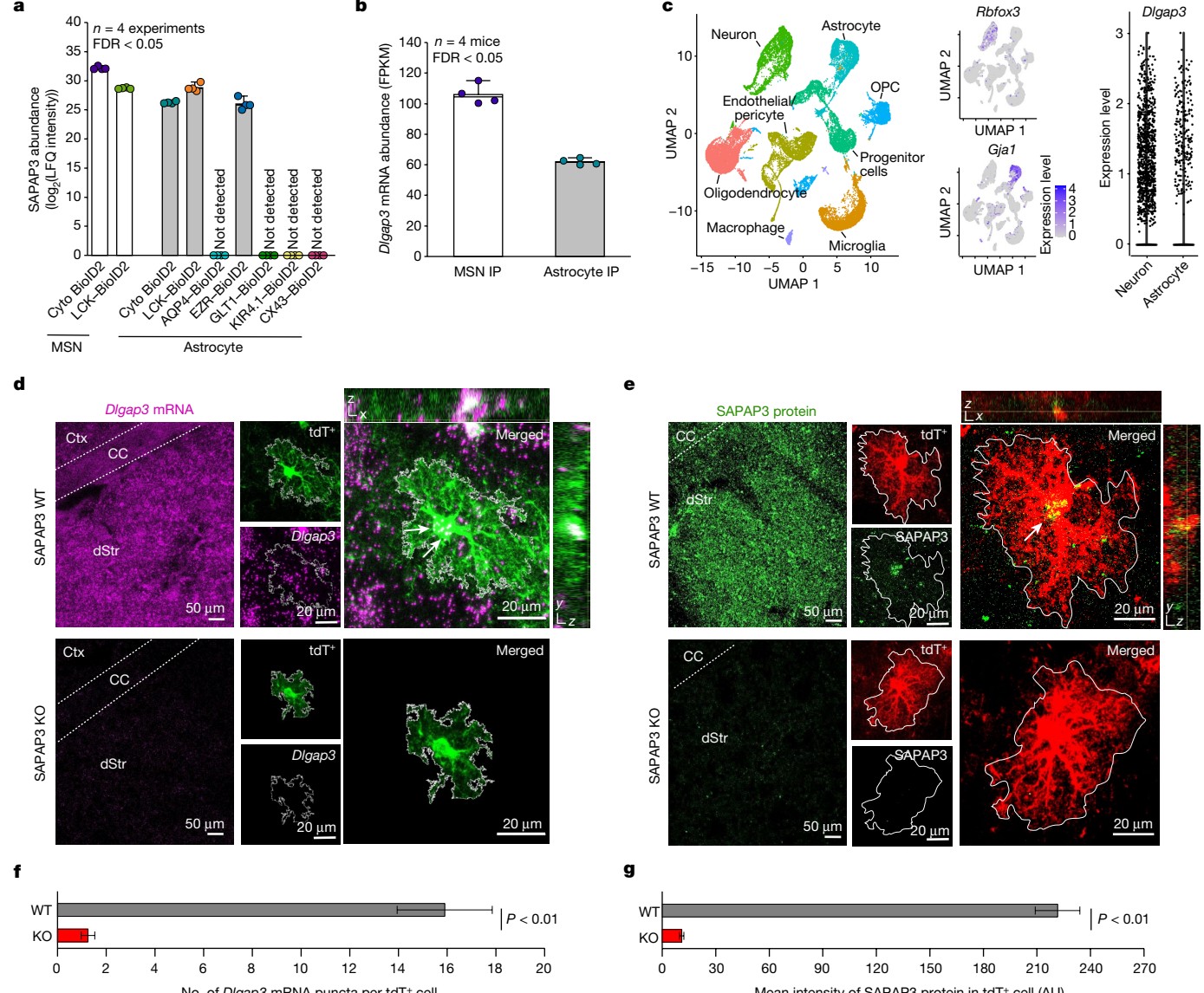

**Fig. 3 | SAPAP3 expression in striatal astrocytes. a**, SAPAP3 protein abundance across the neuronal and astrocytic subcompartments. Not detected, SAPAP3 was not detected. Mean and s.e.m. are shown. **b**, *Dlgap3* mRNA abundance measured in fragments per kilobase of exon per million, mapped fragments (FPKM) in neuronal or astrocyte RiboTag immunoprecipitation (IP). Mean and s.e.m. are shown. **c**, Left, uniform manifold approximation and projection (UMAP) plot of striatal cells (*n* = 31,956 individual cells, replotted from our published scRNA-seq data[36]). Middle, expression of *Rbfox3* in neurons and *Gja1* in the astrocytes. Right, violin plot showing the relative expression level of *Dlgap3* per cell in neurons and astrocytes. Expression level is defined as the log$_2$ normalized gene count per cell. OPC, oligodendrocyte precursor cell. **d**, Representative image of the dorsal striatum (dStr) from either WT or SAPAP3 KO mice labelled by RNAscope in situ hybridization for *Dlgap3* mRNA (purple)

and by IHC for tdTomato[+] (tdT[+]) astrocytes (green). Left images show *Dlgap3* mRNA abundance throughout the entire striatum. Expanded images show dorsal striatum tdTomato[+] astrocytes express *Dlgap3* mRNA (white arrows). **e**, Representative image of the dorsal striatum from either WT or SAPAP3 KO mice labelled by IHC for SAPAP3 protein (green). Left images show SAPAP3 protein abundance in the striatum. Expanded images show tdTomato[+] dorsal striatum astrocytes express SAPAP3 protein (white arrow). **f**, The number of *Dlgap3* mRNA puncta within tdTomato[+] astrocytes in either WT or SAPAP3 KO mice. **g**, Intensity in arbitrary units (AU) of SAPAP3 protein within tdTomato[+] astrocytes in either WT or SAPAP3 KO mice. For **f** and **g**, mean and s.e.m. are shown; *n* = 20 tdTomato[+] astrocytes from 4 mice per group (two-tailed Mann–Whitney test).

in the open-field apparatus and the average total speed (Fig. 5j and Extended Data Fig. 19e). The total distance travelled and the total average speed were similarly rescued by astrocytic or neuronal expression of GFP–SAPAP3 (Fig. 5j). However, ambulation in the centre of the open field and time spent in the open arms of the EPM were only rescued by neuronal GFP–SAPAP3 (Fig. 5i), which suggests a significant effect of neuronal rescue on anxiety-like behaviour. To benchmark these data against a first-line therapeutic effective in some forms of OCD[2] and in SAPAP3 KO mice[4], we assessed the effect of fluoxetine (10 mg kg$^{-1}$ per day for 1 week; Fig. 5k). On the basis of this metric, astrocytic rescue

by GFP–SAPAP3 resulted in beneficial effects comparable to fluoxetine for self-grooming (Extended Data Fig. 19f–h). We summarized the behavioural data with a *Z*-score and compared the per cent recovery by Astro GFP–SAPAP3 and Neuro GFP–SAPAP3 (Fig. 5l). Astrocytic and neuronal GFP–SAPAP3 rescued the distance travelled in the open field, but displayed different degrees of rescue for self-grooming and anxiety-like behaviours.

SAPAP3 is expressed in astrocytes and neurons (Figs. 1–3), and both cell types make contributions to OCD-like phenotypes in mice (Fig. 5). The corticostriatal circuitry is heavily implicated in OCD in humans[37]

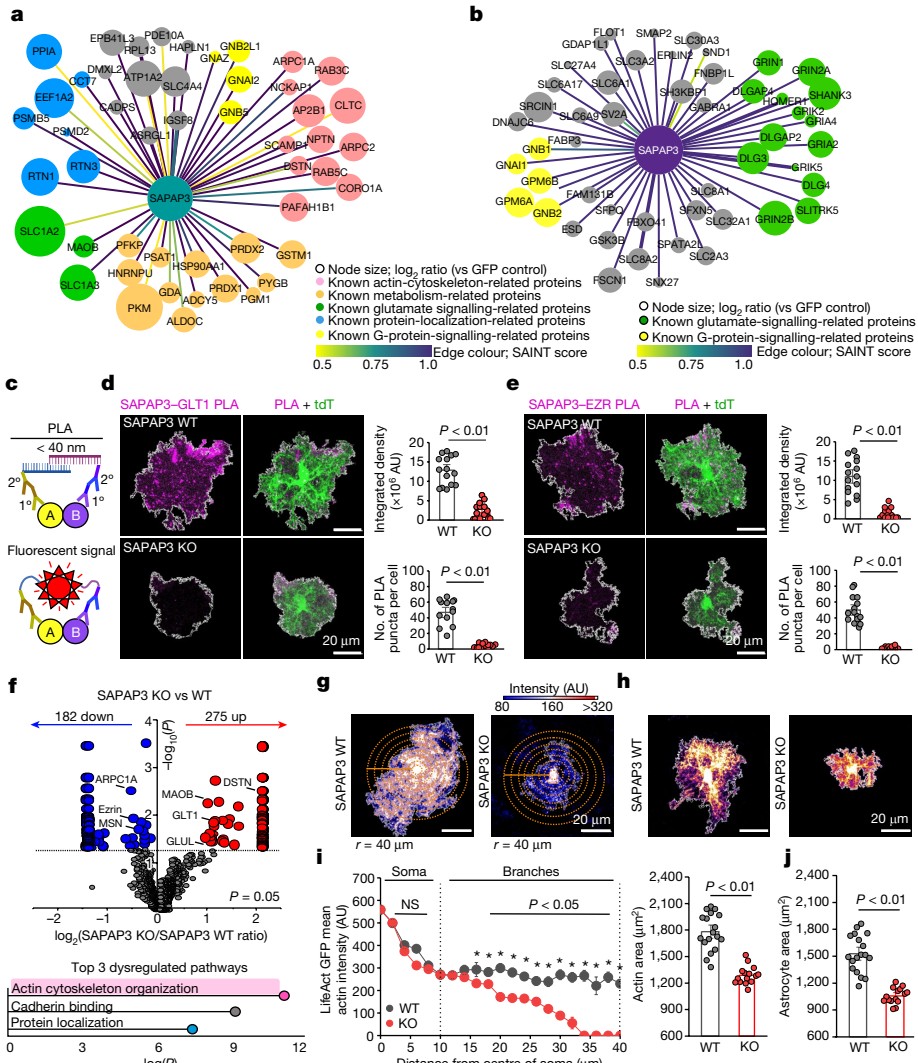

**Fig. 4 | Molecular interactions and cellular mechanisms of SAPAP3. a**, Map of SAPAP3-interacting astrocyte proteins, identifying 49 distinct proteins. Edge colour represents SAINT interaction score. Colour legends denote PANTHER GO terms. **b**, As in **a** but for SAPAP3-interacting neuronal proteins and showing the top 50 distinct proteins. **c**, Schematic of the PLA. **d**, Left, images of PLA puncta for SAPAP3 and GLT1 in tdTomato⁺ astrocytes (WT and SAPAP3 KO). Right, summary graphs. **e**, As in **d** but for SAPAP3 and ezrin proteins (WT and SAPAP3 KO). For **d** and **e**, mean and s.e.m. are shown; $n = 15$ tdTomato⁺ astrocytes from 3 mice per group (integrated density and number of PLA puncta per cell; two-tailed Mann–Whitney test). **f**, Top, differentially displayed astrocyte PM proteins in SAPAP3 KO mice from Astro LCK–BioID2 proteomics. Bold depicts proteins related to the actin cytoskeleton (from **a**). $n = 3$ technical replicates from 5 mice each. Bottom, graph of significant

molecular function Enrichr GO terms for proteins from the top graph. **g**, Images showing LifeAct GFP in WT and SAPAP3 KO astrocytes. Concentric circles (5 μm) were used for intensity measurements. **h**, Images showing WT and SAPAP3 KO tdTomato⁺ astrocytes. **i**, Left, LifeAct GFP mean actin intensity as a function of distance from astrocyte somata (points represent mean intensity from 15–18 cells per group, 4 mice). The error bars depict the s.e.m. (two-way repeated measures analysis of variance (ANOVA) with Bonferroni post hoc test; $P = 0.012$ at 20–40 μm). Right, the astrocyte actin territory area. The mean and s.e.m. are shown; $n = 15$ WT and 18 SAPAP3 KO LifeAct⁺ astrocytes, 4 mice (two-tailed unpaired $t$-test with Welch correction). NS, not significant. **j**. Astrocyte territory area; $n = 15$ WT and 18 SAPAP3 KO tdTomato⁺ astrocytes from 4 mice per group. The mean and s.e.m. are shown (two-tailed unpaired $t$-test with Welch correction).

and mice[38]. To explore links between molecular mechanisms (Figs. 1–4) and behaviour (Fig. 5), we assessed metrics of altered neuronal activity in vivo by evaluating ΔFosB levels, a well characterized marker of increased chronic neuronal activity[39]. We detected increased ΔFosB levels in striatal neurons in SAPAP3 KO mice, which were restored by both astrocyte and neuronal SAPAP3 rescue (Fig. 6a). By contrast, increased ΔFosB levels in cortical neurons of SAPAP3 KO mice (motor cortex and lateral orbitofrontal cortex) were unaffected by striatal astrocyte or neuronal SAPAP3 rescue (Extended Data Fig. 20). This result indicated that behaviourally ameliorative effects of astrocytic and neuronal SAPAP3 rescue originate in the striatum. Furthermore, concomitant with the behavioural rescue and restoration of ΔFosB levels, reduced astrocyte territory sizes and the disrupted EZR–SAPAP3

and GLT1–SAPAP3 interactions that were measured in SAPAP3 KO mice (Fig. 4d–j) were rescued by astrocytic SAPAP3 but not by neuronal SAPAP3 (Fig. 6c,d and Extended Data Fig. 21). Our findings underscore molecular, cellular and behavioural similarities as well as differences in regard to astrocytic and neuronal mechanisms relevant to OCD phenotypes in SAPAP3 KO mice.

## Relationship to human OCD data

To explore the potential relevance of our findings for human OCD, we performed bulk striatal proteomics for WT and SAPAP3 KO mice (Extended Data Fig. 22) to determine how protein changes relate to gene expression alterations in post-mortem tissue from individuals with

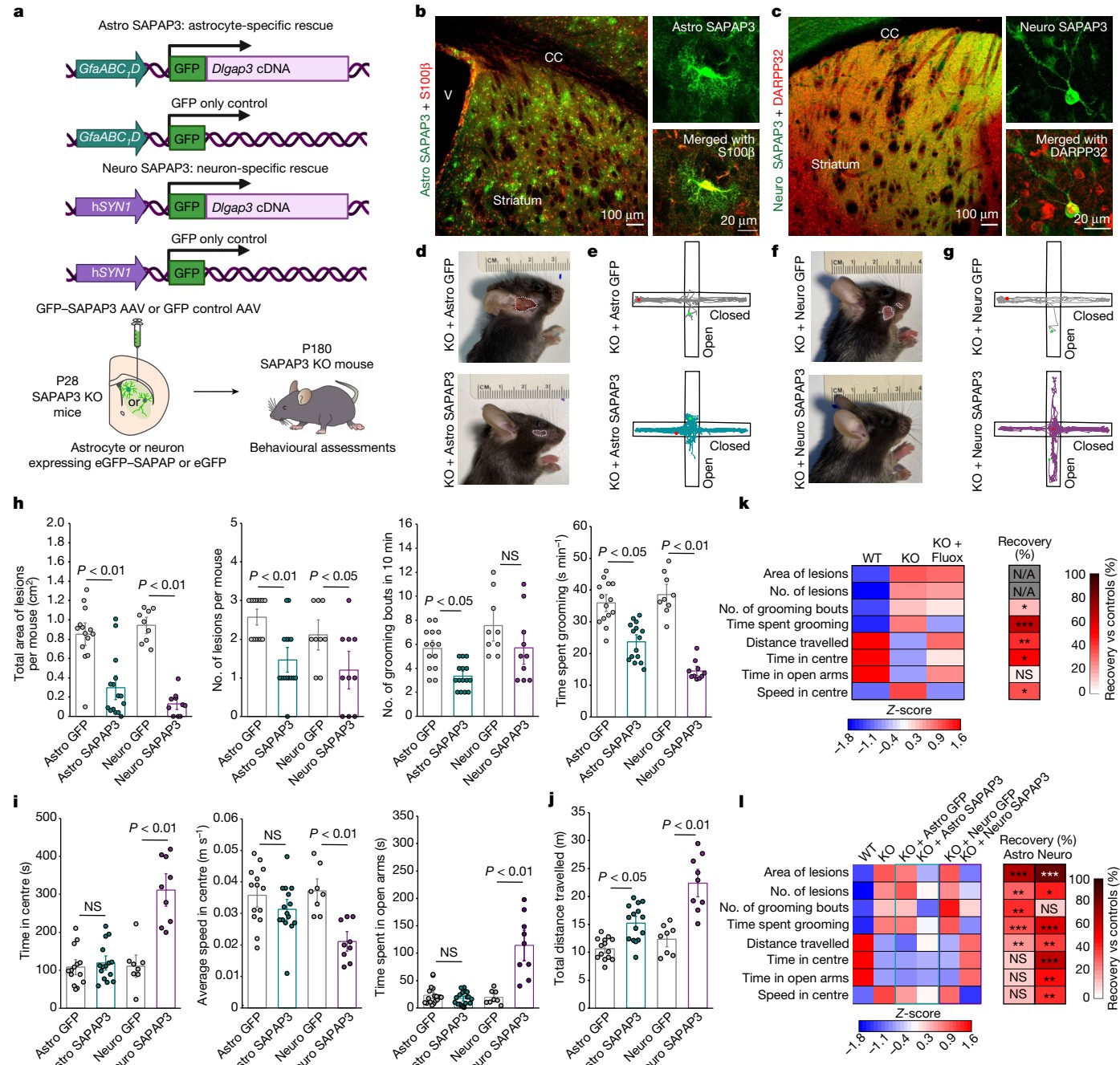

**Fig. 5 | Contributions of astrocytes and neurons to OCD phenotypes in mice. a**, Schematic of cell-specific striatal rescue of SAPAP3. **b**, Striatum injected with Astro SAPAP3 (green). Images show a single S100β⁺ (red) astrocyte expressing Astro SAPAP3 (replicated four times). CC, corpus callosum; V, ventricle. **c**, As in **b** but for Neuro SAPAP3 (green) and DARPP32⁺ (red) neurons expressing Neuro SAPAP3. **d**, Images of lesions from SAPAP3 KO mice injected with Astro GFP or Astro SAPAP3. White outline shows lesion border. **e**, Elevated plus maze traces for SAPAP3 KO mice injected with Astro GFP or Astro SAPAP3. **f**,**g**, As in **d** and **e** but for mice injected with Neuro GFP or Neuro SAPAP3. **h**, Grooming behaviours in SAPAP3 KO mice treated with cell-specific GFP or SAPAP3. **i**, Anxiety-like behaviours in SAPAP3 KO mice treated with either cell-specific GFP or SAPAP3. For **h** and **i**, all behaviours:

Kruskal–Wallis test with Dunn post hoc test. **j**, Locomotor activity in SAPAP3 KO mice treated with cell-specific GFP or cell-specific SAPAP3. Two-way ANOVA with Bonferroni post hoc test. **k**, Heatmap of behavioural Z-scores in WT mice, SAPAP3 KO mice and SAPAP3 KO mice treated with 10 mg kg⁻¹ fluoxetine (intraperitoneal; 7 days). The per cent recovery heatmap shows the average per cent recovery versus the saline control. **l**, Heatmap of behavioural Z-scores in WT and SAPAP3 KO mice or SAPAP3 KO mice treated with cell-specific GFP or SAPAP3. The per cent recovery heatmap shows the average per cent recovery of each cell-specific SAPAP3 rescue versus corresponding controls. Data are mean and the s.e.m. from $n = 14$ mice for Astro GFP, $n = 15$ for Astro SAPAP3, $n = 8$ for Neuro GFP and $n = 9$ for Neuro SAPAP3.

OCD[40] and to astrocyte and neuronal gene expression (Extended Data Fig. 22b). Of the 66 differentially expressed proteins, all were expressed in astrocytes and/or neurons, and genes for 44 were upregulated or downregulated in human OCD[40], with 18 showing similar directional

changes. We next identified the top 30 differentially expressed caudate genes between unaffected controls and human OCD[40]. Many of these genes were highly expressed in astrocytes and neurons and several were within their proteomes (Extended Data Fig. 22c). Next we determined

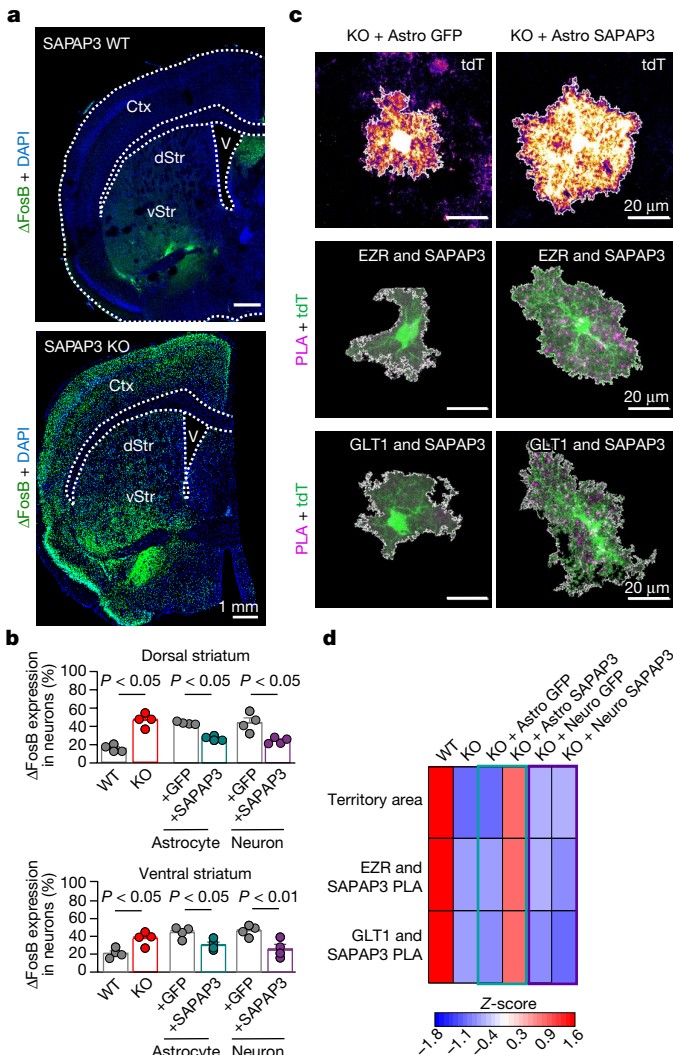

**Fig. 6 | Effect of SAPAP3 loss and rescue on neuronal ΔFosB levels and SAPAP3 interactions. a**, Images of coronal sections containing striatum in WT and SAPAP3 KO mice. Both dorsal striatum (dStr) and ventral striatum (vStr) showed increased numbers of ΔFosB-expressing cells. **b**, The percentage of NeuN⁺ neurons with ΔFosB expression in both dorsal and ventral striatum. Bar denotes the mean, and error bars denote the s.e.m. $n = 4$ animals per group (one-way ANOVA with Bonferroni post hoc test). **c**, Images of tdTomato⁺ striatal astrocytes in SAPAP3 KO mice injected with Astro GFP or Astro SAPAP3 ($n = 20$ cells, 4 mice each). Representative images also show PLA puncta for SAPAP3 and EZR and for SAPAP3 and GLT1 in SAPAP3 KO mice injected with Astro GFP or Astro SAPAP3 ($n = 12$ cells, 4 mice each). Note that in these cases, we used tdTomato as the astrocyte reporter to achieve spectral separation from Astro GFP and GFP-tagged SAPAP3. **d**, Heatmap depicting the $Z$-scores of the measured parameters in astrocytes in WT and SAPAP3 KO mice, SAPAP3 KO mice injected with Astro GFP or with Astro SAPAP3 and SAPAP3 KO mice injected with Neuro GFP or with Neuro SAPAP3. The colour legend shows the $Z$-score. Bar graphs and the raw replicate values are in Extended Data Fig. 21.

astrocytic and neuronal expression for 61 genes associated with, or causal for, repetitive behaviours such as OCD and Tourette's syndrome[41] (Extended Data Fig. 22d). Most showed expression in astrocytes and/or neurons and many were detected in our proteomics data for astrocytes and neurons or as putative SAPAP3 interactors (Extended Data Fig. 22d). These analyses of human data support our findings in mice that molecular changes associated with OCD[40,42] affect signalling in both astrocytes and neurons. Notably, postnatal *Dlgap3* expression in astrocytes and neurons differed in mice (Extended Data Fig. 23), portending future

exploration of how SAPAP3 expression may relate to the emergence of OCD phenotypes during development and adolescence.

## Concluding comments

The relative contributions of astrocytes and neurons to brain disorders has long been discussed, and more recently assessed by RNA-seq[43] and mass spectrometry[44]. These insights, along with the necessity to understand multicellular interactions in the brain, provided the impetus to use tailor-made tools for neurons and astrocytes to determine their proteomes in vivo in a manner that is cell-specific and subcompartment-specific. Our proteomics data for cytosolic and PM compartments revealed shared and distinct proteins and signalling pathways that define the molecular basis for cell-type-specific signalling by astrocytes and neurons. Furthermore, the astrocyte subproteomes defined the molecular basis for distributed physiological functions served by morphologically complex astrocytes. The finding that many proteins were preferentially enriched in astrocyte subproteomes has important implications for understanding pathophysiology during neurodegeneration, injury, stroke, trauma and addiction that are accompanied with altered astrocyte morphology and signalling[20,45]. Overall, our data showed that the relationship between gene and protein expression is not straightforward for astrocytes or their subcompartments. As such, the subcompartment proteomes represent a bounty of previously unknown molecules and pathways, meeting the need for protein as well as gene expression data to comprehensively explore astrocytes and neurons in vivo.

We explored a new discovery concerning astrocytic and neuronal mechanisms relevant to OCD phenotypes that were revealed by our data. This is of interest because OCD is incompletely understood and poorly treated[7]. OCD is characterized by obsessive intrusive thoughts, compulsions manifested as repetitive behaviours and anxiety. OCD is a chronic, disabling psychiatric condition that affects around 2–3% of the population[2]. Classically considered a neuronal disease, OCD involves striatal circuit malfunction[23], but the molecular and cellular basis of the disorder has remained unclear. However, it is emerging that diverse cell types contribute to OCD phenotypes[46,47]. Building on recent work with depression[48] and degeneration[36,44], our experiments showed that astrocyte and neuron SAPAP3 mechanisms are relevant to OCD phenotypes in mice. Our proteomics experiments demonstrated how SAPAP3, a protein shared by astrocytes and neurons and involved in human OCD[2-8], produces effects on OCD-related behavioural phenotypes through distinct astrocyte and neuron molecular interactions, which, within astrocytes, affect the actin cytoskeleton. This is relevant for a larger set of brain diseases involving multicellular molecular dysfunctions, but for which aetiologies remain to be understood and clinically exploited. Therapeutic strategies targeting both astrocytes and neurons may be useful to explore in OCD and other brain disorders.

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

## Methods

### Mouse models

All experiments were conducted in accordance with the National Institutes of Health's Guide for the Care and Use of Laboratory Animals and were approved by the Chancellor's Animal Research Committee at the University of California, Los Angeles (UCLA). Male and female mice aged between 3 and 28 weeks were used in this study (depending on the experiment). Mice were housed in the vivarium managed by the UCLA Division of Laboratory Animal Medicine with a 12 h light–dark cycle and ad libitum access to food and water. WT C57BL/6NTac mice were purchased from Taconic Biosciences. Targeted KO mice for *Dlgap3* (B6.129-Dlgap3tm1Gfng/J) were obtained from the Jackson Laboratory and maintained as a heterozygous line.

### Cell lines

HEK-293 cells (sex: female, RRID: CVCL_0045) were obtained from the American Type Culture Collection and maintained in 25 cm$^2$ cell culture flasks in DMEM/F12 medium with Glutamax (Invitrogen) supplemented with 10% FBS and penicillin–streptomycin. Cells were grown in a humidified cell culture incubator with 95% air and 5% $CO_2$ at 37 °C.

### Striatal astrocyte dissociation

To obtain cell-specific proteomes from adult mice, a common method was used to purify astrocytes by FACS. To conduct such experiments, the cells were prepared by dissociation. To assess astrocyte integrity following this method and to compare the morphology to in situ astrocytes, 8-week-old *Aldh1l1*–eGFP mice were used to purify astrocytes from the striatum by FACS as previously described[28]. In brief, striata from 8 mice were dissected and digested for 45 min at 36 °C in a Petri dish with 2.5 ml papain solution (1× EBSS, 0.46% D-glucose, 26 mM $NaHCO_3$, 50 mM EDTA, 75 U ml$^{-1}$ DNase1, 300 units of papain and 2 mM L-cysteine) while bubbling with 5% $CO_2$ and 95% $O_2$. After this digestion process, the tissue was washed 4 times with ovomucoid solution (1× EBSS, 0.46% D-glucose, 26 mM $NaHCO_3$, 1 mg ml$^{-1}$ ovomucoid, 1 mg ml$^{-1}$ BSA and 60 U ml$^{-1}$ DNase1) and mechanically dissociated with 2 fire-polished borosilicate glass pipettes. A bottom layer of concentrated ovomucoid solution (1× EBSS, 0.46% D-glucose, 26 mM $NaHCO_3$, 5.0 mg ml$^{-1}$ ovomucoid, 5.5 mg ml$^{-1}$ BSA and 25 U ml$^{-1}$ DNase1) was added to the cell suspension. The tubes were centrifuged at room temperature at 300$g$ for 10 min, and the resultant pellet was resuspended in D-PBS with 0.02% BSA and 13 U ml$^{-1}$ of DNase1, and filtered through a 20 μm mesh. FACS was performed using a FACSAria II instrument (BD Bioscience) with a 70 μm nozzle using standard methods at the UCLA Cell Sorting Core. The cells from each mouse were then placed on a glass slide and imaged using a UplanFL x40 0.8 water immersion lens on a confocal scanning microscope (Olympus FV3000) using Fluoview (Olympus) software. Representative images are shown in Extended Data Fig. 1c.

### Lucifer yellow dye filling

This method for filling astrocytes in lightly fixed tissue has been previously described[28]. For lucifer yellow iontophoresis, 8-week-old WT mice were transcardially perfused with 10 ml Ringer's solution with 0.02% lidocaine, followed by 4% paraformaldehyde. Brains were post-fixed at room temperature for 1.5 h and then washed in 0.1 M PBS for 10 min. Next 100 μm coronal sections were cut using a Pelco Vibrotome 3000 and then placed in ice-cold PBS for the duration of the experiment. Lucifer yellow CH di-lithium salt (10 mg; Sigma) was dissolved in 1 ml 5 mM KCl solution and filtered before use. Sharp (200 MΩ) glass electrodes were pulled from a borosilicate glass capillary with a filament (outer diameter of 1.0 mm, inner diameter of 0.58 mm). Electrodes were gravity filled. Sections were transferred to a solution of room temperature PBS for filling. Astrocytes were identified using infrared differential interference contrast (IR-DIC) and then impaled with the sharp electrode. Lucifer yellow was injected into the cell by passing current (2 mA) for 20 s: three times with 15–20 s pauses in between. Sections were post-fixed completely with 4% paraformaldehyde at 4 °C before mounting on glass slides and imaged using a UplanFL x40 0.8 water immersion lens on a confocal scanning microscope (Olympus FV3000) using Fluoview (Olympus) software. Representative images are shown in Extended Data Fig. 1c.

### Generation of AAVs

AAVs were generated as previously described[49]. In brief, the BioID2 sequence with a HA tag was PCR amplified from pAAV-BioID2-Linker-BioID2-HA plasmid (a gift from S. Soderling)[50] and incorporated into a pZac2.1 vector using an In-fusion cloning kit (Takara Bio) to generate plasmids pZac2.1-GfaABC$_1$D-BioID2-Linker-BioID2-HA (Astro BioID2) and pZac2.1-hSYN1-BioID2-Linker-BioID2-HA (Neuro BioID2), which carry the astrocyte-specific promoter and the human synapsin-1 neuronal promoter, respectively. To generate the subcompartment-specific BioID2 constructs, the Astro BioID2 plasmid was cut by restriction digest at the XhoI site. cDNAs for each subcompartment were amplified by PCR, and the sequences were cloned into the XhoI site using an In-fusion cloning kit (Takara Bio). The *Aqp4* sequence was PCR amplified from GeneCopoeia plasmid EX-Mm20326; the *Cx43* sequence was amplified from msfGFP-Cx43 plasmid (Addgene, 69024); the *Glt1* sequence was amplified from GeneCopoeia plasmid Mm27106; the *Ezr* sequence was amplified from GeneCopoeia plasmid Mm2129m; the *Kcnj10* sequence was amplified from GfaABC$_1$D-eGFP-Kir4.1 plasmid (Addgene, 52874). The subcompartment GFP controls were generated by digesting the resulting subcompartment BioID2 plasmids with BamHI, which removed the BioID2 sequence, and inserting the PCR-amplified eGFP sequence from the GfaABC$_1$D-eGFP-Kir4.1 plasmid (Addgene, 52874) using an In-fusion cloning kit (Takara Bio). All BioID2 plasmids and their GFP counterparts were sequenced and sent to the Penn Vector core for AAV production. All SAPAP3-associated plasmids were sent to Virovek for AAV production. Astrocyte-specific constructs were produced in AAV2 and AAV5 serotypes (AAV2/5), whereas neuron-specific constructs were produced in the AAV1 serotype. AAV titres are provided in Supplementary Table 9. For HEK-293 cell transfection, the BioID2 sequence was cloned into a pcDNA3.1 vector between BamHI sites to generate pcDNA3.1-CMV-BioID2-Linker-BioID2-HA and pcDNA3.1-CMV-Lck-BioID2-Linker-BioID2-HA. To generate the eGFP–SAPAP3 and the HA-BioID2-SAPAP3 AAVs, the SAPAP3 sequence was amplified from GeneCopoeia plasmid Mm16264. pZac2.1-GfaABC$_1$D-BioID2-Linker-BioID2-HA and pZac2.1-hSYN--BioID2-Linker-BioID2-HA were digested with BamHI, and the SAPAP3 sequence was cloned into the BamHI sites using an In-fusion cloning kit (Takara Bio), which generated pZac2.1-GfaABC$_1$D-SAPAP3 and pZac2.1-hSYN1-SAPAP3. This resulting plasmid was then digested with BmtI. The eGFP sequence was PCR-amplified from GfaABC$_1$D-eGFP-Kir4.1 plasmid (Addgene, 52874) and cloned into the pZac2.1-SAPAP3 plasmids at the BmtI site with an In-fusion cloning kit (Takara Bio), which produced pZac2.1-GfaABC$_1$D-eGFP-SAPAP3 and pZac2.1-hSYN1-eGFP-SAPAP3. The HA–BioID2 sequence was PCR-amplified from an existing pZac2.1 plasmid and cloned into the pZac2.1-SAPAP3 plasmids at the BmtI site using an Infusion cloning kit (Takara Bio), which produced pZac2.1-GfaABC$_1$D-HA-BioID2-SAPAP3 and pZac2.1-hSYN1-HA–BioID2–SAPAP3. Generation of pZac2.-GfaABC$_1$D-3xHA-SAPAP3 consisted of digesting the pZac2.1-GfaABC$_1$D-SAPAP3 plasmid with BmtI and ligating an annealed oligonucleotide with the 3×HA sequence into the BmtI sites using T4 ligase. To generate the astrocyte-specific LifeAct-eGFP AAV, the LifeAct-eGFP sequence was amplified from pEGFP-C1-LifeAct-EGFP (Addgene, 58470). pZac2.1-GfaABC$_1$D-BioID2-Linker-BioID2-HA plasmid was digested with BamHI, and the LifeAct-eGFP sequence was cloned into the BamHI sites using an In-fusion cloning kit (Takara Bio), which produced pZac2.1-GfaABC$_1$D-LifeAct-eGFP. The 20 new AAV plasmids generated in this study are listed in Supplementary Table 9 along with their Addgene accession identifiers.

## HEK-293 cell studies

HEK-293 cells were prepared for transfection by plating onto 6-well plates, and transfection was performed when cells reached 80% confluence. For expression in HEK-293 cells, 0.4 µg plasmid DNA was transfected using Effectene transfection reagent (Qiagen). Biotin (Sigma, B4501) was dissolved in sterile 0.1 M PBS to make a 1 mM stock solution. The stock solution was added to the HEK-293 cell medium to obtain a final concentration of 50 mM biotin. After 48 h of transfection, the cells were trypsinized and transferred to poly-D-lysine coverslips. After 24 h, the cells were washed once with 0.1 M PBS and then fixed with 10% formalin for 10 min. Cells were washed in 0.1 M PBS and then incubated with agitation in a blocking solution containing 5% NGS in 0.1 M PBS with 0.2% Triton-X (Sigma) in PBS for 1 h at room temperature. The cells were then incubated with agitation in mouse anti-HA primary antibody (1:1,000; BioLegend, 901514) diluted in 0.1 M PBS with 5% NGS at 4 °C overnight. Cells were incubated with agitation with the following secondary antibodies and fluorophores in a solution containing 5% NGS in 0.1 M PBS for 2 h at room temperature (1:1,000; Molecular Probes): Alexa Fluor 546 goat anti-mouse (A11003); and streptavidin, Alexa Fluor 488 conjugate (S11223). The cells were rinsed 3 times in 0.1 M PBS for 10 min each. The coverslips containing the cells were then mounted onto microscope slides in fluoromount-G. Fluorescence images were taken using a UplanSApo ×40 1.30 NA oil-immersion objective lens on a confocal laser-scanning microscope (FV10-ASW, Olympus). Laser settings were the same for all cells. Images represent maximum intensity projections of optical sections with a step size of 1.0 µm. For western blotting, the cells were directly lysed in RIPA buffer (150 mM NaCl, 50 mM Tris pH 8.0, 1% Triton-X, 0.5% sodium deoxycholate, 0.1% SDS and Halt protease inhibitor (Thermo Scientific, 78429)). The cells were homogenized using a cell scratcher and the lysate was incubated at 4 °C while rotating for 30 min. The samples were sonicated for 10 min and then centrifuged at 16,500g for 10 min at 4 °C. The supernatant was collected and the protein concentrations were measured using a BCA protein assay (Thermo Scientific). The samples were then mixed with 2× Laemmli solution (Bio-Rad) containing β-mercaptoethanol. The samples were boiled at 95 °C for 10 min before being electrophoretically separated by 10% SDS–PAGE (30 µg protein per lane) and transferred onto a nitrocellulose membrane (0.45 µm). The membrane was incubated with agitation in a solution containing 5% BSA, 0.1% Tween-20 and 0.1 M PBS for 1 h. The membrane was probed with streptavidin–HRP (Sigma, RABHRP3) at 1:250 for 2 h. The membrane was then treated with Pierce chemiluminescence solution for 1 min and imaged. The blot was incubated overnight at 4 °C with rabbit anti-β-actin (1:1,000; Abcam, ab8227). IRDye 800CW anti-rabbit (1:10,000; Li-Cor) was used as the secondary antibody, and images were acquired on a Li-Cor odyssey infrared imager. Signal intensities at expected molecular weights were quantified using ImageJ. The streptavidin signal levels were normalized to β-actin by dividing the streptavidin signal intensity by the β-actin signal intensity.

## Stereotaxic microinjections

All surgical procedures were conducted under general anaesthesia using continuous isoflurane (induction at 5%, maintenance at 2% v/v) in 6-week-old C57/BL6NTac mice unless otherwise stated. Anaesthetic depth was continuously monitored and adjusted when necessary. After induction of anaesthesia, mice were fitted into the stereotaxic frame (David Kopf Instruments), their noses placed into a veterinary-grade anaesthesia ventilation system (VetEquip) and their heads were secured using blunt ear bars. Mice were subcutaneously administered with 0.1 mg kg$^{-1}$ of buprenorphine (Bupranex) before surgery. The surgical incision site was cleaned 3 times with 10% povidone iodine and 70% ethanol (v/v). A skin incision was made followed by craniotomies (1–2 mm in diameter) above the left parietal cortex using a small steel burr (NeoBurr) powered by a high-speed drill (Midwest Tradition). Sterile saline (0.9%) was applied onto the skull to reduce heating caused by drilling. One craniotomy was made for unilateral injections, and two craniotomies were made for bilateral injections. The injections were carried out using the stereotaxic apparatus to guide the placement of bevelled glass pipettes (1B100-4, World Precision Instruments). For the left striatum, the following coordinates were used: 0.8 mm anterior to bregma, 2 mm lateral to the midline, and 2.4 mm from the pial surface. AAV was injected using a syringe pump (SmartTouch Pump, World Precision Instruments). Following AAV microinjection, the glass pipette was left in place for at least 10 min before slow withdrawal. Surgical wounds were closed with external 5-0 nylon sutures. Following surgery, animals were allowed to recover overnight in cages placed on a low-voltage heating pad. Buprenorphine was administered 2 times a day for 48 h after surgery. Trimethoprim sulfamethoxazole was provided in food to the mice for 1 week. Virus-injected mice were used for experiments at least 3 weeks after surgery. All AAV titres were adjusted to $1.0 \times 10^{13}$ genome copies per ml with sterile 0.1 M PBS. The following viruses were used: 0.5 µl AAV2/5 GfaABC$_1$D-BioID2-Linker-BioID2-HA; 0.5 µl AAV2/5 GfaABC$_1$D-Lck-BioID2-Linker-BioID2-HA; 0.5 µl AAV2/5 GfaABC$_1$D-Aqp4-BioID2-Linker-BioID2-HA; 0.5 µl AAV2/5 GfaABC$_1$D-Cx43-BioID2-Linker-BioID2-HA; 0.5 µl AAV2/5 GfaABC$_1$D-Ezr-BioID2-Linker-BioID2-HA; 0.5 µl AAV2/5 GfaABC$_1$D-Glt1-BioID2-Linker-BioID2-HA; 0.5 µl AAV2/5 GfaABC$_1$D-HA-BioID2-Linker-BioID2-Kir4.1; 0.5 µl AAV2/5 GfaABC$_1$D-tdTomato (Addgene, 44332-AAV5); 0.5 µl AAV2/5 GfaABC$_1$D-Lck-GFP (Addgene, 105598-AAV5); 0.5 µl AAV2/5 GfaABC$_1$D-Aqp4-eGFP; 0.5 µl AAV2/5 GfaABC$_1$D-Cx43-eGFP; 0.5 µl AAV2/5 GfaABC$_1$D-Ezr-eGFP; 0.5 µl AAV2/5 GfaABC$_1$D-Glt1-eGFP; 0.5 µl AAV2/5 GfaABC$_1$D-eGFP-Kir4.1; 0.5 µl AAV1 hSYN1-BioID2-Linker-BioID2-HA; 0.5 µl AAV1 hSYN1-Lck-BioID2-Linker-BioID2-HA; 0.5 µl AAV1 hSYN1-eGFP (Addgene, 50465-AAV1); 0.5 µl AAV1 hSYN1-Lck-GFP; 0.5 µl AAV2/5 GfaABC$_1$D-Rpl22-HA (Addgene, 111811); and 0.5 µl AAV1 hSYN1-Rpl22-HA. For co-IP experiments, 0.1 µl of GfaABC$_1$D-Ezr–eGFP was injected with 0.1 µl of GfaABC$_1$D-3×HA-SAPAP3 (Addgene, 190200) and 0.1 µl of GfaABC$_1$D-Glt1-eGFP was injected with 0.1 µl of GfaABC$_1$D-3×HA-SAPAP3.

## AAVs for SAPAP3 and LifeAct GFP

Surgical procedures for SAPAP3 KO mice were conducted as described above. In brief, GFP or GFP–SAPAP3 AAVs was bilaterally injected into the striatum of 3–4-week-old mice through 2 sites at 3 locations per hemisphere. At each of the injection sites, the microinjection needle was advanced to the deepest (ventral) position for the first injection, whereas the additional injections were made every 0.3 mm while withdrawing the injection needle. The coordinates from bregma were as follows: injection site 1: anterior, 0.5 mm; medial–lateral, 1.5 mm, dorsal–ventral, 2.9 mm, 2.6 mm and 2.3 mm from the pial surface; injection site 2: anterior, 0.5 mm, medial–lateral, −1.5 mm; dorsal–ventral, 2.9 mm, 2.6 mm and 2.3 mm from the pial surface. For each injection location, 150 nl of virus was injected and the needle was left in place for 5 min after each injection. These injection procedures were chosen to cover most of the striatum (dorsal and ventral). All AAV titres were adjusted to $1.0 \times 10^{13}$ genome copies per ml with sterile 0.1 M PBS. The following viruses were used: AAV 2/5 GfaABC$_1$D-LifeAct-eGFP (Addgene, 190199); AAV2/5 GfaABC$_1$D-eGFP-SAPAP3; AAV2/5 GfaABC$_1$D-eGFP; AAV1 hSYN1-eGFP-SAPAP3; and AAV1 hSYN1-eGFP (Addgene, 50465-AAV1). Litters with multiple SAPAP3 KO mice were split between experimental groups.

As shown in the figures, we confirmed that intrastriatal microinjection of AAV2/5-delivered cargo was cell selective and restricted to the striatum, although there was some expression proximal to the needle tract in cells of the cortex and sometimes of the corpus callosum. We suspect such expression occurred in all previous studies that used viruses, as it is impossible to reach subcortical brain structures without advancing the needle through the overlying tissue. Consequently, all studies that use microinjections (including ours) need to be interpreted with this anatomical caveat in mind. We have previously reported and discussed this issue in regard to our surgical procedures[51].

### In vivo BioID2 protein biotinylation
Three weeks after AAV microinjection, mice were treated with a subcutaneous injection of biotin at 24 mg kg$^{-1}$ (Millipore Sigma, RES1052B-B7) dissolved in sterile 0.1 M PBS once per day for 7 consecutive days. The mice were processed 16 h after the last biotin injection.

### IHC analysis
Mice were transcardially perfused with chilled 0.1 M PBS followed by 10% formalin. After gentle removal of the skull, the brains were post-fixed in 10% formalin for 6 h. The brains were then cryoprotected in 30% sucrose with 0.1 M PBS solution for at least 48 h at 4 °C. Serial coronal sections (40 μm) containing striatum were prepared using a cryostat microtome (Leica) at −20 °C and processed for IHC. Sections were washed 3 times in 0.1 M PBS for 10 min each and then incubated in a blocking solution consisting of 5% NGS in 0.1 M PBS with 0.2% Triton-X for 1 h at room temperature with agitation. Sections were then incubated in primary antibodies diluted in 5% NGS in 0.1 M PBS solution overnight at 4 °C. The following primary antibodies were used: mouse anti-HA (1:1,000; BioLegend, 901514); rabbit anti-HA (1:1,000; Abcam, ab9110); rabbit anti-S100β (1:1,000; Abcam, ab13970); rabbit anti-NeuN (1:1,000; Cell Signaling, 12943S); guinea pig anti-NeuN (1:1,000; Synaptic Systems, 266004), rabbit anti-DARPP32 (1:1,000; Abcam, ab40801); guinea pig anti-DARPP32 (1:1,000; Frontier Institute, DARPP-Gp-A250); chicken anti-GFP (1:1,000; Abcam, ab13970); rabbit anti-SAPAP3 (1:100; a gift from G. Feng); mouse anti-RFP (1:500; Rockland, 600906379); rabbit anti-PAICS (1:100; Invitrogen, 92985); mouse anti-Nebl (1:100; Santa Cruz Biotechnology, 393784); rabbit anti-Slc4a4/NBC1 (1:100; Novus, NBP32020); rabbit anti-Arpc1a (1:100; Invitrogen, 102339); rabbit anti-Faim2 (1:100; Origen, TP300196); rabbit anti-Hepacam (1:100; Novus Biologicals, 04983); mouse anti-APC (1:500; Abcam, ab16794); rabbit anti-Olig2 (1:500; Millipore, AB9610); and rabbit anti-ΔFosB (1:500; Cell Signaling Technology, 14695S). Sections were then incubated with the following secondary antibodies for 2 h at room temperature (1:1,000; Molecular Probes): Alexa Fluor 488 goat anti-chicken (A11039); Alexa Fluor 647 goat anti-rabbit (A21244); Alexa Fluor 546 goat anti-mouse (A11003); Alexa Fluor 488 goat anti-rabbit (A11008); and streptavidin, Alexa Fluor 488 conjugate (S11223). The free-floating sections were mounted on microscope slides in fluoromount-G. Fluorescence images were taken using a UplanFL ×40 1.30 NA oil-immersion or a PlanApo N ×60 1.45 NA oil-immersion objective lens on a confocal laser-scanning microscope (FV3000, Olympus) using Fluoview (Olympus) software. Laser settings were kept the same within each experiment. Images represent maximum intensity projections of optical sections with a step-size of 1.0 μm. Images were processed using ImageJ. Cell counting was done on maximum intensity projections using the Cell Counter plugin; only cells with somata completely in the region of interest were counted. Colocalization analysis was conducted using the Fiji/ImageJ Coloc2 plugin.

### RNAscope and IHC
Fixed-frozen tissue was processed as described above. Serial coronal sections (14 μm) containing striatum were prepared using a cryostat microtome (Leica) at −20 °C and mounted immediately onto glass slides. Dual ISH-IHC was performed using a Multiplex RNAscope (v.2) with integrated co-detection work flow (ACDBio 323180 and 323110). Sections were baked for 30 min at 60 °C. Sections were washed for at least 15 min in 0.1 M PBS and then incubated in 1× Target Retrieval reagents for 5 min at 95 °C. After washing with ddH$_2$O twice, the sections were dehydrated with 100% ethanol and dried at room temperature. Sections were then incubated with primary antibody rabbit anti-S100β (1:500; Abcam, ab13970), rabbit anti-RFP to amplify astrocyte-specific tdTomato signal (1:500; Rockland, 600-401-379), and guinea pig anti-NeuN (1:500; Synaptic Systems, 266004) overnight at 4 °C. Sections were then incubated with Protease Pretreat-4 solution (ACDBio,

322340) for 30 min at 40 °C. The sections were washed with ddH$_2$O twice for 1 min each and then incubated with probe for 2 h at 40 °C: Mm-DLGAP3-C3 (ACDBio, 586091-C3), Mm-Mapt-C1 (ACDBio, 400351) and Mm-Tjp1-C1 (ACDBio, 440411). The sections were incubated in Amp 1-FL for 30 min, AMP 2-FL for 15 min, AMP 3-FL for 30 min and AMP 4-FL for 15 min at 40 °C while washing in 1× wash buffer (ACDBio, 310091) between incubations. The HRP-C3 signal was developed with Opal 690 fluorophore (Akoya Biosciences, FP1497001KT). The HRP-C1 signal was developed with Opal 520 fluorophore (Akoya Biosciences, FP1487001KT). All incubations at 40 °C or 60 °C were performed in a HybEZ hybridization system (ACDBio). Last, sections were incubated with Alexa Fluor goat secondary antibodies described in the IHC section for 45 min at room temperature. Images were obtained in the same way as for IHC (described above) with a step size of 0.5 μm. Images were processed using ImageJ. Astrocyte somata were labelled with S100β signals, and the number of puncta within each soma was measured. Astrocyte territories were labelled with tdTomato signals, and the number of puncta within each territory was measured.

### PLA analysis
The PLA detects native interacting proteins within about 40 nm of each other. Fixed-frozen tissue was processed as described above. Serial coronal sections (20 μm) containing striatum sparsely labelled with astrocyte-specific AAV2/5 GfaABC$_1$D-tdTomato were prepared using a cryostat microtome (Leica) at −20 °C and mounted immediately onto glass slides. PLAs were performed using the Sigma-Aldrich Duolink PLA fluorescence protocol (Sigma-Aldrich DUO92101 and DUO92013). Sections were baked for 30 min at 60 °C. Sections were washed for at least 15 min in 0.1 M PBS. After washing, sections were incubated in 1× citrate pH 6.0 antigen retrieval buffer (Sigma, C999) for 10 min at 90 °C. After washing 3 times in 0.2% Triton-X in PBS (PBS-T), the sections were blocked for 45 min at room temperature with 5% donkey serum (Sigma, D9663) in PBS-T. Sections were then incubated with the following primary antibodies overnight at 4 °C: rabbit anti-SAPAP3 (1:50); mouse anti-GLT1 (1:50; Santa Cruz Biotechnology, sc-365634); mouse anti-EZR (1:100; BioLegend, 866401); and guinea pig anti-RFP (1:500; Synaptic Systems, 390004). Sections were then incubated with PLA probe cocktail containing the anti-rabbit PLUS primer probe (DUO92002) and the anti-mouse MINUS primer probe (DUO92004) for 1 h at 37 °C. The sections were washed twice in 1× wash buffer A (DUO82049). Sections were then incubated with ligation solution containing ligase for 30 min at 37 °C. Sections were once again washed twice with 1× wash buffer A and then incubated with amplification solution containing DNA polymerase for at least 3 h at 37 °C. Sections were then washed twice in 1× wash buffer B (DUO82049) and then washed in 0.01× wash buffer B. To amplify the tdTomato signal, sections were then incubated with donkey anti-guinea pig Cy3 (1:500; Jackson ImmunoResearch, 706-165-148) for 45 min at room temperature. Sections were washed twice with PBS and then coverslips were mounted with DuoLink mounting medium with DAPI (DUO82040). Images were obtained in the same way as for IHC (described above) with a step size of 0.5 μm. Images were processed using ImageJ. Astrocyte territories were labelled with tdTomato signals, and the number of puncta and integrated intensity within each territory were measured.

### Western blotting
Mice were decapitated and striata were dissected and homogenized with a dounce and pestle in ice-cold RIPA buffer (150 mM NaCl, 50 mM Tris pH 8.0, 1% Triton-X, 0.5% sodium deoxycholate, 0.1% SDS and Halt protease inhibitor (Thermo Scientific, 78429)). The homogenate was incubated at 4 °C while spinning for 1 h. The homogenate was sonicated and then centrifuged at 4 °C for 10 min at 15,600$g$. The clarified lysate was collected and the protein concentration was measured using a BCA protein assay kit (Thermo). The samples were then processed as described above and analysed as stated above with 30 μg of protein loaded into each gel well.

## Co-IP

To validate SAPAP3–EZR and SAPAP3–GLT1 interactions, we used recombinant proteins expressed in striatal astrocytes in vivo for co-IP experiments. This is because SAPAP3, EZR and GLT1 are expressed natively in multiple cell types (see http://dropviz.org/) and immunoprecipitation of the endogenous proteins would not be specific for astrocytes. To this end, mice were injected with one of the following combinations in the striatum: GfaABC$_1$D-3×HA-SAPAP3 and GfaABC$_1$D-Glt1-eGFP, GfaABC$_1$D-3×HA-SAPAP3 and GfaABC$_1$D-Ezr-eGFP, GfaABC$_1$D-3×HA-SAPAP3 only, GfaABC$_1$D-Glt1-eGFP only or GfaABC$_1$D-Ezr-eGFP only. Mice were decapitated and striata were dissected and homogenized with a dounce and pestle in ice-cold lysis buffer (25 mM HEPES pH 7.5, 150 mM NaCl, 1 mM EDTA, 1% NP-40, 5 mM NaF, 1 mM orthovanadate and Halt protease inhibitor cocktail (Thermo Scientific 78429)). The homogenate was incubated at 4 °C while spinning for 1 h. The homogenate was then centrifuged at 4 °C for 15 min at 15,000$g$. The supernatant was further cleared by ultracentrifugation at 100,000$g$ for 30 min at 4 °C. The cleared lysate was then incubated with GFP-trap beads (Chromotek, gtma) or incubated with anti-HA tag beads (Thermo, 88836) overnight at 4 °C. The beads were then washed 3 times with wash buffer (25 mM HEPES pH 7.5, 500 mM NaCl, 1 mM EDTA, 1% NP-40, 5 mM NaF, 1 mM orthovanadate and Halt protease inhibitor cocktail). Next 1× Laemmli buffer was prepared (Bio-Rad, 1610737) and added to the beads. The beads were boiled in the Laemmli buffer for 10 min at 95 °C. The bead supernatants were cooled and loaded on a SDS–PAGE gel for western blot analyses as described above. The following primary antibodies were used: chicken anti-GFP (1:1,000; Abcam, ab13970) and rabbit anti-HA (1:1,000; Abcam, ab9110). The following secondary antibodies were used: goat anti-rabbit plus 647 (1:2,000; Invitrogen, A32733) and goat anti-chicken plus 555 (1:2,000; Invitrogen, A32932).

## Behavioural evaluations

Behavioural tests were performed during the light cycle between the hours of 10:00 and 14:00. Mice were assessed at 6 months of age or 5 months after AAV microinjection. All experimental mice were transferred to the behaviour testing room at least 30 min before testing to acclimatize to the environment and to reduce stress. The temperature and humidity of the experimental rooms were kept at $23 \pm 2$ °C and $55 \pm 5$%, respectively. The brightness of the experimental room was kept dimly lit unless otherwise stated. Background noise (60–65 dB) was generated using a white noise box (San Diego Instruments). Litters with multiple SAPAP3 KO mice were split between experimental groups. The mice were randomly allocated to a group as they became available and of age from the breeding colony in alternation. Experimenters were blinded to group allocation during data collection and analyses.

## Self-grooming behaviour

The procedure of self-grooming behaviour measurement was adapted from previously published work[25]. The recording was conducted at 35 lux. Mice were placed individually into plastic cylinders (15 cm in diameter and 35 cm tall) and allowed to habituate for 20 min. Self-grooming behaviour was recorded for 10 min. A timer was used to assess the cumulative time spent in self-grooming behaviour, which included paw licking, unilateral and bilateral strokes around the nose, mouth and face, paw movement over the head and behind the ears, body-fur licking, body scratching with hind paws, tail licking and genital cleaning. The number of self-grooming bouts and rearing bouts was also counted. Separate grooming bouts were considered when the pause was more than 5 s or if behaviours other than self-grooming occurred.

## Assessment of skin lesions

Mice were anaesthetized with 5% isoflurane and 1% O$_2$ through a veterinary-grade anaesthesia ventilation system (VetEquip). Mice were placed on an opaque Plexiglass board, and photos of their head and torso were taken bilaterally. Images were scaled with a ruler (Fine Science Tools) and the images were analysed using ImageJ software. Measurements were all scaled to the ruler on ImageJ.

## Open-field test

The open-field chamber was illuminated at 35 lux. The open-field chamber consisted of a square arena (28 × 28 cm) enclosed by walls made of opaque white Plexiglass (19 cm tall). The periphery of the arena was defined as the area within 2.5 cm adjacent to the walls of the chamber and the centre of the arena was defined as the area 2.5 cm away from the chamber walls. Activity was then recorded for 20 min using a camera (Logitech) located immediately above the open-field chamber. Anymaze video analysis software was used to quantify the time spent in the centre, the total distance travelled and speed.

## Elevated plus maze

All four arms of the elevated plus maze were illuminated at 25 lux. The elevated plus maze consisted of arms that were 30 × 7 cm with closed arm walls with a height of 20 cm. The maze was elevated 65 cm above floor level and was placed in the centre of the room away from other stimuli. Mice were placed in the centre of the maze facing an open arm. Mice were recorded for 10 min using a camera (Logitech) located above the maze. Anymaze video analysis software was used to quantify time spent in open arms and the per cent time spent in open arms.

## Whole-tissue protein extraction

Striata from SAPAP3 WT or KO mice were lysed in 200 µl lysis buffer (8 M urea, 50 mM Tris-HCl pH 8.2, 75 mM NaCl, 5 mM EDTA, 5 mM EGTA, 10 mM sodium pyrophosphate and protease inhibitor cocktail). Tissue was dounce homogenized and extracts were sonicated for 10 min at 80% power in a bath sonicator. Samples were centrifuged at 15,000$g$ for 20 min at 4 °C to remove debris. The supernatant was collected and then further processed.

## In vivo BioID2 biotinylated protein pull-down

Purification of biotinylated proteins was conducted as previously described[50]. Each AAV BioID2 probe and its counterpart AAV GFP control were injected into the striatum of 6-week-old C57/BL6NTac mice. At 3 weeks after AAV microinjection, biotin (Millipore Sigma, RES1052B-B7) was subcutaneously injected at 24 mg kg$^{-1}$ for 7 consecutive days. All mice were processed 16 h after the last biotin injection. Eight mice were used for each biotinylated protein purification, and each purification was performed independently four times for a total of four technical replicates. Mice were decapitated and striata were microdissected. Striata from 4 mice were dounce homogenized with 600 µl of lysis buffer 1 (1 mM EDTA, 150 mM NaCl, 50 mM HEPES pH 7.5 supplemented with Halt protease inhibitor (Thermo Scientific, 78429)). Immediately after homogenization, 600 µl of lysis buffer 2 (2% sodium deoxycholate, 2% Triton-X, 0.5% SDS, 1 mM EDTA, 150 mM NaCl and 50 mM HEPES pH 7.5) was added. The lysed samples were sonicated for 5 min at 60% power and then centrifuged at 15,000$g$ for 15 min at 4 °C. The resulting supernatant was then ultracentrifuged at 100,000$g$ for 30 min at 4 °C. SDS was added to the supernatant to obtain a final concentration of 1%. The sample was then boiled at 95 °C for 5 min. The sample was cooled on ice and incubated with 35 µl of equilibrated anti-pyruvate carboxylase (5 µg; Abcam, 110314) conjugated agarose beads (Pierce 20398) for 4 h at 4 °C while rotating. Subsequently, the sample was centrifuged at 2,000 r.p.m. for 5 min at 4 °C and the supernatant was incubated with 80 µl NeutrAvidin agarose at 4 °C overnight while rotating. The NeutrAvidin beads were then washed twice with 0.2% SDS, twice with wash buffer (1% sodium deoxycholate, 1% Triton-X and 25 mM LiCl), twice with 1 M NaCl and 5 times with 50 mM ammonium bicarbonate. Proteins bound to the agarose were then eluted in elution buffer (5 mM biotin, 0.1% Rapigest SF surfactant and 50 mM ammonium

bicarbonate) at 60 °C for a minimum of 2 h. The final protein concentration was measured by BCA.

## MS analysis

Protein samples were subjected to reduction using 5 mM Tris (2-carboxyethyl) phosphine for 30 min, alkylated by 10 mM iodoacetamide for another 30 min and then sequentially digested with Lys-C and trypsin at a 1:100 protease-to-peptide ratio for 4 and 12 h, respectively. The digestion reaction was terminated by the addition of formic acid to 5% (v/v) with centrifugation. Each sample was then desalted with C18 tips (Thermo Scientific, 87784) and dried in a SpeedVac vacuum concentrator. The peptide pellet was reconstituted in 5% formic acid before analysis by LC–MS/MS.

Tryptic peptide mixtures were loaded onto a 25-cm long, 75 µm inner diameter fused-silica capillary, packed in-house with bulk 1.9 µM ReproSil-Pur beads with 120 Å pores. Peptides were analysed using a 140 min water–acetonitrile gradient delivered by a Dionex Ultimate 3000 UHPLC (Thermo Fisher Scientific) operated initially at a 400 nl min$^{-1}$ flow rate with 1% buffer B (acetonitrile solution with 3% DMSO and 0.1% formic acid) and 99% buffer A (water solution with 3% DMSO and 0.1% formic acid). Buffer B was increased to 6% over 5 min, at which time the flow rate was reduced to 200 nl min$^{-1}$. A linear gradient from 6 to 28% of buffer B was applied to the column over the course of 123 min. The linear gradient of buffer B was then further increased to 28–35% for 8 min followed by a rapid ramp-up to 85% for column washing. Eluted peptides were ionized using a Nimbus electrospray ionization source (Phoenix S&T) by application of a distal voltage of 2.2 kV. Spectra were collected using data-dependent acquisition on an Orbitrap Fusion Lumos Tribrid mass spectrometer (Thermo Fisher Scientific) with a MS1 resolution of 120,000 followed by sequential MS2 scans at a resolution of 15,000. Data generated by LC–MS/MS were searched using the Andromeda search engine integrated into the MaxQuant[52] bioinformatics pipelines against the UniProt *Mus musculus* reference proteome (UP000000589) and then filtered using a 'decoy' database-estimated FDR < 1%. LFQ was carried out by integrating the total extracted ion chromatogram of peptide precursor ions from the MS1 scan. These LFQ intensity values were used for protein quantification across samples. Statistical analysis of differentially expressed proteins was done using the Bioconductor package limma (v.3.54). To generate a list of proteins with high confidence, all mitochondrial proteins, including carboxylases and dehydrogenases, were manually filtered as they are artefacts of endogenously biotinylated proteins. Proteins with $\log_2(FC) > 1$ and FDR < 0.05 versus GFP controls were considered putative hits and used for subsequent comparison between subcompartments and cell types. A comparison between subcompartments and cell types was also performed with limma utilizing the same thresholds ($\log_2(FC) > 1$ and FDR < 0.05). To account for variations in pull-down efficiency, all proteins and their LFQ values were normalized to pyruvate carboxylase (UniProt identifier Q05920). Downstream analysis was conducted only on proteins with non-zero LFQ values in three or more experimental replicates. Data analysis for whole bulk tissue analyses was carried out in an identical manner, except samples were normalized by median intensity.

The GO enrichment analysis for cellular compartments and biological function was performed using PANTHER overrepresentation test (GO database released 1 January 2020) with FDR < 0.05 and with all *M. musculus* genes used as reference and with STRING (https://string-db.org) with a confidence score of 0.5 and with all *M. musculus* genes used as a reference. GO pathway analysis for the astrocytic subcompartments was performed using Enrichr (https://maayanlab.cloud/Enrichr/).

## Protein–protein interaction analysis

Network figures were created using Cytoscape (v.3.8), with nodes corresponding to the gene name for proteins identified in the proteomic analysis. A list of protein–protein associations and putative interactions from published datasets was assembled using STRING. STRING database interactions were filtered to include affinity purification–MS validations. We caution that such interactions are putative and have been labelled as such, and further validations are necessary on a case-by-case basis, as we have done for the key interactions reported herein. In all networks, the node size is proportional to the fold enrichment over GFP control. To identify interactors of SAPAP3 protein, significance analysis of interactome (SAINTexpress) was used with a FDR cut-off of 0.05. The Bioconductor artMS package was used to re-format the MaxQuant results (evidence.txt file) to make them compatible with SAINTexpress.

## RNA-seq analysis

RNA extraction from striatal astrocytes and neurons was performed using standard methods. In brief, RiboTag AAV (AAV2/5 GfaABC$_1$D-Rpl22-HA or AAV1 hSYN1-Rpl22-HA) was injected into the dorsal striatum of adult C57BL/6NTac mice at 6 weeks of age. For RNA extraction from SAPAP3 KO mice, RiboTag AAV was injected into the dorsal striatum at 4.5 months of age. Freshly microdissected striata were collected and individually homogenized. RNA was extracted from 10–20% of cleared lysate as the input sample, which contained RNA from all striatal cell types. The remaining lysate was incubated with mouse anti-HA antibody (1:250; BioLegend, 901514) with rotation for 4 h at 4 °C followed by addition of IgG magnetic beads (Invitrogen, Dynabeads 110.04D). The samples were left for overnight incubation at 4 °C. The beads were then washed three times in high-salt solution. RNA was purified from the immunoprecipitate and the corresponding input samples using a Qiagen RNAeasy kit (Qiagen, 74034). RNA concentration and quality were assessed using an Agilent 2100 Bioanalyzer. RNA samples with a RNA integrity number greater than 7 were used for multiplexted library preparation with Nugen Ovation RNA-seq System V2. For each experiment, all samples were multiplexed into a single pool to avoid batch effects[53]. Sequencing was performed on an Illumina NextSeq 4000 for 2× 75 to produce at least 45 million reads per sample. Demultiplexing was performed using the Illumina Bcl2fastq2 (v.2.17) program. Reads were aligned to the mouse mm10 reference genome using the STAR spliced read aligner[54]. Approximately 70% of the reads mapped specifically to the mouse genome and were used for downstream analyses. Differential gene expression analysis was performed on genes with FPKM > 5 in at least 4 samples per condition and $\log_2(FC) > 1$ or $< -1$ using the Bioconductor package limmaVoom (v.3.36) with the FDR threshold set at <0.05. Differentially expressed genes that were more than twofold higher in the immunoprecipitated samples than the input samples were designated as astrocyte-enriched or neuron-enriched differentially expressed genes. RNA-seq data have been deposited within the Gene Expression Omnibus repository (https://www.ncbi.nlm.nih.gov/geo) with the accession identifier GSE184773.

## Human and mouse datasets in OCD

The 61 genes associated with human OCD and Tourette's syndrome were obtained from Phenopedia (https://phgkb.cdc.gov; accessed January 2021). The genes were chosen on a threshold of at least two publications. A total of 63 OCD and 23 Tourette's syndrome genes were obtained. When compared, 15 genes overlapped between the OCD and Tourette's syndrome lists. The 61 genes plotted represent genes that have homologues in mice and were detected at any quantity (FPKM > 0) in our mouse RNA-seq studies.

## Quantification and statistical analyses

Data from every experiment represent at least four replicates. All statistical tests, unless otherwise stated, were run in OriginPro 2018 and GraphPad InStat3. Data are presented as the mean ± s.e.m. along with the individual data points. The results of statistical comparisons, *n* numbers and significance levels are shown in the figure panels or in the

figure legends along with the average data. *N* is defined as the number of cells or mice on a case-by-case basis throughout the manuscript. We determined whether each set of data was normally distributed using GraphPad Instat3 and OriginPro 2018. If the data were normally distributed, we used parametric tests, whereas if they were not normally distributed, we used nonparametric tests. Paired and unpaired Student's two-tailed *t*-tests (as appropriate), two tailed Mann–Whitney tests and one-way and two-way analysis of variance tests were used for most statistical analyses with significance declared at $P < 0.05$. When $P$ values were greater than 0.05, they are stated as not significant. When the $P$ value was less than 0.01, it is stated as <0.01. All proteomics and transcriptomics analyses used a statistical FDR < 0.05 unless otherwise stated. All mice were assigned to particular experimental groups at random. No data points were excluded from any experiment. Replicate values and the results of statistical tests are provided in Supplementary Tables 7 and 8.

### Reporting summary

Further information on research design is available in the Nature Portfolio Reporting Summary linked to this article.

## Data availability

All the proteomics data are available at the Proteomics Identification Database with accession identifier PXD029257. The UniProt reference proteome used was UP000000589 for *M. musculus*. The RNA-seq data are available at Gene Expression Omnibus with accession identifier GSE184773. All proteomics data are provided in Supplementary Tables 1–5. The analysed RNA-seq data are provided in Supplementary Table 6. All raw replicate data values used to generate the figures and the associated statistical tests are provided in Supplementary Tables 7 and 8.

## Code availability

No custom software was used. For proteomics, LFQ was carried out using MaxQuant with an integrated search engine, Andromeda (https://www.maxquant.org/). Principal component data visualization was conducted with the R package Factoextra fviz (v.1.0.6) (https://rpkgs.datanovia.com/factoextra/reference/fviz_pca.html). Differential protein expression and enrichment analysis was conducted using the Bioconductor R package limma (v.3.54) (https://bioconductor.org/packages/release/bioc/html/limma.html). Protein network visualization, including STRING analysis, was conducted using Cytoscape (v.3.8) (https://apps.cytoscape.org/apps/stringapp). The artMS package (v.1.16) (https://bioconductor.org/packages/release/bioc/html/artMS.html) was used to re-format the Maxquant results (evidence.

txt file) to make them compatible with the SAINTexpress program. Protein interaction probability scoring was done using SAINT (http://saint-apms.sourceforge.net/Main.html). For RNA-seq, differential gene expression and enrichment analysis used the R package limmaVoom (v.3.36) to process RNA counts (https://rdrr.io/bioc/limma/man/voom.html), and batch correction was done using the R package ComBat (v.3.46) (https://rdrr.io/bioc/sva/man/ComBat.html).

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

**Acknowledgements** J.S.S. was supported by the National Science Foundation Graduate Research Fellowship Program (NSF-GRFP; DGE-2034835) and by the UCLA Eugene V. Cota-Robles Fellowship. This work was supported by the National Institutes of Health (R35 NS111583, DA047444), an Allen Distinguished Investigator Award, a Paul G. Allen Frontiers Group advised grant of the Paul G. Allen Family Foundation, and the Ressler Family Foundation (to B.S.K.). We thank staff at the UCLA Neuroscience Genomics Core and F. Gao for assistance with RNA-seq and analysis; X. Yu for discussions and advice on analysis of RNA-seq and protein data, behaviour analysis and for helping establish the SAPAP3 KO mouse line; F. Endo for advice with IHC, imaging and single-cell sequencing analysis; L. Wu for help with AAV injections for the revision experiments; A. Adhikari and M. La-Vu for advice on mouse anxiety behaviours and analysis; V. Pandey for mass spectrometry help for the revision experiment in Fig. 4f; M. Gangwani from the Khakh Laboratory for sharing the hSYN1-Rpl22HA plasmid; G. Feng for the SAPAP3 antibody; S. Soderling for the BirA plasmid; T. Vondriska for use of the sonicator; and X. Yu, A. Huang and K. Linker for comments and discussions.

**Author contributions** J.S.S. performed most of the experiments, including plasmid cloning, stereotaxic injections, IHC, RNAscope, western blotting, protein extraction, sample preparation for all proteomics experiments, proteomics data analysis, RNA-seq data analysis and mouse behavioural experiments. Y.J.-A. performed the LC–MS/MS experiments and analysed the proteomics data. J.C. performed some of the plasmid cloning and performed HEK-293 immunocytochemistry and western blot experiments. B.D.-C. performed FACS isolation, analysed FACS-isolated astrocytes and guided J.C. S.L.M. performed Lucifer yellow iontophoresis, analysed Lucifer yellow filled astrocytes and helped with early behavioural analyses. J.A.W. helped troubleshoot the proteomics work, guided J.S.S. and Y.J.-A. with LC–MS/MS data analyses, and with Y.J., was responsible for mass spectrometer operation. B.S.K. conceived the study, directed the experiments, assembled the figures with help from J.S.S. and wrote the manuscript with help from J.S.S. All authors commented on the manuscript, which was finalized by B.S.K. and J.S.S.

**Competing interests** The authors declare no competing interests.

**Additional information**
**Correspondence and requests for materials** should be addressed to Baljit S. Khakh.

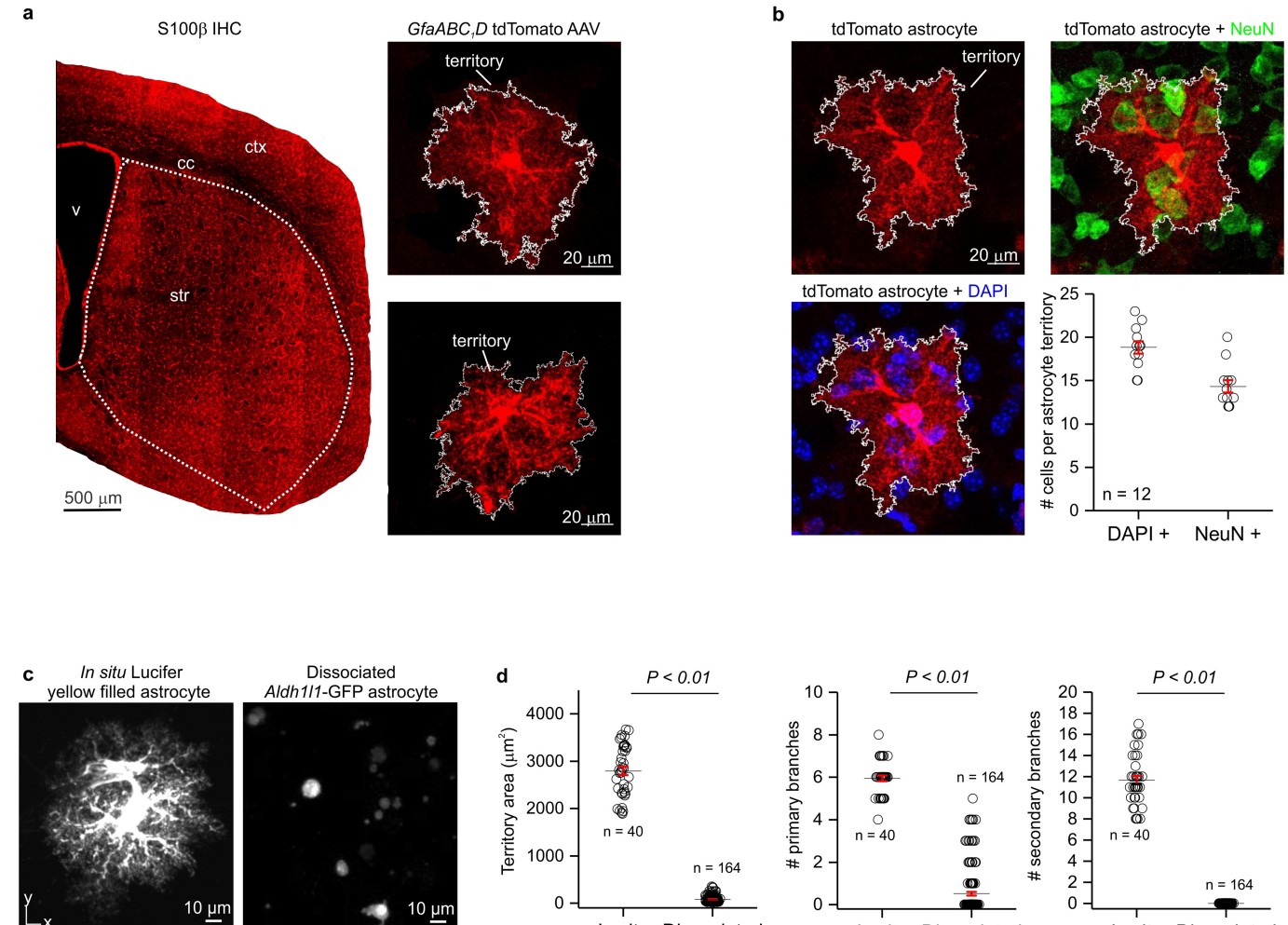

**Extended Data Fig. 1 | Morphologically complex striatal astrocytes lose their complexity upon dissociation. a**. Representative image of S100β immunostaining in a mouse coronal section containing the striatum. Subpanels show representative striatal astrocytes sparsely expressing cytosolic tdTomato fluorescent protein after *GfaABC₁D* tdTomato AAV microinjection. The experiment was replicated five times. **b**. Striatal astrocyte sparsely labeled with tdTomato and co-stained with nuclear marker DAPI and neuronal marker NeuN. Quantification shows DAPI+ nuclei and NeuN+ cell bodies found in a single tdTomato+ astrocyte's territory. Mean and SEM are shown; (n = 12 tdTomato+ cells per group from 4 animals). **c**. *In situ* Lucifer yellow iontophoretically filled striatal astrocyte from 8-week-old mice. Right panel shows GFP+ striatal astrocytes after dissociation for fluorescence activated cell sorting (FACS) from 8-week-old mice. **d**. Quantification of territory area, number of primary branches, and number of secondary branches showed that dissociated astrocytes display decreased cellular complexity. Mean and SEM are shown; (n = 40 cells from 15 mice for Lucifer yellow and n = 164 cells from 8 mice for *Aldh1l1*-GFP, respectively; territory area and number of primary branches: two-tailed student's unpaired t-test, P = 0.0002; number of secondary branches: two-tailed one-sample t-test, P = 0.003).

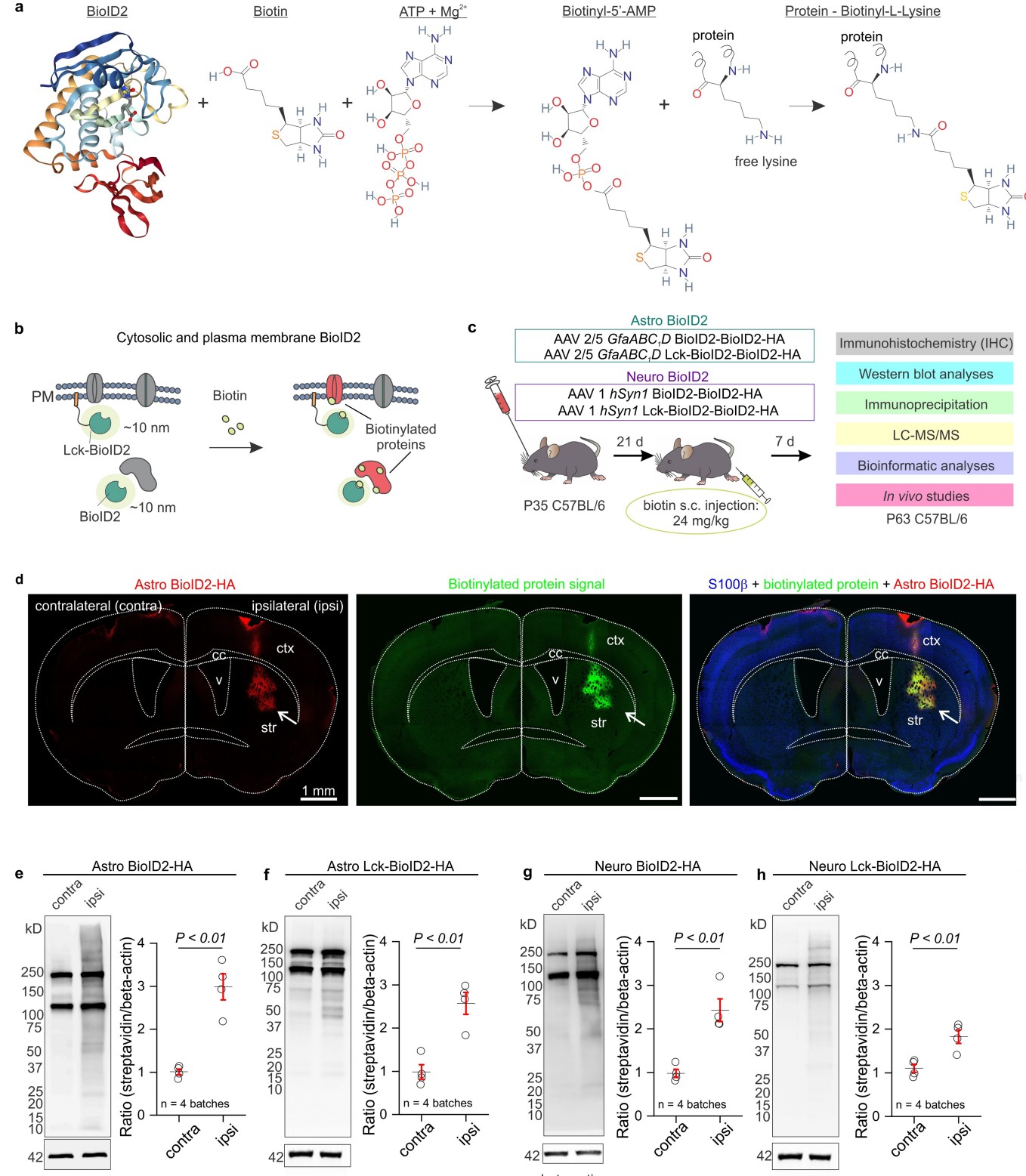

**a**

BioID2 Biotin ATP + Mg²⁺ Biotinyl-5'-AMP Protein - Biotinyl-L-Lysine

protein

free lysine

protein

**b**

Cytosolic and plasma membrane BioID2

PM

Lck-BioID2 ~10 nm

Biotin

BioID2 ~10 nm

Biotinylated proteins

**c**

Astro BioID2
AAV 2/5 *GfaABC₁D* BioID2-BioID2-HA
AAV 2/5 *GfaABC₁D* Lck-BioID2-BioID2-HA

Neuro BioID2
AAV 1 *hSyn1* BioID2-BioID2-HA
AAV 1 *hSyn1* Lck-BioID2-BioID2-HA

Immunohistochemistry (IHC)
Western blot analyses
Immunoprecipitation
LC-MS/MS
Bioinformatic analyses
*In vivo* studies

P35 C57BL/6 21 d biotin s.c. injection: 24 mg/kg 7 d P63 C57BL/6

**d**

Astro BioID2-HA Biotinylated protein signal S100β + biotinylated protein + Astro BioID2-HA

contralateral (contra) ipsilateral (ipsi)

cc ctx v str 1 mm

**e** Astro BioID2-HA **f** Astro Lck-BioID2-HA **g** Neuro BioID2-HA **h** Neuro Lck-BioID2-HA

contra ipsi

kD
250
150
100
75
50
37
25
20
15
10
42
beta-actin

*P < 0.01*
Ratio (streptavidin/beta-actin)
n = 4 batches
contra ipsi

kD
250
150
100
75
50
37
25
20
15
10
42
beta-actin

*P < 0.01*
Ratio (streptavidin/beta-actin)
n = 4 batches
contra ipsi

kD
250
150
100
75
50
37
25
20
15
10
42
beta-actin

*P < 0.01*
Ratio (streptavidin/beta-actin)
n = 4 batches
contra ipsi

kD
250
150
100
75
50
37
25
20
15
10
42
beta-actin

*P < 0.01*
Ratio (streptavidin/beta-actin)
n = 4 batches
contra ipsi

**Extended Data Fig. 2** | See next page for caption.

**Extended Data Fig. 2 | Proteomics workflow and western blot data.**
**a**. Schematic of the BioID2 biotinylation reaction in the presence of biotin, $Mg^{2+}$, and ATP. Free lysine residues on a protein are biotinylated. **b**. Diagram of BioID2 biotin labeling in mammalian cells. Proximal (~10 nm) proteins are biotinylated upon the addition of exogenous biotin. **c**. Schematic of the cell and compartment specific *in vivo* BioID2 experimental design using adeno-associated viruses (AAVs). **d**. 40 μm coronal section of mouse brain microinjected in the dorsal striatum with AAV 2/5 *GfABC1D* BioID2-HA (Astro BioID2) and injected subcutaneously with biotin for 7 days. Arrows show the site of Astro BioID2-HA injection and the site of increased biotinylation.
**e**. Western blot analysis of brain unilaterally microinjected with Astro BioID2. Dark bands at 130 kD and 250 kD show the endogenously biotinylated mitochondrial proteins, Pyruvate carboxylase and acetyl-CoA carboxylase. Graph depicting the streptavidin signal intensity divided by the β-actin signal intensity for each data point. Black horizontal line depicts the mean, red lines depict the SEM (n = 4 mice; two-tailed paired t-test, P = 0.0052). **f**. Western blot analysis of brain unilaterally microinjected with plasma membrane localized Astro Lck-BioID2. Graph depicting the streptavidin signal intensity divided by the β-actin signal intensity for each data point. Black horizontal line depicts the mean, red lines depict the SEM (n = 4 mice; two-tailed paired t-test, P = 0.0088). **g**. Western blot analysis of brain unilaterally microinjected with AAV1 *hSyn1* BioID2-HA (Neuro BioID2). Graph depicting the streptavidin signal intensity divided by the β-actin signal intensity for each data point. Black horizontal line depicts the mean, red lines depict the SEM (n = 4 mice; two-tailed paired t-test, P = 0.015). **h**. Western blot analysis of brain unilaterally microinjected with plasma membrane localized Neuro Lck-BioID2. Graph depicting the streptavidin signal intensity divided by the β-actin signal intensity for each data point. Black horizontal line depicts the mean, red lines depict the SEM (n = 4 mice; two-tailed paired t-test, P = 0.0023). For blot source data, see Supplementary Fig. 1.

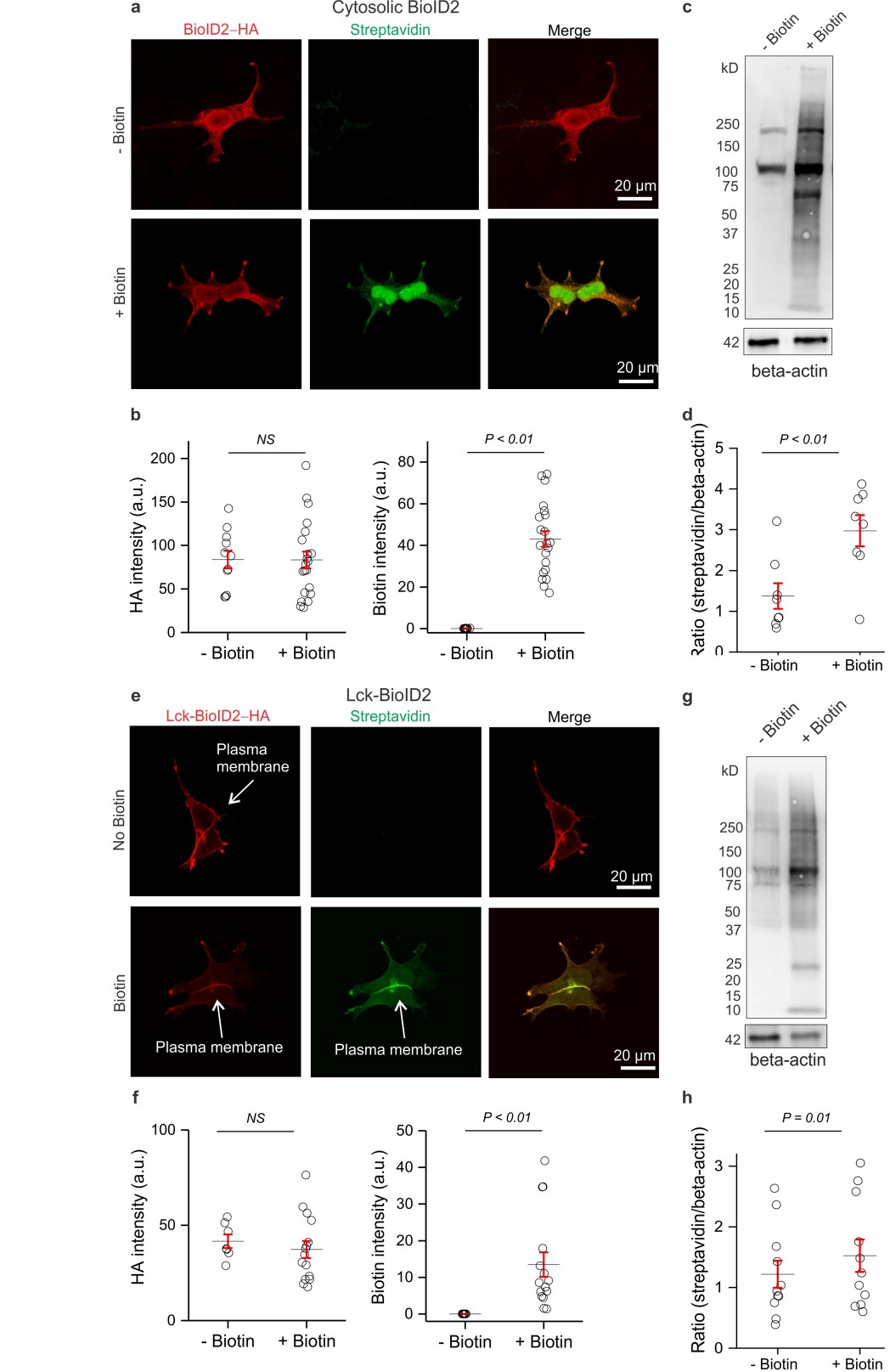

**Extended Data Fig. 3** | See next page for caption.

**Extended Data Fig. 3 | Validation experiments in HEK-293T cells.**
**a**. Representative images of HEK-293T cells transfected with the cytosolic BioID2 construct and then treated with either biotin in PBS or solely PBS (no biotin). The cells were immunostained with both anti-HA antibody (red) and a fluorophore conjugated streptavidin probe (green). **b**. Quantification of HA intensity and biotin intensity (arbitrary units) in cells transfected with cytosolic BioID2. Black horizontal line depicts the mean, red lines depict the SEM (n = 11 cells in the no biotin group and n = 21 cells in the biotin group from 4 transfections; two-tailed unpaired t-test, P = 0.97 and P = 0.0001, respectively). **c**. Representative western blot of HEK-293T cells transfected with cytosolic BioID2 and treated with either biotin in PBS or solely PBS (no biotin). **d**. Western blot quantification of HEK293T cells transfected with cytosolic BioID2 and treated with either biotin in PBS or solely PBS (no biotin). Graph depicting the streptavidin signal intensity divided by the β-actin signal intensity for each data point. Black horizontal line depicts the mean, red lines depict the SEM (n = 8 coverslips from 3 transfections; two-tailed unpaired t-test, P = 0.0062). **e**. Representative images of HEK-293T cells transfected with the plasma membarane Lck-BioID2 construct and then treated with either biotin in PBS or solely PBS (no biotin). The cells were immunostained with both anti-HA antibody (red) and a fluorophore conjugated streptavidin probe (green). Arrows show the plasma membrane localization. **f**. Quantification of HA intensity and biotin intensity (arbitrary units) in cells transfected with Lck-BioID2. Black horizontal line depicts the mean, red lines depict the SEM (n = 8 cells in the no biotin group and n = 21 cells in the biotin group from 4 transfections; two-tailed unpaired t-test, P = 0.91 and P = 0.0084, respectively). **g**. Representative western blot of HEK-293T cells transfected with Lck-BioID2 and treated with either biotin in PBS or solely PBS (no biotin). **h**. Western blot quantification of HEK293T cells transfected with Lck-BioID2 and treated with either biotin in PBS or solely PBS (no biotin). Graph depicting the streptavidin signal intensity divided by the β-actin signal intensity for each data point. Black horizontal line depicts the mean, red lines depict the SEM (n = 11 coverslips from 3 transfections; two-tailed unpaired t-test, P = 0.01). For blot source data, see Supplementary Fig. 1.

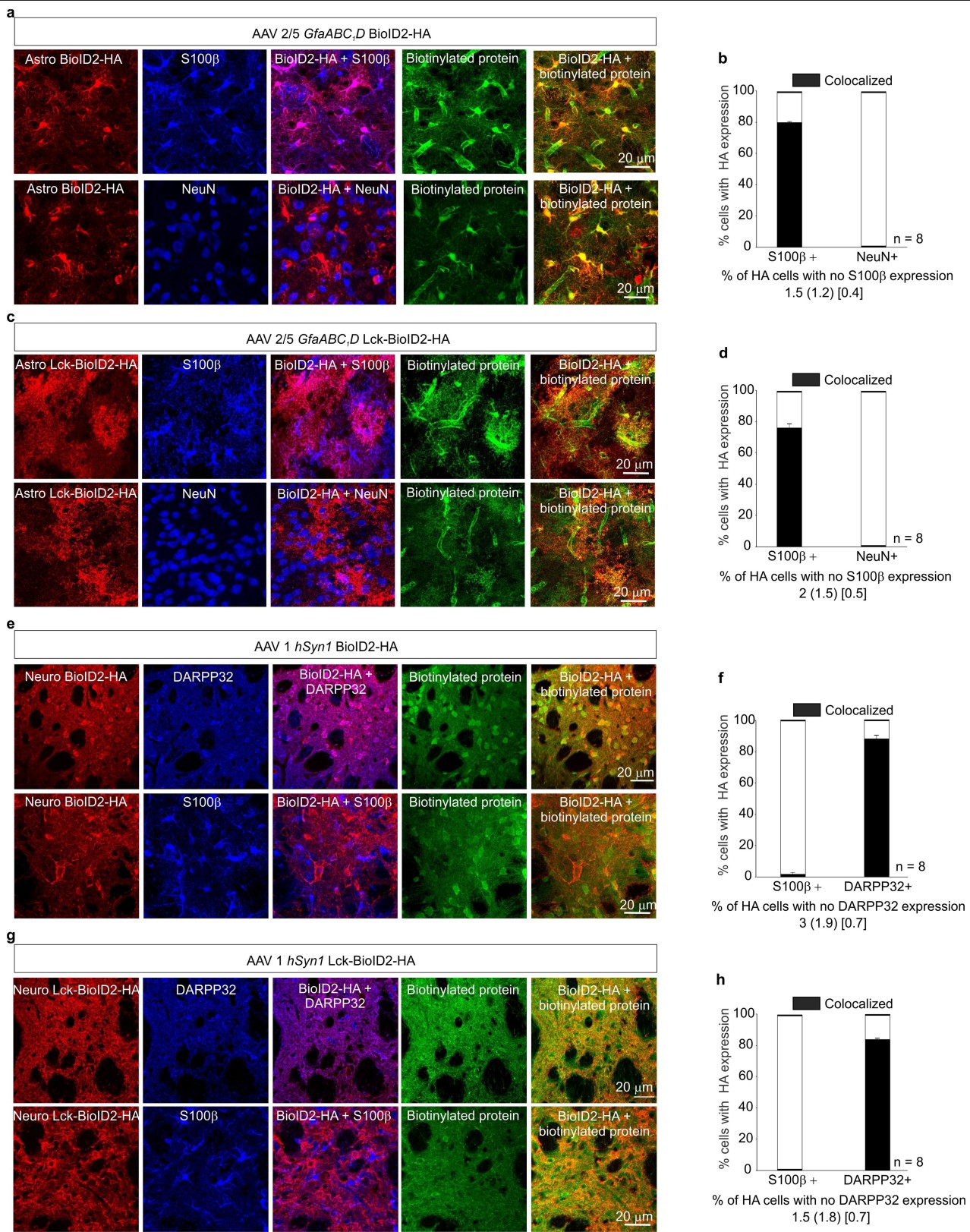

**Extended Data Fig. 4** | See next page for caption.

**Extended Data Fig. 4 | Validation of cell-specific BioID2 and Lck-BioID2.**
**a**. Representative images of immunostained mouse striatum injected with astrocyte-specific cytosolic BioID2 and then treated with biotin for 7 days. Top panel shows the immunostaining pattern with S100β as an astrocyte cell marker and bottom panel shows the immunostaining pattern with NeuN as a neuronal cell marker. **b**. Bar graphs depicting the percent of S100β positive or NeuN positive cells with HA expression in a 40x magnification field of view. Black portion of the bar graphs show the percent co-localization. Bottom descriptive statistics represent percent of HA+ cells that were not S100β positive as the mean (SD) [SEM] (n = 8 fields of view at 40x magnification from 4 mice). **c**. Representative images of immunostained mouse striatum injected with astrocyte-specific plasma membrane Lck-BioID2 and then treated with biotin for 7 days. Top panel shows the immunostaining pattern with S100β as an astrocyte cell marker and bottom panel shows the immunostaining pattern with NeuN as a neuronal cell marker. **d**. Bar graphs depicting the percent of S100β positive or NeuN positive cells with HA expression in a 40x magnification field of view. Black portion of the bar graphs show the percent co-localization. Bottom descriptive statistics represent percent of HA+ cells that were not S100β positive as the mean (SD) [SEM] (n = 8 fields of view at 40x magnification from 4 mice). **e**. Representative images of immunostained mouse striatum injected with neuron-specific cytosolic BioID2 and then treated with biotin for 7 days. Top panel shows the immunostaining pattern with DARPP32 as a neuronal cell marker and bottom panel shows the immunostaining pattern with S100β as an astrocyte cell marker. **f**. Bar graphs depicting the percent of S100β positive or DARPP32 positive cells with HA expression in a 40x magnification field of view. Black portion of the bar graphs show the percent co-localization. Bottom descriptive statistics represent percent of HA+ cells that were not DARPP32 positive as the mean (SD) [SEM] (n = 8 fields of view at 40x magnification from 4 mice). **g**. Representative images of immunostained mouse striatum injected with neuron-specific plasma membrane Lck-BioID2 and then treated with biotin for 7 days. Top panel shows the immunostaining pattern with DARPP32 as a neuronal cell marker and bottom panel shows the immunostaining pattern with S100β as an astrocytic cell marker. **h**. Bar graphs depicting the percent of S100β positive or DARPP32 positive cells with HA expression in a 40x magnification field of view. Black portion of the bar graphs show the percent co-localization. Bottom descriptive statistics represent percent of HA+ cells that were not DARPP32 positive as the mean (SD) [SEM] (n = 8 fields of view at 40x magnification from 4 mice).

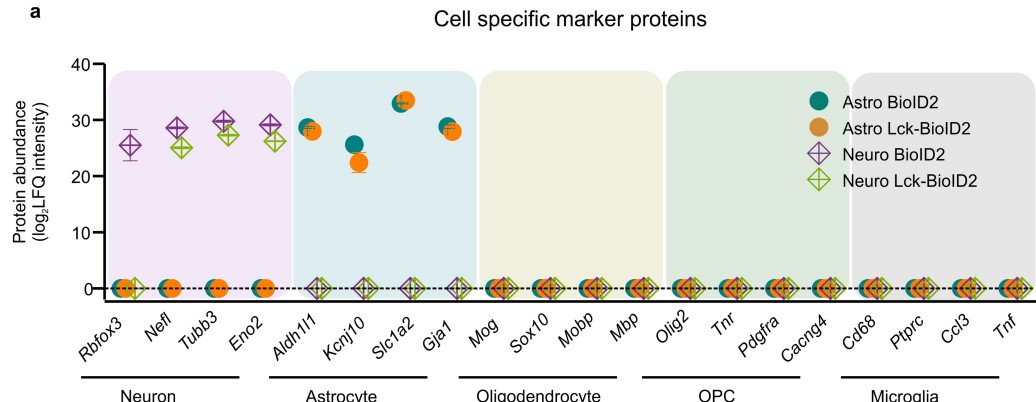

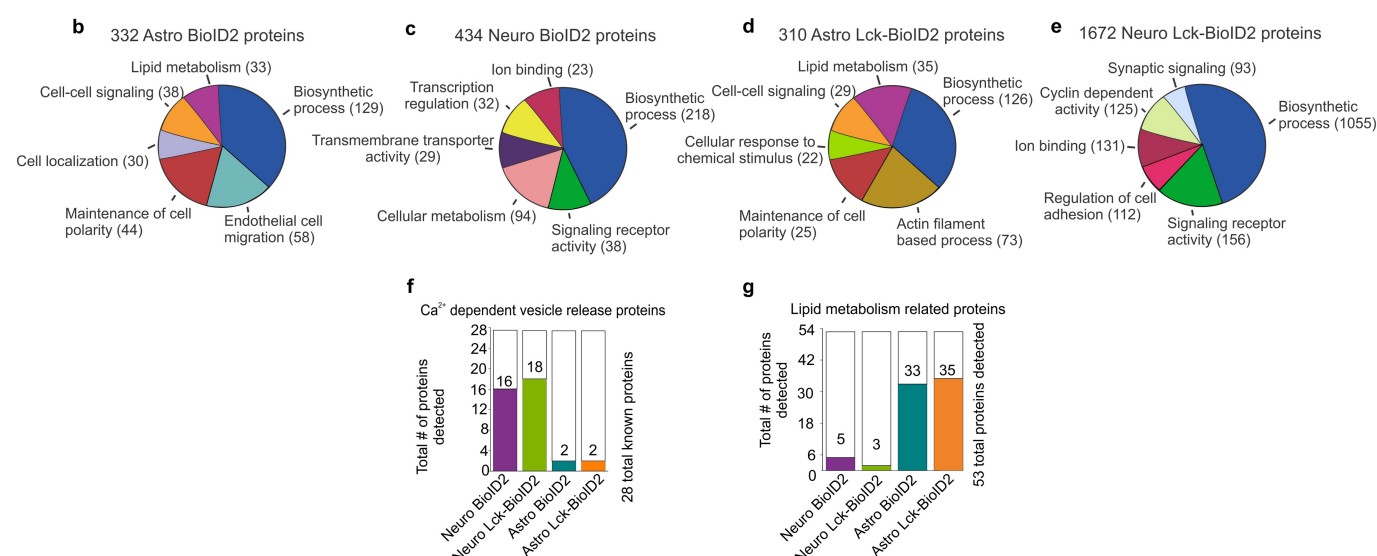

**Extended Data Fig. 5 | Broad assessments of BioID2 detected proteins.**
**a**. Expression levels (in label-free quantification intensity, LFQ intensity) of
cell-enriched markers identified in each BioID2 construct (n = 4 experimental
runs with 8 mice in each run). Proteins in this analysis represent proteins that
were enriched (Log2FC > 1 versus GFP controls). The mean LFQ value and SEM
are shown. **b**. Pie chart of PANTHER pathway analysis terms for "biological
processes". Pie chart shows the number of proteins found for each term from

the 332 Astro BioID2 proteins. **c**. As in **b**, but for the 434 Neuro BioID2 proteins.
**d**. As in **b**, but for 310 Astro Lck-BioID2 proteins. **e**. As in **b**, but for the 1672 Neuro
Lck-BioID2 proteins. **f**. Bar graph denotes the number of calcium dependent
vesicle release protein isoforms detected in each BioID2 construct experiment.
**g**. Bar graph denotes the number of lipid metabolism related proteins that were
detected in each BioID2 construct experiment.

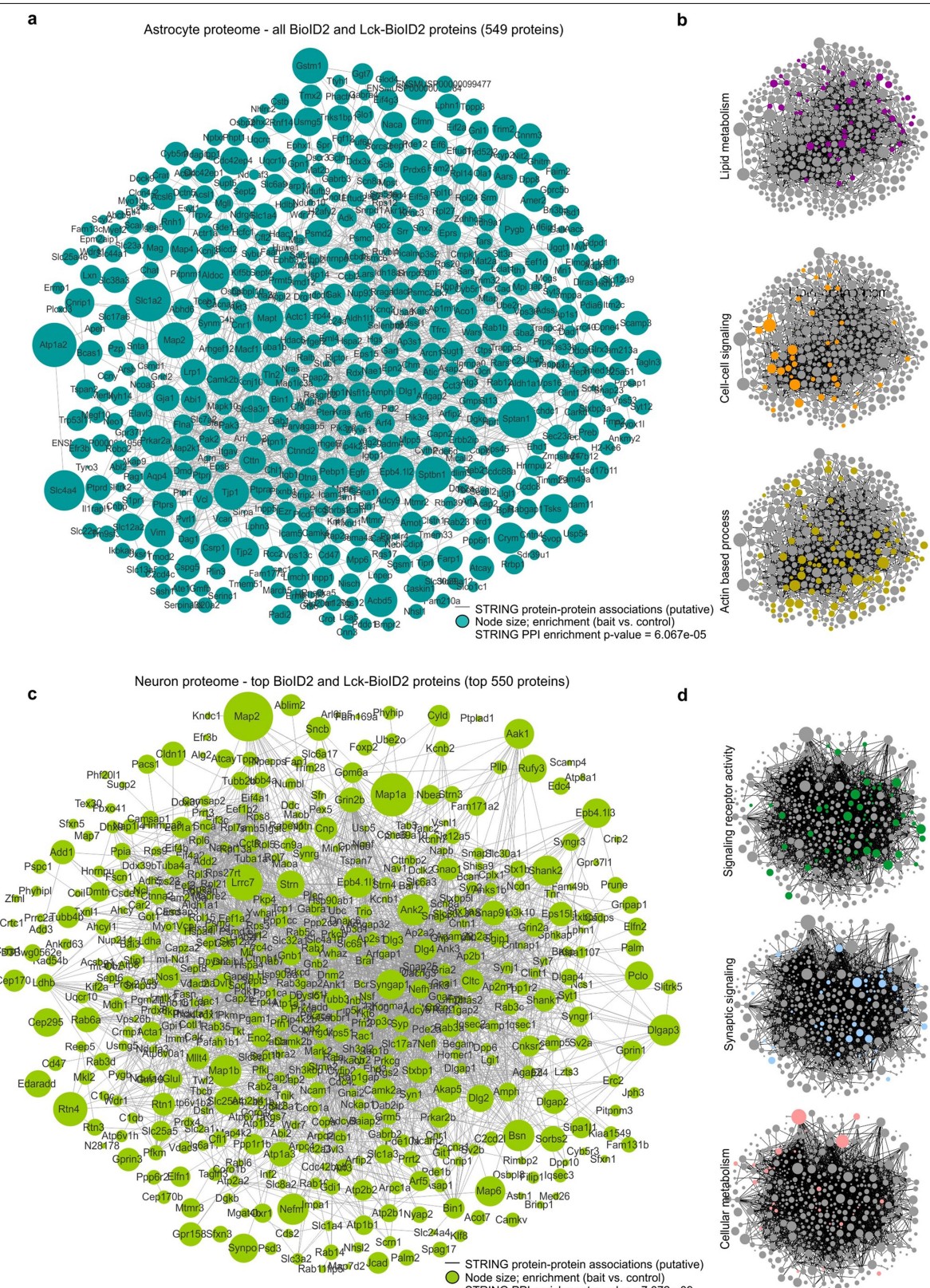

**Extended Data Fig. 6 | Interaction maps for the astrocyte and neuron proteomes. a**. Scale-free STRING analysis protein-protein association map of all the 549 unique and enriched biotinylated proteins identified in astrocytes with Astro BioID2 and Astro Lck-BioID2 . Node size represents the enrichment of each protein vs the GFP control (log₂(BioID2/GFP)). Edges represent putative interactions from the STRING database. **b**. Small clustergrams show selected functional categories from Panther GO analysis for "biological process" in

different colors. **c**. Scale-free STRING analysis interaction map of the top 550 unique and enriched biotinylated proteins identified in neurons with Neuro BioID2 and Neuro Lck-BioID2. Node size represents the enrichment of each protein vs the GFP control (log₂(BioID2/GFP)). Edges represent known interactions from the STRING database. **d**. Small clustergrams show selected functional categories from Panther GO analysis for "biological process" in different colors.

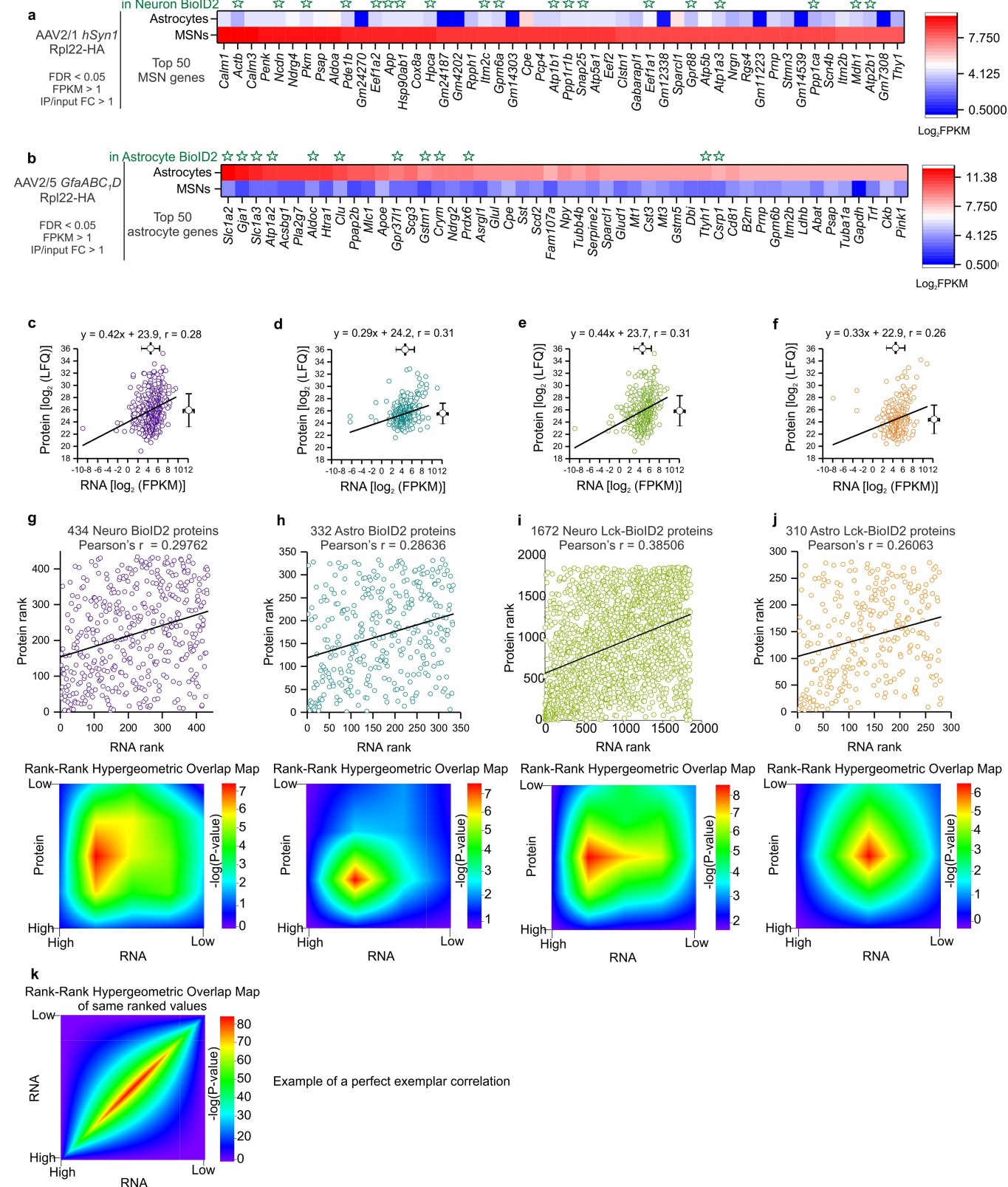

**Extended Data Fig. 7** | See next page for caption.

**Extended Data Fig. 7 | Detailed analyses comparing RNA-seq and proteomic data. a.** Top 50 medium spiny neuron (MSN) genes found by neuron-specific RiboTag AAV RNA-sequencing listed by the highest expression value (fragments per kilobase per million, FPKM). Astrocyte expression values for the top 50 MSN genes are shown. Green stars label gene products that were found at the proteomic level in Neuro BioID2 or Neuro Lck-BioID2. Scale shows $Log_2$FPKM. **b.** Top 50 astrocyte genes found by astrocyte-specific RiboTag AAV RNA-sequencing listed by the highest expression value (fragments per kilobase per million, FPKM). Neuronal expression values for the top 50 astrocyte genes are shown. Green stars label gene products that were found at the proteomic level in Astro BioID2 or Astro Lck-BioID2. Scale shows $Log_2$FPKM. **c.** Scatter graph showing protein expression ($Log_2$LFQ intensity) as a function of mRNA expression ($Log_2$FPKM) for the 434 Neuro BioID2 proteins identified. Pearson's **r** is shown. **d.** Scatter graph showing protein expression ($Log_2$LFQ intensity) as a function of mRNA expression ($Log_2$FPKM) for the 332 Astro BioID2 proteins identified. Pearson's **r** is shown. **e.** Scatter graph showing protein expression ($Log_2$LFQ intensity) as a function of mRNA expression ($Log_2$FPKM) for the 1672 Neuro Lck-BioID2 proteins identified. Pearson's **r** is shown. **f.** Scatter graph showing protein expression ($Log_2$LFQ intensity) as a function of mRNA expression ($Log_2$FPKM) for the 310 Astro Lck-BioID2 proteins identified. Pearson's **r** is shown. **g.** Scatter graph showing the protein rank (by LFQ abundance) as a function of mRNA rank (by FPKM abundance) of the 434 Neuro BioID2 proteins identified. Two-tailed Pearson's **r** is shown. Heat map shows the rank-rank hypergeometric overlap (RRHO) of the RNA and protein rank. Each pixel represents the significance of overlap between the two datasets in −$log_{10}$(P-value). Red pixels represent highly significant overlap. Color scale denotes the range of P-values at the negative $log_{10}$ scale (Bin size = 100). **h.** Scatter graph showing the protein rank (by LFQ abundance) as a function of mRNA rank (by FPKM abundance) of the 332 Astro BioID2 proteins identified. Two-tailed Pearson's **r** is shown. Heat map shows the rank-rank hypergeometric overlap (RRHO) of the RNA and protein rank. (Bin size = 100). **i.** Scatter graph showing the protein rank (by LFQ abundance) as a function of mRNA rank (by FPKM abundance) of the 1672 Neuro Lck-BioID2 proteins identified. Two-tailed Pearson's **r** is shown. Heat map shows the rank-rank hypergeometric overlap (RRHO) of the RNA and protein rank (Bin size = 100). **j.** Scatter graph showing the protein rank (by LFQ abundance) as a function of mRNA rank (by FPKM abundance) of the 310 Astro Lck-BioID2 proteins identified. Two-tailed Pearson's **r** is shown. Heat map shows the rank-rank hypergeometric overlap (RRHO) of the RNA and protein rank. (Bin size = 100). **k.** Heat map shows and simulated rank-rank hypergeometric overlap (RRHO) of the mRNA of the 434 Neuro BioID2 proteins versus the very same idealized mRNA of the 434 proteins. (Bin size = 100).

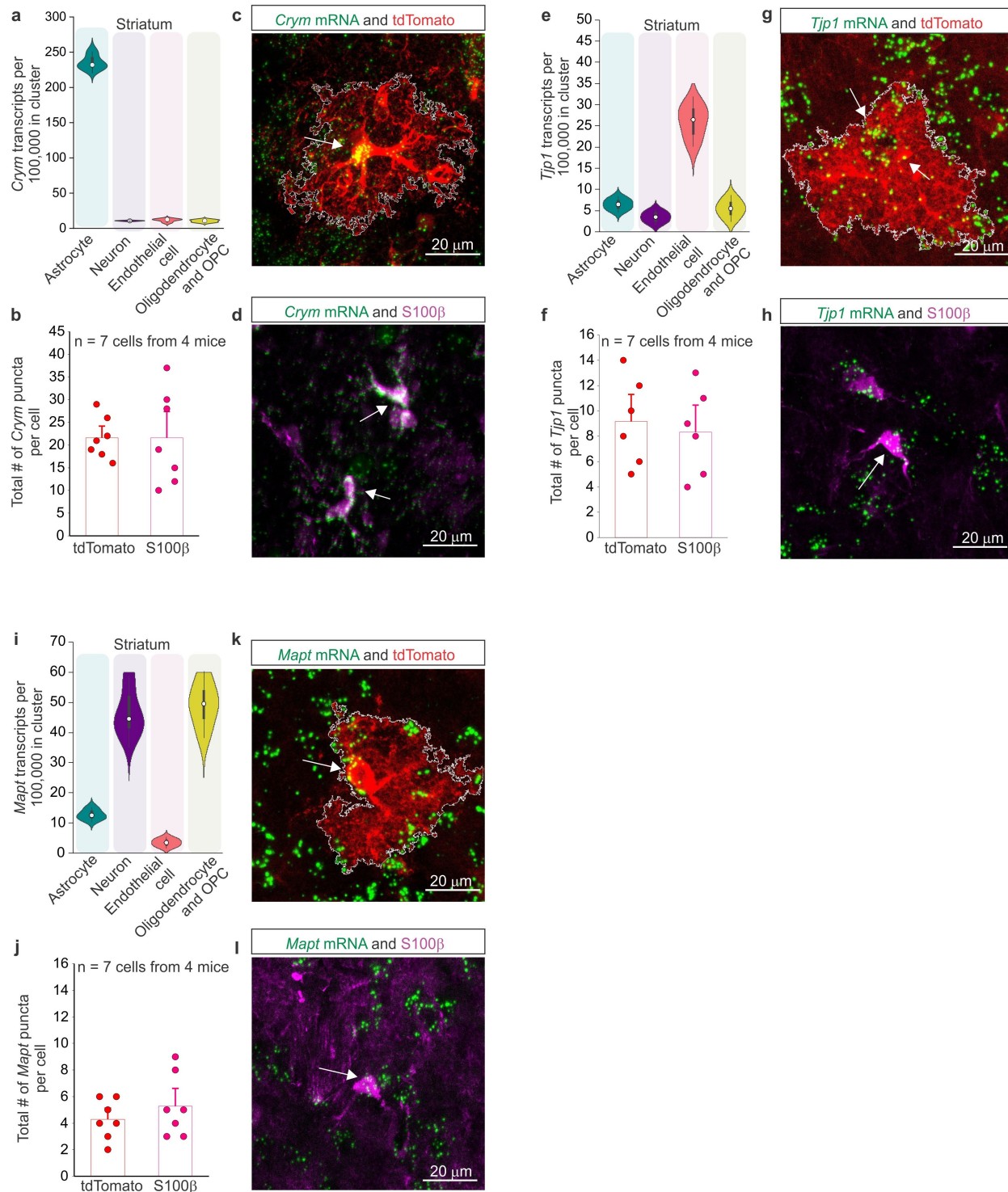

**Extended Data Fig. 8** | See next page for caption.

**Extended Data Fig. 8 | mRNA expression validation of astrocyte identified proteins. a**. Violin plot from DropViz.org showing the expression of *Crym* in astrocytes, neurons, endothelial cells, and myelin associated cells of the striatum. Center of the box plot for all violin plots represent the median, the box limits indicate the first and third quartiles, and the whiskers/range indicate the distribution of the data range. **b**. Bar graphs denoting the total number of *Crym* mRNA puncta per tdTomato+ astrocyte and S100β+ cell. Mean and SEM are shown (n = 7 cells from 4 mice per group). **c**. Representative image of an *Aldh1l1* tdTomato+ striatal astrocyte with *Crym* mRNA puncta expression (green). Arrow denotes puncta within the cell. **d**. Representative image of S100β+ cells with *Crym* mRNA puncta expression (green). Arrows denote the puncta within the cells. **e**. Violin plot from DropViz.org showing the expression of *Tjp1* in astrocytes, neurons, endothelial cells, and myelin associated cells of the striatum. **f**. Bar graphs denoting the total number of *Tjp1* mRNA puncta per tdTomato+ astrocyte and S100β+ cell. Mean and SEM are shown (n = 7 cells from 4 mice per group) **g**. Representative image of an *Aldh1l1* tdTomato+ striatal astrocyte with *Tjp1* mRNA puncta expression (green). Arrows denote puncta within the cell. **h**. Representative image of S100β+ cells with *Tjp1* mRNA puncta expression (green). Arrows denote the puncta within the cells. **i**. Violin plot from DropViz.org showing the expression of *Mapt* in astrocytes, neurons, endothelial cells, and myelin associated cells of the striatum. **j**. Bar graphs denoting the total number of *Mapt* mRNA puncta per tdTomato+ astrocyte and S100β+ cell. Mean and SEM are shown (n = 7 cells from 4 mice per group). **k**. Representative image of an *Aldh1l1* tdTomato+ striatal astrocyte with *Mapt* mRNA puncta expression (green). Arrow denotes puncta within the cell. **l**. Representative image of an S100β+ cell with *Mapt* mRNA puncta expression (green). Arrow denotes the puncta within the cell.

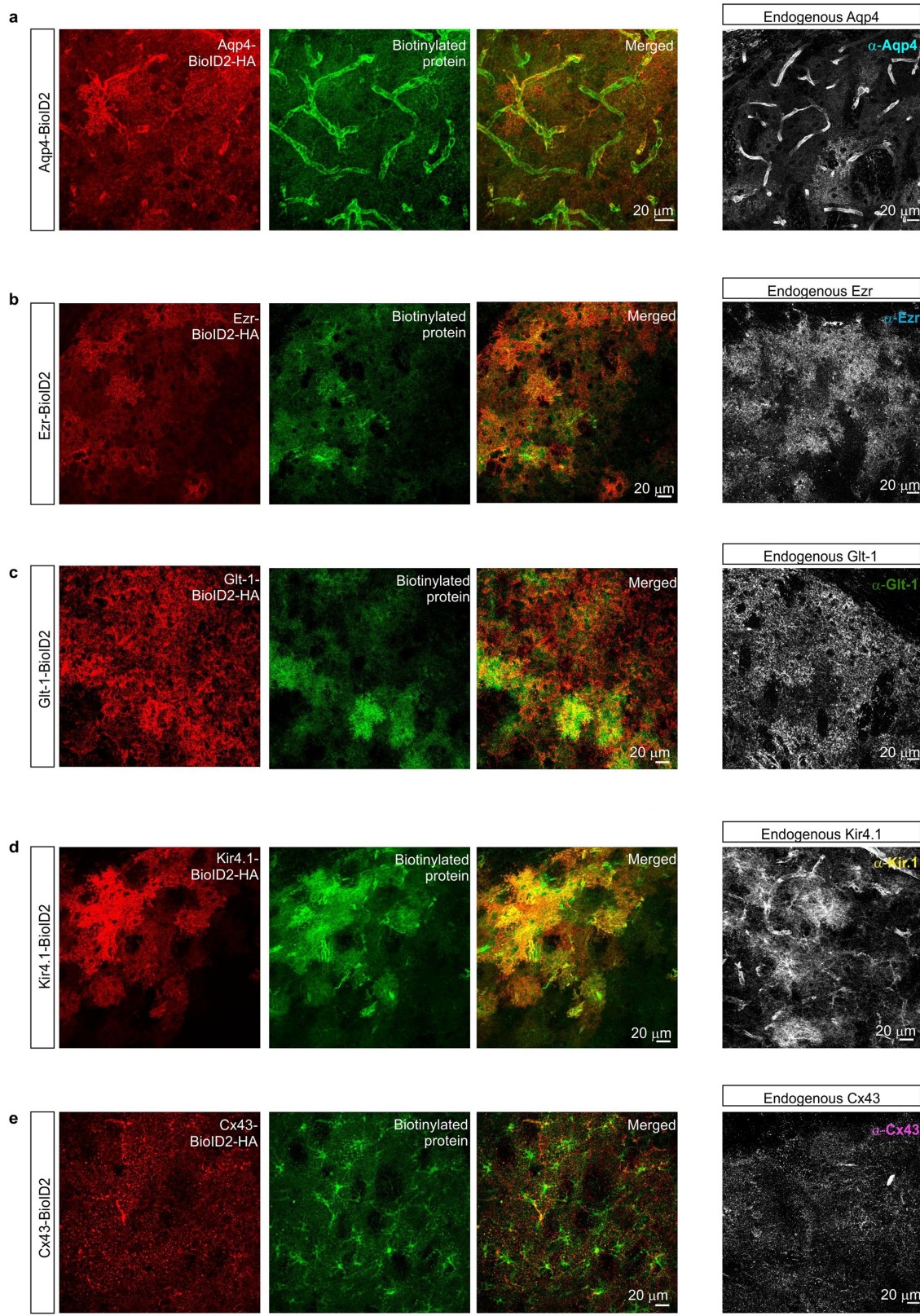

**Extended Data Fig. 9 | Immunohistochemistry of subcompartment specific BioID2. a**. Representative images of immunostained mouse striatum injected with astrocyte specific Aquaporin4-BioID2 (Aqp4-BioID2) and then treated with biotin for 7 days. The tissue was immunostained with both anti-HA antibody (red) and a fluorophore conjugated streptavidin probe (green).

Farthest panel on the right shows the endogenous Aqp4 expression pattern in the striatum after immunostaining with anti-Aqp4 antibody. **b-e**. As in a, but for Ezrin-BioID2 (**b**), Glt1-BioID2 (**c**), Kir4.1-BioID2 (**d**), and Cx43-BioId2 (**e**). The experiments depicted here were each replicated four times.

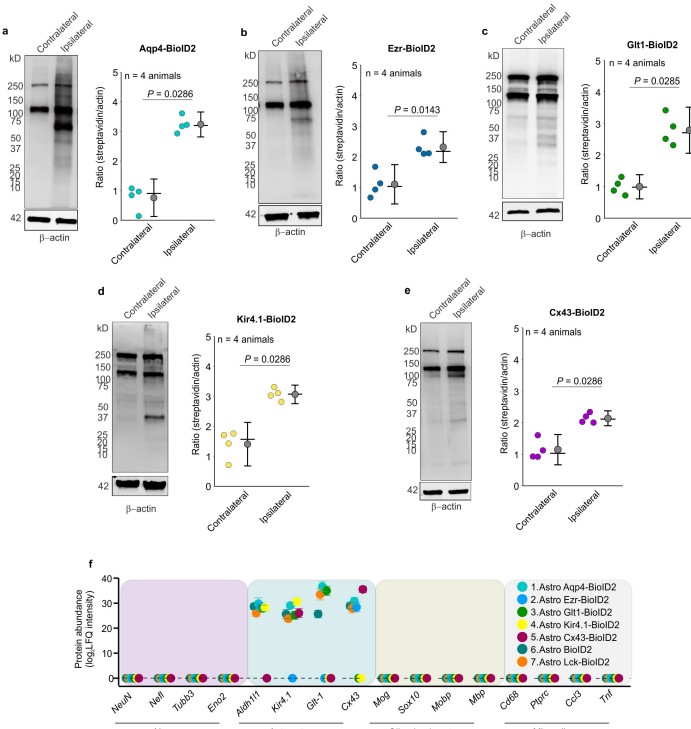

**Extended Data Fig. 10 | Biotinylation assessments of subcompartment BioID2 constructs. a**. Western blot analysis of brain unilaterally microinjected with astrocyte specific Aqp4-BioID2. Dark bands at 130 kD and 250 kD show the endogenously biotinylated mitochondrial proteins, Pyruvate carboxylase and acetyl-CoA carboxylase. Graph depicting the streptavidin signal intensity divided by the β-actin signal intensity for each data point. Black horizontal line depicts the mean, circle represents the median, and error bars show the SEM (n = 4 mice; Mann-Whitney test). **b-e**. As in a, but for Ezrin-BioID2 (**b**),

Glt1-BioID2 (**c**), Kir4.1-BioID2 (**d**), and Cx43-BioId2 (**e**) (n = 4 mice in each case; Two-tailed Mann-Whitney test). **f**. Expression levels (in label-free quantification intensity, LFQ intensity) of cell-enriched markers identified in each astrocytic sub-compartment BioID2 construct (n = 4 experimental runs). Proteins in this analysis represent proteins that were enriched (Log2FC > 1 versus GFP controls). The mean LFQ value and SEM are shown. For blot source data, see Supplementary Fig. 1.

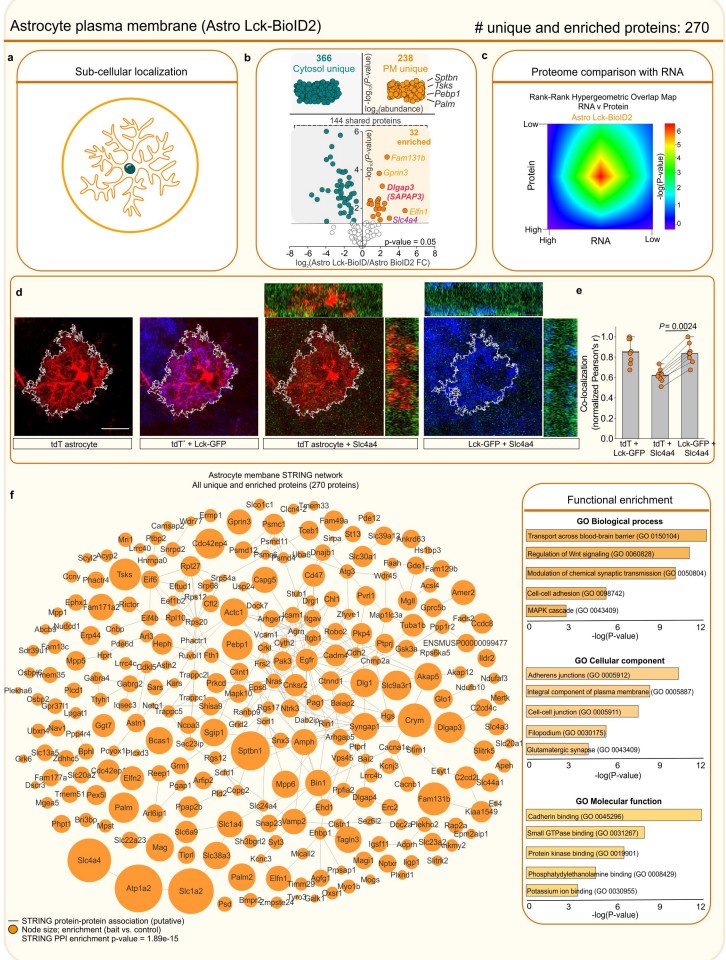

**Extended Data Fig. 11 | Astrocyte subcompartment card 1: plasma membrane. a.** BioID2 that is targeted to the plasma membrane biotinylates proteins near the plasma membrane. **b.** Label-free based quantification comparison of significant proteins ($Log_2FC > 1$ and FDR < 0.05 *versus* GFP controls) detected in the cytosolic Astro BioID2 and plasma membrane Astro Lck-BioID2 reveal plasma membrane enriched proteins. Top half of the volcano plot shows 238 unique Lck-BioID2 proteins when compared to cytosol. The top four most abundant proteins for Lck-BioID2 are shown. Lower half of volcano plot shows comparison of 144 proteins that were common in both cytosolic BioID2 and Lck-BioID2. The five highest enriched proteins for Lck-BioID2 ($Log_2FC > 2$) are shown. Magenta label shows protein that was validated with IHC in panel **d**. Red label shows Dlgap3/SAPAP3 is enriched in the astrocyte plasma membrane. **c.** Heat map shows the rank-rank hypergeometric overlap (RRHO) of the RNA and protein rank for the 270 plasma membrane proteins. Each pixel represents the significance of overlap between the two datasets

in $-log_{10}$(P-value). Red pixels represent highly significant overlap. Color scale denotes the range of P-values at the negative $log_{10}$ scale (bin size = 100). **d.** IHC analysis of Slc4a4 (Nbc1) protein in tdTomato and Lck-GFP labeled astrocytes shows co-localization within the astrocyte territory. Scale bar represents 20 μm. **e.** Co-localization analysis using Pearson's r co-efficient shows high co-localization between Lck-GFP and Slc4a4 (Nbc1). The mean and SEM are shown (n = 8 tdTomato+ cells from 4 mice; Two-tailed paired t-test). **f.** Scale-free STRING analysis protein-protein association map of the 270 unique and enriched biotinylated proteins identified in astrocyte plasma membrane with Astro Lck-BioID2. Node size represents the enrichment of each protein vs the GFP control (log2(BioID2/GFP)). Edges represent putative interactions from the STRING database. Bar graphs show the functional enrichment analysis of the 270 proteins using "Biological process", "Cellular component", and "Molecular function" terms from Enrichr.

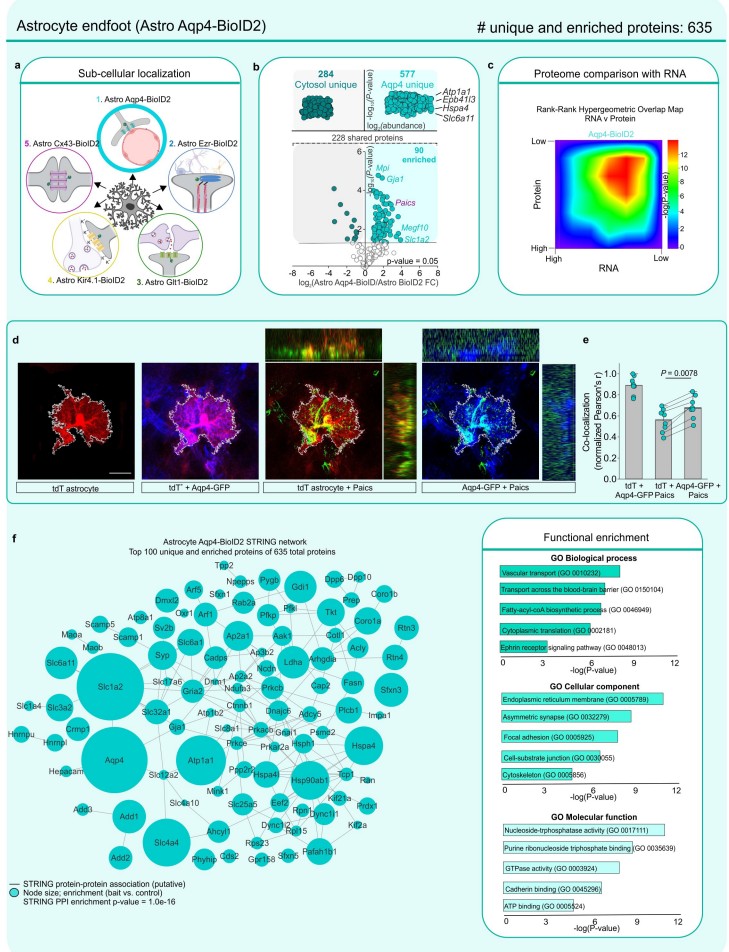

**Extended Data Fig. 12 | Astrocyte subcompartment card 2: end foot.**
**a**. BioID2 that is targeted with Aqp4-BioID2 biotinylates proteins at the astrocyte end foot. **b**. Label-free based quantification comparison of significant proteins (Log$_2$FC >1 and FDR < 0.05 *versus* GFP controls) detected in the cytosolic Astro BioID2 and end foot Astro Aqp4-BioID2 reveal end foot enriched proteins. Top half of the volcano plot shows 577 unique Aqp4-BioID2 proteins when compared to cytosol. The top four most abundant proteins for Aqp4-BioID2 are shown. Lower half of volcano plot shows comparison of 228 proteins that were common in both cytosolic BioID2 and Aqp4-BioID2. The five highest enriched proteins for Aqp4-BioID2 (Log$_2$FC > 2) are shown. Magenta label shows protein that was validated with immunohistochemistry in panel **d**. **c**. Heat map shows the rank-rank hypergeometric overlap (RRHO) of the RNA and protein rank for the 635 endfoot proteins. Each pixel represents the significance of overlap between the two datasets in −log$_{10}$(P-value). Red pixels represent highly significant overlap. Color scale denotes the range of P-values at the negative log$_{10}$ scale (bin size = 100). **d**. IHC analysis of PAICS protein in tdTomato and Lck-GFP labeled astrocytes shows co-localization within the astrocyte territory. Scale bar represents 20 µm. **e**. Co-localization analysis using Pearson's r co-efficient shows high co-localization between Aqp4-GFP and PAICS. The mean and SEM are shown (n = 8 tdTomato+ cells from 4 mice; Two-tailed paired t-test). **f**. Scale-free STRING analysis protein-protein association map of the top 100 unique and enriched biotinylated proteins identified in the astrocyte endfoot with Astro Aqp4-BioID2. Node size represents the enrichment of each protein vs the GFP control (log2(BioID2/GFP)). Edges represent putative interactions from the STRING database. Bar graphs show the functional enrichment analysis of the 635 proteins using "Biological process", "Cellular component", and "Molecular function" terms from Enrichr. The image of the astrocyte subcompartments in panel a was created using BioRender (https://www.biorender.com/).

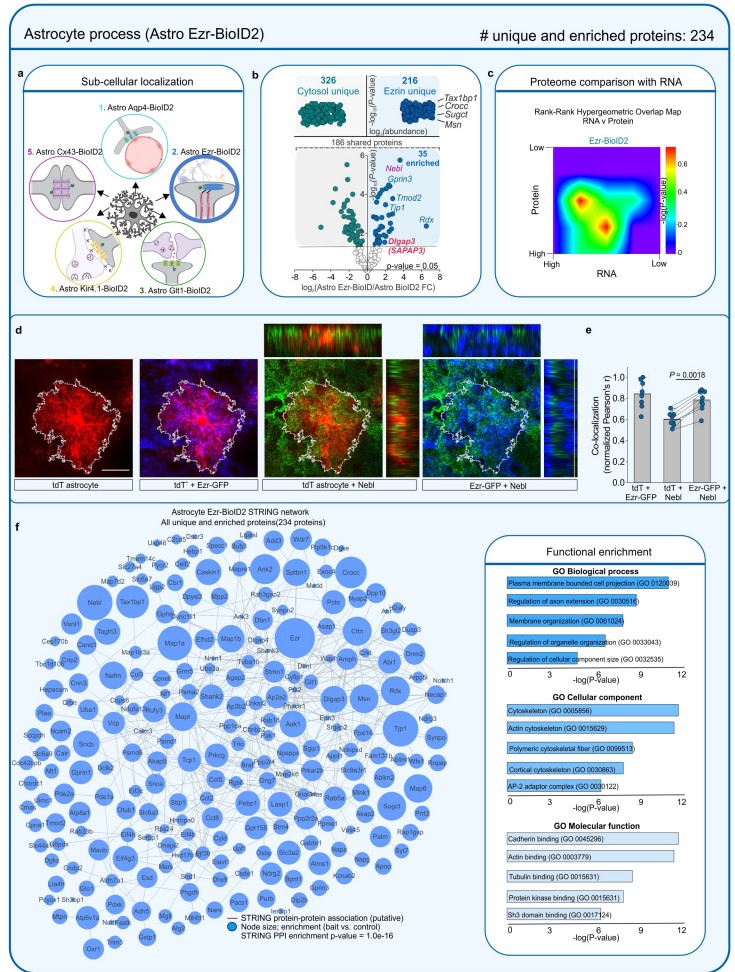

**Extended Data Fig. 13 | Astrocyte subcompartment card 3: astrocyte processes. a**. BioID2 that is targeted with Ezr-BioID2 biotinylates proteins at astrocyte processes. **b**. Label-free based quantification comparison of significant proteins (Log$_2$FC > 1 and FDR < 0.05 *versus* GFP controls) detected in the cytosolic Astro BioID2 and fine process Astro Ezrin-BioID2 reveal fine process enriched proteins. Top half of the volcano plot shows 216 unique Ezrin-BioID2 proteins when compared to cytosol. The top four most abundant proteins for Ezrin-BioID2 are shown. Lower half of volcano plot shows comparison of 186 proteins that were common in both cytosolic BioID2 and Ezrin-BioID2. The five highest enriched proteins for Ezrin-BioID2 (Log$_2$FC > 2) are shown. Magenta label shows protein that was validated with immunohistochemistry in panel **d**. Red label shows Dlgap3/SAPAP3 is enriched in the astrocyte fine processes. **c**. Heat map shows the rank-rank hypergeometric overlap (RRHO) of the RNA and protein rank for the 234 Ezr-BioID2 proteins. Each pixel represents the significance of overlap between the two datasets

in −log$_{10}$(P-value). Red pixels represent highly significant overlap. Color scale denotes the range of P-values at the negative log$_{10}$ scale (bin size = 100). **d**. IHC analysis of Nebl protein in tdTomato and Ezr-GFP labeled astrocytes shows co-localization within the astrocyte territory. Scale bar represents 20 µm. **e**. Co-localization analysis using Pearson's r co-efficient shows high co-localization between Ezr-GFP and Nebl. The mean and SEM are shown (n = 8 tdTomato+ cells from 4 mice; Two-tailed paired t-test). **f**. Scale-free STRING analysis protein-protein association map of the 234 unique and enriched biotinylated proteins identified in astrocyte processes with Astro Ezr-BioID2. Node size represents the enrichment of each protein vs the GFP control (log2(BioID2/GFP)). Edges represent putative interactions from the STRING database. Bar graphs show the functional enrichment analysis of the 234 proteins using "Biological process", "Cellular component", and "Molecular function" terms from Enrichr. The image of the astrocyte subcompartments in panel a was created using BioRender (https://www.biorender.com/).

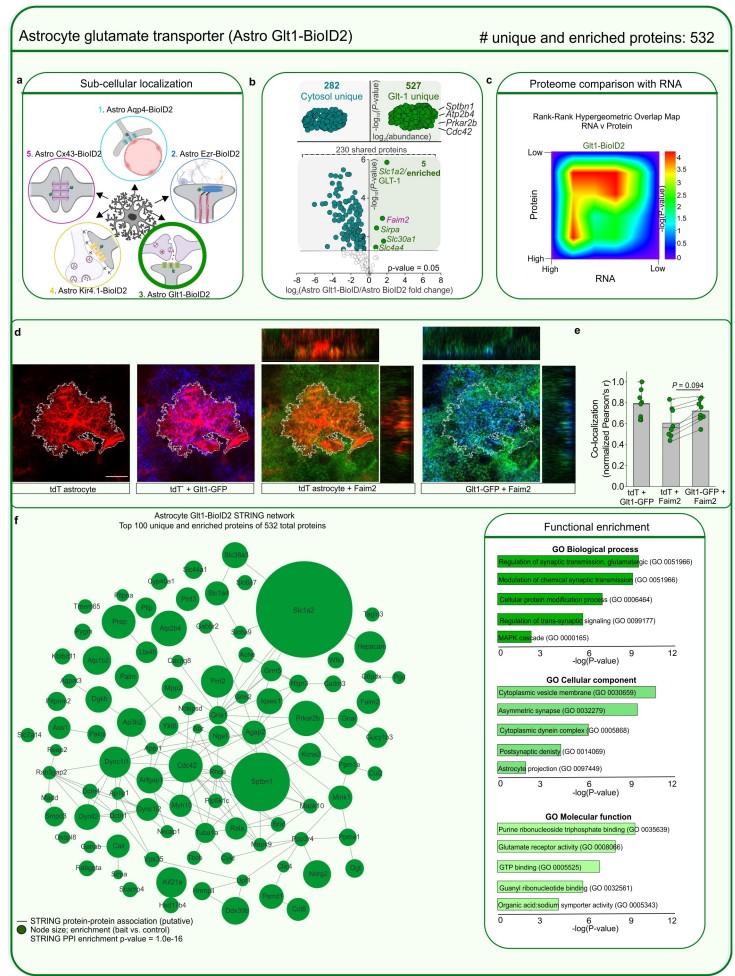

**Extended Data Fig. 14 | Astrocyte subcompartment card 4: astrocyte glutamate transporter. a**. BioID2 that is targeted with Glt1-BioID2 biotinylates proteins at astrocyte sites of glutamate uptake. **b**. Label-free based quantification comparison of significant proteins (Log$_2$FC > 1 and FDR < 0.05 *versus* GFP controls) detected in the cytosolic Astro BioID2 and Astro Glt1-BioID2 reveal Glt1 enriched proteins. Top half of the volcano plot shows 527 unique Glt1-BioID2 proteins when compared to cytosol. The top four most abundant proteins for Glt1-BioID2 are shown. Lower half of volcano plot shows comparison of 230 proteins that were common in both cytosolic BioID2 and Glt1-BioID2. The five highest enriched proteins for Glt1-BioID2 are shown. Magenta label shows protein that was validated with immunohistochemistry. **c**. Heat map shows the rank-rank hypergeometric overlap (RRHO) of the RNA and protein rank for the 532 Glt1-BioID2 proteins. Each pixel represents the significance of overlap between the two datasets in –log$_{10}$(P-value). Red pixels represent highly significant overlap. Color scale denotes the range of P-values

at the negative log$_{10}$ scale (bin size = 100). **d**. IHC analysis of Faim2 protein in tdTomato and Glt1-GFP labeled astrocytes shows co-localization within the astrocyte territory. Scale bar represents 20 μm. **e**. Co-localization analysis using Pearson's r co-efficient shows high co-localization between Glt1-GFP and Faim2. The mean and SEM are shown (n = 8 tdTomato+ cells from 4 mice; Two-tailed paired t-test). **f**. Scale-free STRING analysis protein-protein association map of the top 100 unique and enriched biotinylated proteins identified with Astro Glt1-BioID2. Node size represents the enrichment of each protein vs the GFP control (log2(BioID2/GFP)). Edges represent putative interactions from the STRING database. Bar graphs show the functional enrichment analysis of all 532 proteins using "Biological process", "Cellular component", and "Molecular function" terms from Enrichr. The image of the astrocyte subcompartments in panel a was created using BioRender (https://www.biorender.com/).

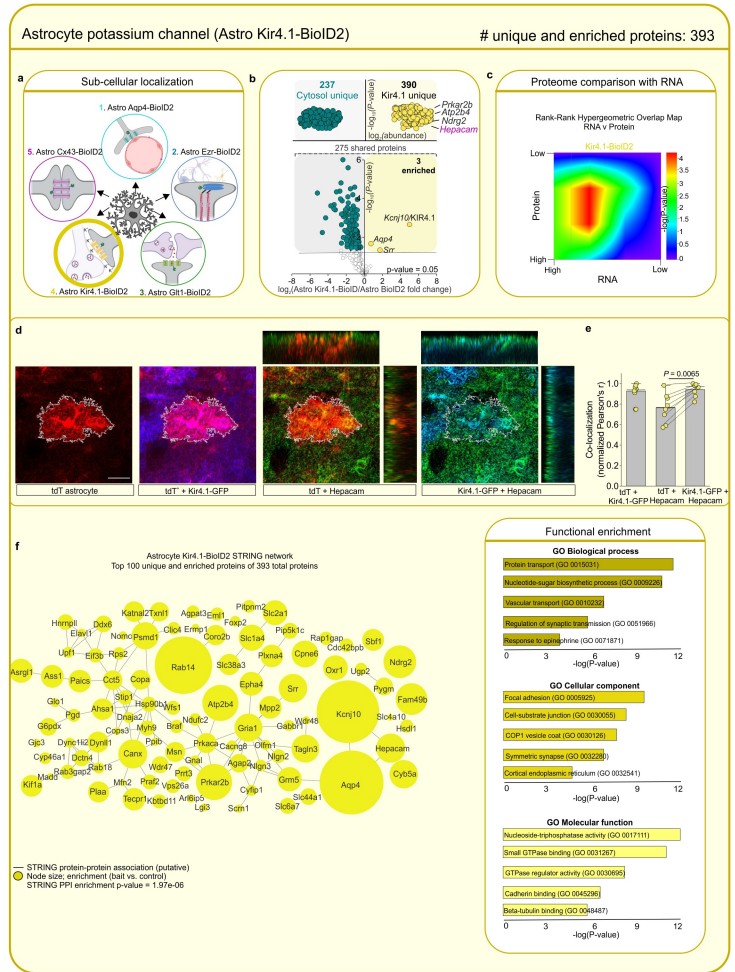

**Extended Data Fig. 15 | Astrocyte subcompartment card 5: astrocyte potassium channel. a**. BioID2 that is targeted with Kir4.1-BioID2 biotinylates proteins at astrocyte sites of potassium uptake. **b**. Label-free based quantification comparison of significant proteins (Log$_2$FC > 1 and FDR < 0.05 *versus* GFP controls) detected in the cytosolic Astro BioID2 and Astro Kir4.1-BioID2 reveal Kir4.1 enriched proteins. Top half of the volcano plot shows 390 unique Kir4.1-BioID2 proteins when compared to cytosol. The top four most abundant proteins for Kir4.1-BioID2 are shown. Lower half of volcano plot shows comparison of 275 proteins that were common in both cytosolic BioID2 and Kir4.1-BioID2. The five highest enriched proteins for Kir4.1-BioID2 are shown. Magenta label shows protein that was validated with immunohistochemistry. **c**. Heat map shows the rank-rank hypergeometric overlap (RRHO) of the RNA and protein rank for the 393 Kir4.1-BioID2 proteins. Each pixel represents the significance of overlap between the two datasets in –log$_{10}$(P-value). Red pixels represent highly significant overlap. Color scale

denotes the range of P-values at the negative log$_{10}$ scale (Bin size = 100). **d**. IHC analysis of Hepacam protein in tdTomato and Kir4.1-GFP labeled astrocytes shows co-localization within the astrocyte territory. Scale bar represents 20 μm. **e**. Co-localization analysis using Pearson's r co-efficient shows high co-localization between Kir4.1-GFP and Hepacam. The mean and SEM are shown (n = 8 tdTomato+ cells from 4 mice; Two-tailed paired t-test). **f**. Scale-free STRING analysis protein-protein association map of the top 100 unique and enriched biotinylated proteins identified with Astro Kir4.1-BioID2 . Node size represents the enrichment of each protein vs the GFP control (log2(BioID2/GFP)). Edges represent putative interactions from the STRING database. Bar graphs show the functional enrichment analysis of all 393 proteins using "Biological process", "Cellular component", and "Molecular function" terms from Enrichr. The image of the astrocyte subcompartments in panel a was created using BioRender (https://www.biorender.com/).

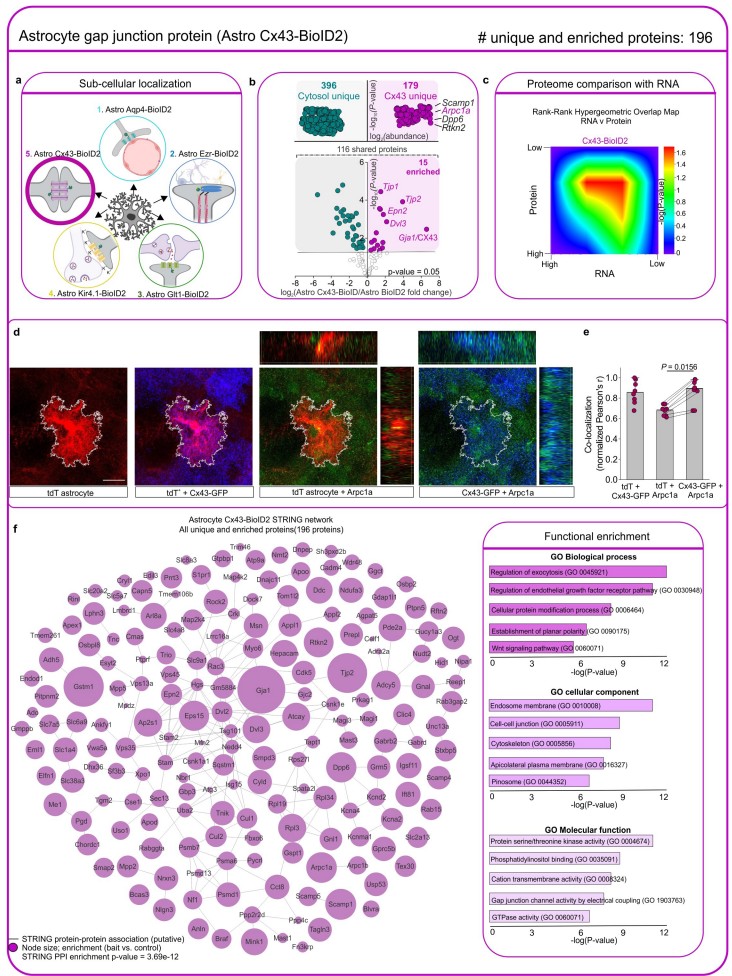

**Extended Data Fig. 16 | Astrocyte subcompartment card 6: astrocyte gap junction protein. a**. BioID2 that is targeted with Cx43-BioID2 biotinylates proteins at astrocyte sites of gap junction coupling. **b**. Label-free based quantification comparison of significant proteins (Log$_2$FC >1 and FDR < 0.05 *versus* GFP controls) detected in the cytosolic Astro BioID2 and Astro Cx43-BioID2 reveal Cx43 enriched proteins. Top half of the volcano plot shows 179 unique Cx43-BioID2 proteins when compared to cytosol. The top four most abundant proteins for Cx43-BioID2 are shown. Lower half of volcano plot shows comparison of 116 proteins that were common in both cytosolic BioID2 and Cx43-BioID2. The five highest enriched proteins for Cx43-BioID2 are shown. Magenta label shows protein that was validated with immunohistochemistry. **c**. Heat map shows the rank-rank hypergeometric overlap (RRHO) of the RNA and protein rank for the 196 Cx43-BioID2 proteins. Each pixel represents the significance of overlap between the two datasets in $-\log_{10}$(P-value). Red pixels represent highly significant overlap. Color scale denotes the range of P-values

at the negative $\log_{10}$ scale (bin size = 100). **d**. IHC analysis of Arpc1a protein in tdTomato and Cx43-GFP labeled astrocytes shows co-localization within the astrocyte territory. Scale bar represents 20 μm. **e**. Co-localization analysis using Pearson's r co-efficient shows high co-localization between Cx43-GFP and Aprc1a. The mean and SEM are shown (n = 8 tdTomato+ cells from 4 mice; Two-tailed Wilcoxon matched-pairs signed rank test). **f**. Scale-free STRING analysis protein-protein association map of the 196 unique and enriched biotinylated proteins identified with Astro Cx43-BioID2. Node size represents the enrichment of each protein vs the GFP control (log2(BioID2/GFP)). Edges represent putative interactions from the STRING database. Bar graphs show the functional enrichment analysis of all 196 proteins using "Biological process", "Cellular component", and "Molecular function" terms from Enrichr. The image of the astrocyte subcompartments in panel a was created using BioRender (https://www.biorender.com/).

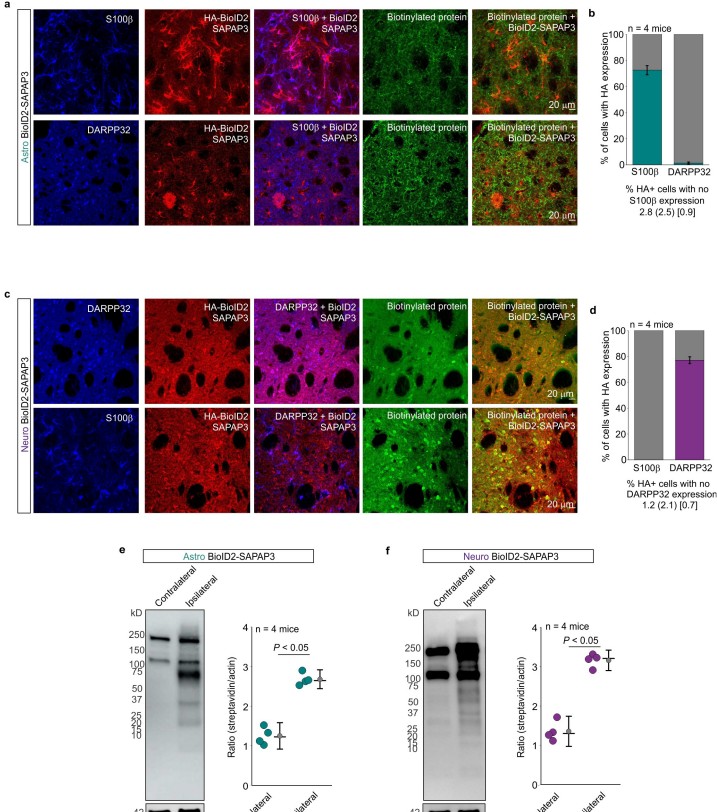

**Extended Data Fig. 17 | IHC and Western blots for SAPAP3 BioID2 constructs. a**. Representative images of immunostained mouse striatum injected with astrocyte-specific BioID2-SAPAP3 and then treated with biotin for 7 days. Top panel shows the immunostaining pattern with S100β as an astrocyte cell marker and bottom panel shows the immunostaining pattern with DARPP32 as a neuronal cell marker. **b**. Bar graphs depicting the percent of S100β positive or DARPP32 positive cells with HA expression in a 40x magnification field of view. Teal portion of the bar graphs show the percent co-localization. Bottom descriptive statistics represent percent of HA+ cells that were not S100β positive as the mean (SD) [SEM] (n = 8 fields of view at 40x magnification from 4 mice). **c**. Representative images of immunostained mouse striatum injected with neuron-specific BioID2-SAPAP3 and then treated with biotin for 7 days. Top panel shows the immunostaining pattern with DARPP32 as a neuronal cell marker and bottom panel shows the immunostaining pattern with S100β as an astrocyte cell marker. **d**. Bar graphs depicting the percent of S100β positive or DARPP32 positive cells with HA expression in a 40x magnification field of view. Purple portion of the bar graphs show the percent co-localization. Bottom descriptive statistics represent percent of HA+ cells that were not DARPP32 positive as the mean (SD) [SEM] (n = 8 fields of view at 40x magnification from 4 mice) **e**. Western blot analysis of brain unilaterally microinjected with astrocyte specific BioID2-SAPAP3. Dark bands at 130 kD and 250 kD show the endogenously biotinylated mitochondrial proteins, pyruvate carboxylase and acetyl-CoA carboxylase. Graph depicting the streptavidin signal intensity divided by the β-actin signal intensity for each data point. Black horizontal line depicts the mean, circle represents the median, and error bars show the SEM (n = 4 mice; Two-tailed Mann-Whitney Test, P = 0.029). **f**. Western blot analysis of brain unilaterally microinjected with neuron specific BioID2-SAPAP3. Graph depicting the streptavidin signal intensity divided by the β-actin signal intensity for each data point. Black horizontal line depicts the mean, circle represents the median, and error bars show the SEM (n = 4 mice; Two-tailed Mann-Whitney Test, P = 0.014). For blot source data, see Supplementary Fig. 1.

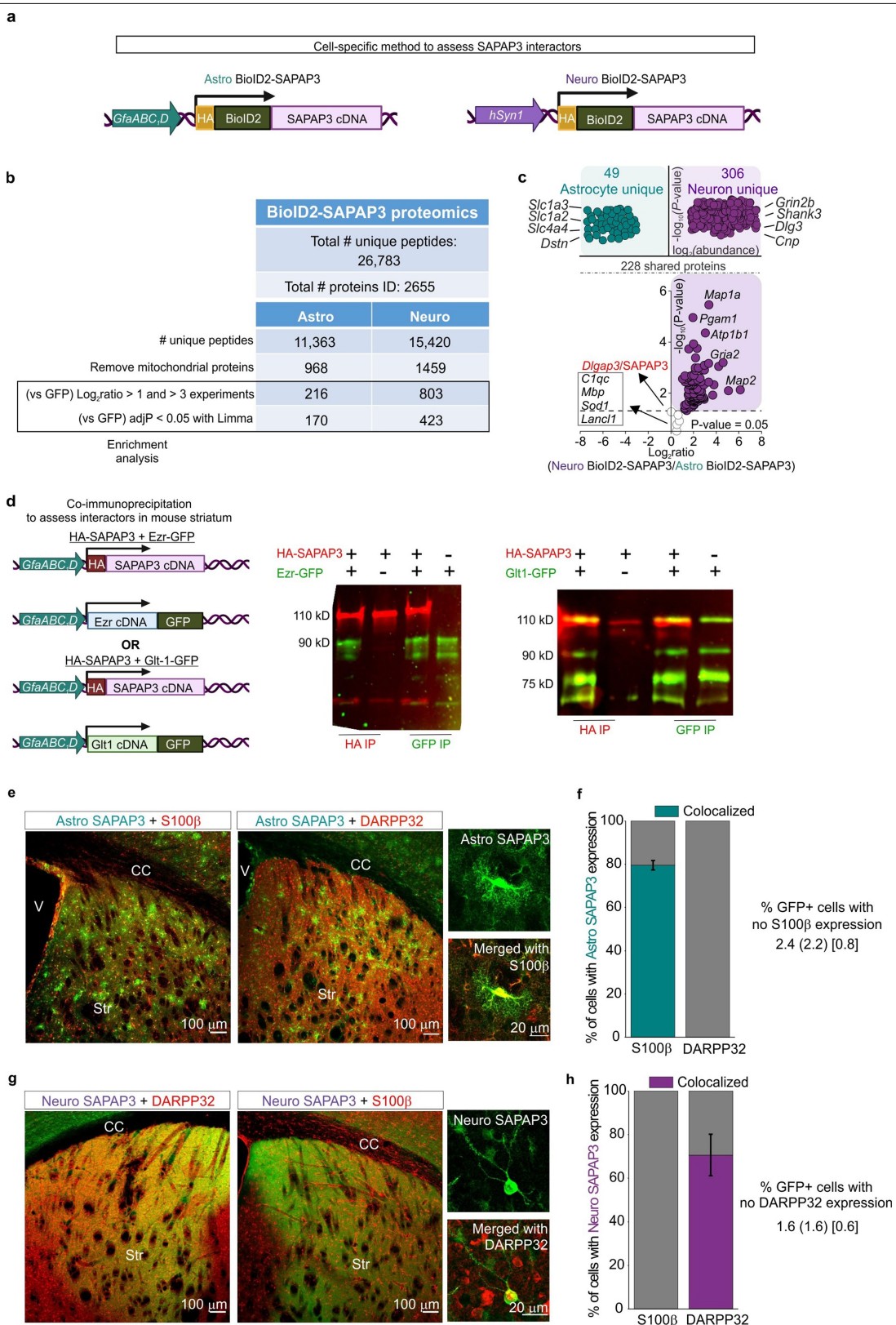

**Extended Data Fig. 18** | See next page for caption.

**Extended Data Fig. 18 | Assessment of cell-specific interactors of SAPAP3 in astrocytes using BioID2 proteomics. a**. Schematic of cell-specific BioID2 fused to SAPAP3 used in the AAV constructs. **b**. Table shows the number of peptides and proteins found in the cell-specific BioID2-SAPAP3 proteomics experiments. Each row shows the number of proteins after filtering. **c**. Label-free based quantification comparison of significant proteins ($Log_2FC > 1$ and FDR < 0.05 *versus* GFP controls) detected in the Astro BioID2-SAPAP3 and Neuro BioID2-SAPAP3 reveal unique astrocyte and neuron SAPAP3 interactors. Top half of the volcano plot shows 306 unique Neuro BioID2-SAPAP3 proteins and 49 unique Astro BioID2-SAPAP3 proteins when compared to each other. The top four most abundant proteins for each cell type are shown. Lower half of volcano plot shows comparison of 228 proteins that were common in both Astro BioID2-SAPAP3 and Neuro BioID2-SAPAP3. The five highest enriched proteins ($Log_2FC > 2$) for neurons are shown. Proteins that did not pass the enrichment threshold for either cell type are represented in the gray box. **d**. Schematic shows astrocyte specific HA tagged SAPAP3, GFP fused Ezrin, and GFP fused Glt-1 used in AAV constructs to assess interactions via co-immunoprecipitation. 16 week old wild type mice were injected in the striatum with one of the following combinations: HA-SAPAP3 + Ezr-GFP, HA-SAPAP3 + Glt-1-GFP, HA-SAPAP3 only, Ezr-GFP only, or Glt1-GFP only. Western blot shows the immunoprecipitation of either HA or GFP after protein complex isolation. The band 110 kD represents the HA-SAPAP3 band, while the 90 kD bands represent Ezrin-GFP (93 kD) or Glt1-GFP (92 kD). n = 4 mice per combination, 3 technical replicates. **e**. Representative images of immunostained mouse striatum injected with astrocyte-specific GFP-SAPAP3 (Astro SAPAP3). Left panel shows the immunostaining pattern with S100β as an astrocyte cell marker and right panel shows the immunostaining pattern with DARPP32 as a neuron cell marker. **f**. Bar graphs depicting the percent of S100β positive or NeuN positive cells with HA expression in a 20x magnification field of view. Teal portion of the bar graphs show the percent co-localization. Bottom descriptive statistics represent percent of HA+ cells that were not S100β positive as the mean (SD) [SEM] (n = 8 fields of view at 20x magnification from 4 mice) **c**. Representative images of immunostained mouse striatum injected with astrocyte-specific GFP-SAPAP3 (Astro SAPAP3). Left panel shows the immunostaining pattern with S100β as an astrocyte cell marker and right panel shows the immunostaining pattern with DARPP32 as a neuron cell marker. **g**. Representative images of immunostained mouse striatum injected with neuron-specific GFP-SAPAP3 (Neuro SAPAP3). Left panel shows the immunostaining pattern with DARPP32 as a neuron cell marker and right panel shows the immunostaining pattern with S100β as an astrocyte cell marker. **h**. Bar graphs depicting the percent of S100β positive or NeuN positive cells with HA expression in a 20x magnification field of view. Purple portion of the bar graphs show the percent co-localization. Bottom descriptive statistics represent percent of HA+ cells that were not DARPP32 positive as the mean (SD) [SEM] (n = 8 fields of view at 20x magnification from 4 mice). The image of the DNA constructs in panels a and d was created using BioRender (https://www.biorender.com/).

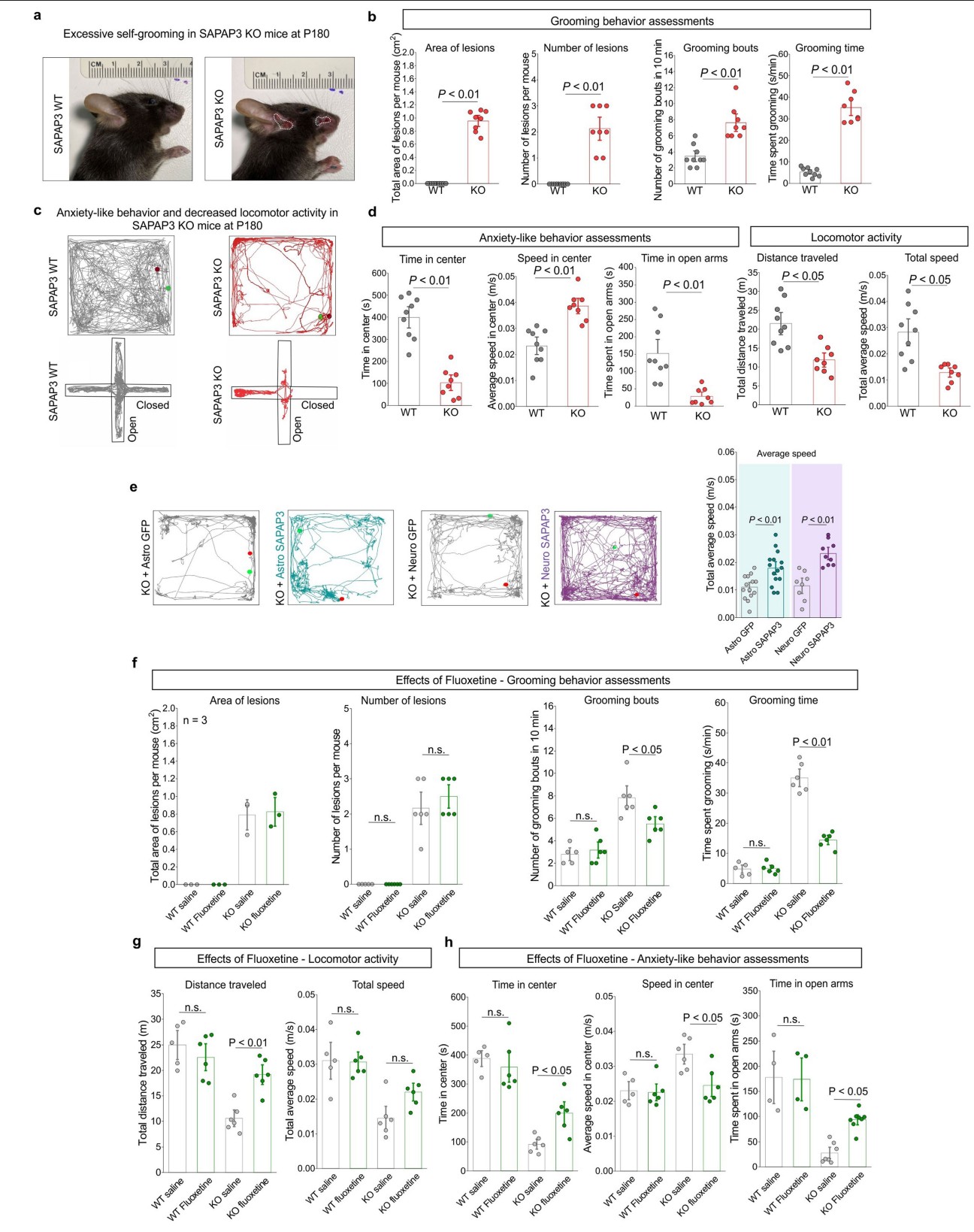

**Extended Data Fig. 19** | See next page for caption.

**Extended Data Fig. 19 | Behavior of SAPAP3 WT, KO, and fluoxetine treated mice. a**. Representative photographs of SAPAP WT and SAPAP3 KO mice at P180. White outline shows the border of each lesion. **b**. Bar graphs show assessment of grooming behavior in SAPAP3 WT and SAPAP3 KO by four different metrics: area of open lesions, number of lesions, grooming bouts, and time spent grooming. n = 9 for the SAPAP3 WT mice and n = 8 for the SAPAP3 KO mice. The mean and SEM are shown (area of open lesions: WT v KO, two-tailed one sample t-test, P < 0.001; number of lesions per mouse: WT v KO, two-tailed one sample t-test, P = 0.002; number of grooming bouts: WT v KO, two-tailed Mann-Whitney test, P = 0.009; time spent grooming: WT v KO, two-tailed unpaired t-test with Welch correction, P < 0.001). **c**. Representative traces of 20 min open field recordings for SAPAP3 WT and SAPAP3 KO mice at P180. Green dot depicts the location of the mouse at the start of the recording, while the red dot depicts the location of the mouse at the end of the recording. The center of the open field was 2.5 cm from the boundary wall. Bottom panels show representative traces of 10 min elevated plus maze recordings for SAPAP3 WT and SAPAP3 KO mice at P180. Green dot depicts the location of the mouse at the start of the recording, while the red dot depicts the location of the mouse at the end of the recording. **d**. Bar graphs show anxiety-like behavior and locomotor activity in SAPAP3 WT and SAPAP3 KO mice by four different metrics: time in center of the open field, speed in center of the open field, time spent in open arms of the elevated plus maze, total distance traveled, and total average speed. n = 9 for the SAPAP3 WT mice and n = 8 for the SAPAP3 KO mice. Mean and SEM are shown (time in center: WT v KO, two-tailed unpaired t-test with Welch correction, P < 0.001; speed in center: WT v KO, two-tailed unpaired t-test with Welch correction, P = 0.001; time in open arms: WT v KO, two-tailed unpaired t-test with Welch correction, P = 0.002; distance traveled: WT v KO, two-tailed unpaired t-test with Welch correction, P = 0.001; average speed: WT v KO, two-tailed unpaired t-test with Welch correction, P = 0.002).

**e**. Representative traces of 20 min open field recordings for P180 SAPAP3 KO mice injected with either Astro GFP (gray) or Astro SAPAP3 (teal) and Neuro GFP (gray) or Neuro SAPAP3 (purple) injected at P28. Green dot depicts the location of the mouse at the start of the recording, while the red dot depicts the location of the mouse at the end of the recording. The center of the open field was 2.5 cm from the boundary wall. Bar graph shows assessment of locomotor activity by the metric of average speed. n = 14 mice for the Astro GFP group, n = 15 mice for the Astro SAPAP3 group, n = 8 for the Neuro GFP group, and n = 9 for the Neuro SAPAP3 group. The mean and SEM are shown (two-way ANOVA with Bonferroni post-hoc test). **f**. Bar graphs show assessment of grooming behavior in SAPAP3 WT and SAPAP3 KO treated with either saline or fluoxetine for 7 days by four different metrics: area of open lesions, number of lesions, grooming bouts, and time spent grooming. n = 5-6 mice for each group. The mean and SEM are shown (number of lesions per mouse: Kruskal Wallis test; number of grooming bouts: One way ANOVA with Bonferroni correction; time spent grooming: One way ANOVA with Bonferroni correction). **g**. Bar graphs show assessment of locomotor activity in SAPAP3 WT and SAPAP3 KO treated with either saline or fluoxetine for 7 days by two different metrics: total distance traveled and total average speed. n = 5 mice WT + saline, n = 6 mice WT + fluoxetine, n = 6 mice KO + saline, n = 6 mice KO + fluoxetine. The mean and SEM are shown (total distance traveled: one way ANOVA with Bonferroni correction; total average speed: one way ANOVA with Bonferroni correction). **h**. Bar graphs show assessment of anxiety-like behaviors in SAPAP3 WT and SAPAP3 KO treated with either saline or fluoxetine for 7 days by three different metrics: time in center, speed in center, and time spent in open arms. n = 4 mice WT + saline, n = 4 mice WT + fluoxetine, n = 7 mice KO + saline and n = 8 mice KO + fluoxetine. The mean and SEM are shown (time in center: One way ANOVA with Bonferroni correction; speed in center: one way ANOVA with Bonferroni correction; time spent in open arms: Kruskal-Wallis test).

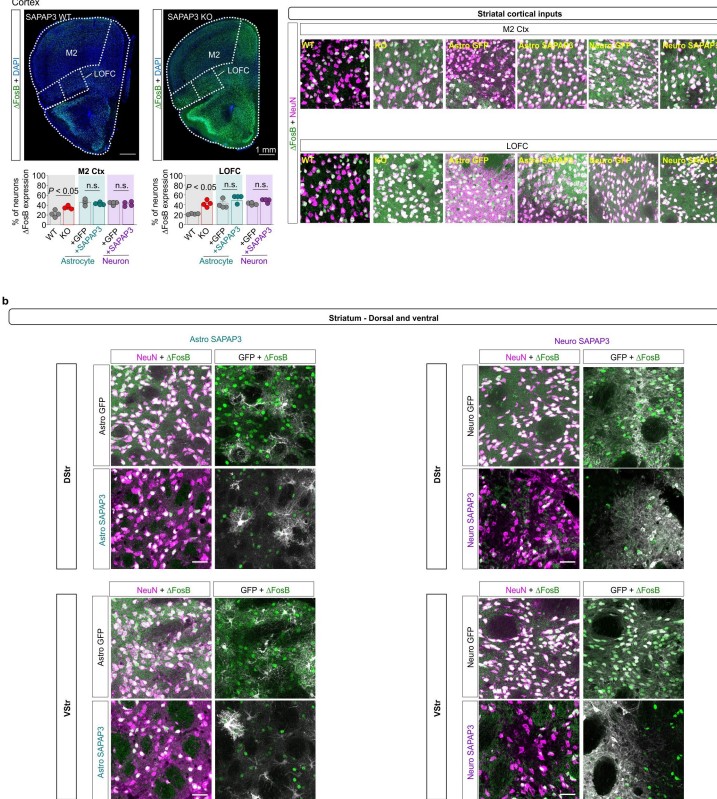

**Extended Data Fig. 20 | Mapping SAPAP3 associated changes with ΔFosB alterations. a.** Representative images show coronal sections containing both motor cortex (M2) and lateral orbitofrontal cortex (LOFC) in WT and SAPAP3 KO mice. Both LOFC and M2 of SAPAP3 KO mice show increased numbers of ΔFosB expressing cells. Scale bar represents 1 mm. Expanded images show neuronal (magenta) expression of ΔFosB (green) in M2 and OFC of wild type, SAPAP3 KO mice, SAPAP3 KO mice plus Astro GFP, SAPAP3 KO mice plus Astro SAPAP3, SAPAP3 KO mice plus Neuro GFP, and SAPAP3 KO mice plus Neuro SAPAP3. Scale bar represents 20 µm. Bar graphs depict the percent of NeuN+ neurons with ΔFosB expression. Bar denotes the mean, and error bars denote the SEM. n = 4 animals per group (One-way ANOVA with Bonferroni post hoc test). **b.** Representative zoom-in images of dorsal and ventral striatum in WT, SAPAP3 KO, SAPAP3 KO plus Astro GFP, SAPAP3 KO plus Astro SAPAP3, SAPAP3 KO plus Neuro GFP, and SAPAP3 KO plus Neuro SAPAP3 mice. Images show NeuN+ neurons (magenta) with expression of ΔFosB (green). Images also show the Astro GFP, Astro SAPAP3, Neuro GFP, and Neuro SAPAP3 signal (white) in the dorsal and ventral striatum. Scale bar represents 20 µm. This experiment was replicated four times.

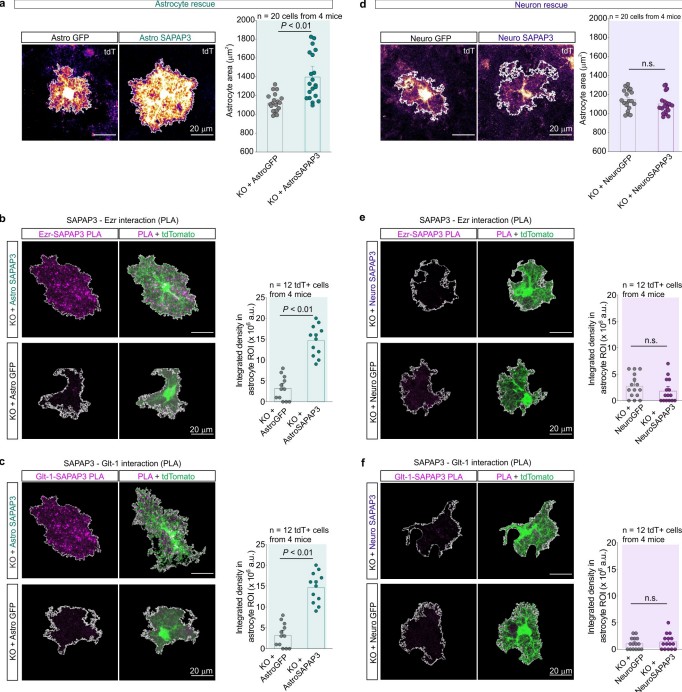

**Extended Data Fig. 21 | Effects of cell-specific rescue on astrocyte morphology and astrocyte-specific SAPAP3 interactions. a**. Representative images of tdTomato+ striatal astrocytes in SAPAP3 KO mice injected with Astro GFP (left) or Astro SAPAP3 (right). Bar graph shows the measurements of astrocyte territory area. n = 20 cells from 4 mice per group. The mean and SEM are shown (Astro GFP v Astro SAPAP3, two-tailed unpaired t-test with Welch correction, P = 0.001). **b**. Representative images of PLA puncta for SAPAP3 and Ezrin in SAPAP3 KO mice injected with Astro GFP or Astro SAPAP3. Images show SAPAP3-Ezr puncta within a tdTomato+ astrocyte. Bar graph denotes the integrated density in arbitrary units (a.u.) of the PLA signal within the astrocyte ROI. n = 12 tdTomato+ astrocytes from 4 mice per group. The mean and SEM are shown (Astro GFP v Astro SAPAP3, two-tailed unpaired t-test with Welch correction, P = 0.002). **c**. Representative images of PLA puncta for SAPAP3 and Glt-1 in SAPAP3 KO mice injected with Astro GFP or Astro SAPAP3. Images show SAPAP3-Glt-1 puncta within a tdTomato+ astrocyte. Bar graph denotes the integrated density in arbitrary units (a.u.) of the PLA signal within the astrocyte ROI. n = 12 tdTomato+ astrocytes from 4 mice per group. The mean and SEM are shown (Astro GFP v Astro SAPAP3, two-tailed unpaired t-test with Welch

correction, P < 0.001). **d**. Representative images of tdTomato+ striatal astrocytes in SAPAP3 KO mice injected with Neuro GFP (left) or Neuro SAPAP3 (right). Bar graph shows the measurements of astrocyte territory area. n = 20 cells from 4 mice per group. The mean and SEM are shown (Neuro GFP v Neuro SAPAP3, two-tailed unpaired t-test with Welch correction). **e**. Representative images of PLA puncta for SAPAP3 and Ezrin in SAPAP3 KO mice injected with Neuro GFP or Neuro SAPAP3. Images show SAPAP3-Ezr puncta within a tdTomato+ astrocyte. Bar graph denotes the integrated density in arbitrary units (a.u.) of the PLA signal within the astrocyte ROI. n = 12 tdTomato+ astrocytes from 4 mice per group. The mean and SEM are shown (Neuro GFP v Neuro SAPAP3, two-tailed Mann-Whitney test, P = 0.59). **f**. Representative images of PLA puncta for SAPAP3 and Glt-1 in SAPAP3 KO mice injected with Neuro GFP or Neuro SAPAP3. Images show SAPAP3-Glt-1 puncta within a tdTomato+ astrocyte. Bar graph denotes the integrated density in arbitrary units (a.u.) of the PLA signal within the astrocyte ROI. n = 12 tdTomato+ astrocytes from 4 mice per group. The mean and SEM are shown (Neuro GFP v Neuro SAPAP3, two-tailed Mann-Whitney test, P = 0.98).

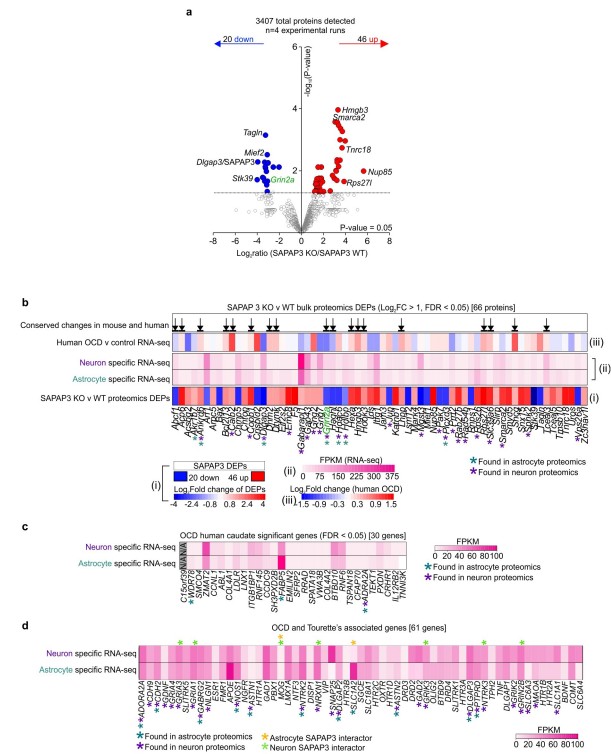

**Extended Data Fig. 22 | Bulk proteomics for WT and SAPAP3 KO mice and comparison to human striatal data. a**. Volcano plot shows the label-free based quantification comparison of 3407 proteins detected in the striatum. Red circles show upregulated proteins (Log2FC >1 and FDR < 0.05) while blue circles show downregulated proteins (Log2FC <1, FDR < 0.05) in the SAPAP3 KO striatum when compared to wild-type. The top 5 most down regulated and up regulated proteins are shown. Green label shows protein that also appeared in the neuron SAPAP3 interactome. **b**. List of the 66 proteins that were differentially expressed in the striatum of SAPAP3 KO mice when compared to wild type controls. Heat map and color scale (*i*) shows the Log$_2$fold change of the 66 proteins *versus* wild type control. Heat map and color scale (*ii*) shows the mRNA abundance (FPKM) of the 66 proteins in our neuron or astrocyte specific mouse RNA-seq datasets. Heat map and color scale (*iii*) shows the Log$_2$fold change at the mRNA level of the 66 proteins in human caudate of OCD subjects compared to controls. Arrowheads show genes that had conserved changes in the mouse SAPAP3 KO model and human OCD. Teal asterisks denote whether the protein was found in the astrocyte specific proteomics datasets, while the purple asterisks denote whether the protein was found in the neuron specific datasets. **c**. List of proteins shows the 30 significantly (FDR < 0.05) changed genes in human OCD caudate *versus* control. Heat map depicts the genes' respective mRNA abundances (FPKM) in our neuron or astrocyte specific mouse RNA-seq datasets. Teal asterisks denote whether the protein was also found in the astrocyte specific proteomics datasets while the purple asterisks denote whether the protein was found in the neuron specific datasets. The gene, C15orf39, was not found in our mouse datasets. **d**. List of 61 genes associated with OCD and Tourette's syndrome. Heat map depicts the genes' respective mRNA abundances (FPKM) in our neuron or astrocyte specific mouse RNA-seq datasets. Teal asterisks denote whether the protein was also found in the astrocyte specific proteomics datasets while the purple asterisks denote whether the protein was found in the neuron specific datasets. Orange asterisks denote whether the protein was an astrocytic SAPAP3 interactor, while green asterisks denote whether the protein was a neuronal SAPAP3 interactor.

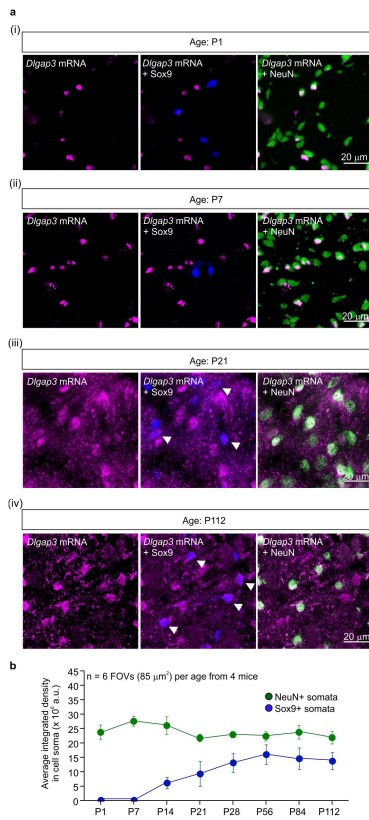

**Extended Data Fig. 23 | Assessment of *Dlgap3* mRNA expression across age using *in situ* hybridization. a**. (i-iv) Representative images of P1-P112 mouse striata with *Dlgap3* mRNA in magenta, Sox9 astrocyte marker protein in blue, and NeuN neuronal marker protein in green. White arrowheads show *Dlgap3* expression in Sox9+ astrocytes. **b**. Line graph shows the average integrated intensity of *Dlgap3* mRNA in either NeuN+ somata (green) or Sox9+ somata (blue) across 8 postnatal ages. Data are presented as the mean and the SEM; n = 6 FOVs per age from 4 mice per age.

# Reporting Summary

## Statistics

For all statistical analyses, confirm that the following items are present in the figure legend, table legend, main text, or Methods section.

| n/a | Confirmed | |
|---|---|---|
| ☐ | ☒ | The exact sample size ($n$) for each experimental group/condition, given as a discrete number and unit of measurement |
| ☐ | ☒ | A statement on whether measurements were taken from distinct samples or whether the same sample was measured repeatedly |
| ☐ | ☒ | The statistical test(s) used AND whether they are one- or two-sided<br>*Only common tests should be described solely by name; describe more complex techniques in the Methods section.* |
| ☐ | ☒ | A description of all covariates tested |
| ☐ | ☒ | A description of any assumptions or corrections, such as tests of normality and adjustment for multiple comparisons |
| ☐ | ☒ | A full description of the statistical parameters including central tendency (e.g. means) or other basic estimates (e.g. regression coefficient) AND variation (e.g. standard deviation) or associated estimates of uncertainty (e.g. confidence intervals) |
| ☐ | ☒ | For null hypothesis testing, the test statistic (e.g. $F$, $t$, $r$) with confidence intervals, effect sizes, degrees of freedom and $P$ value noted<br>*Give P values as exact values whenever suitable.* |
| ☒ | ☐ | For Bayesian analysis, information on the choice of priors and Markov chain Monte Carlo settings |
| ☒ | ☐ | For hierarchical and complex designs, identification of the appropriate level for tests and full reporting of outcomes |
| ☐ | ☒ | Estimates of effect sizes (e.g. Cohen's $d$, Pearson's $r$), indicating how they were calculated |

*Our web collection on statistics for biologists contains articles on many of the points above.*

## Software and code

Policy information about availability of computer code

**Data collection**
Proteomics: The spectra were collected using data dependent acquisition on Orbitrap Fusion Lumos Tribrid mass spectrometer (Thermo Fisher Scientific) with an MS1 resolution of 120,000 followed by sequential MS2 scans at a resolution of 15,000. Data generated by LC-MS/MS were searched using the Andromeda search engine integrated into the MaxQuant (Cox et al,. 2008) bioinformatic pipelines against the Uniprot Mus musculus reference proteome (UP000000589 9606) and then filtered using a "decoy" database-estimated false discovery rate (FDR) < 1%. Label-free quantification (LFQ) was carried out by integrating the total extracted ion chromatogram (XIC) of peptide precursor ions from the MS1 scan. These LFQ intensity values were used for protein quantification across samples. Label-free quantification was carried out by the MaxQuant software with integrated search engine, Andromeda (https://www.maxquant.org/).
RNA-seq: Sequencing was performed on Illumina NextSeq 4000 for 2 x 75 yielding at least 45 million reads per sample. Demultiplexing was performed with Illumina Bcl2fastq2 v 2.17 program. Reads were aligned to the mouse mm10 reference genome using the STAR spliced read aligner (Dobin et al 2013)
Behavior: Open field and elevated plus maze behavior data including locomotor and anxiety behavior was collected and analyzed simultanously by Anymaze (Stoelting Co. Wooddale, IL, USA).
Imaging for IHC, ICC, RNA-scope, and proximity ligation assay was conducted on an Olympus FV3000 confocal microscope using Fluoview software.
Western blot data was collected on a GE Amersham 680 imager and on a Licor odyssey infrared imager.

**Data analysis**
Proteomics: Label-free quantification was carried out by the MaxQuant software with integrated search engine, Andromeda (https://www.maxquant.org/). Principal component data visualization was conducted with R package Factoextra fviz 1.0.6 (https://rpkgs.datanovia.com/factoextra/reference/fviz_pca.html). Differential protein expression and enrichment analysis was conducted with Bioconductor R package, limma v 3.54 (https://bioconductor.org/packages/release/bioc/html/limma.html). Protein network visualization, including STRING analysis was conducted with Cytoscape v 3.8 (https://apps.cytoscape.org/apps/stringapp). The artMS package v 1.16 (https://bioconductor.riken.jp/packages/3.8/bioc/html/artMS.html) was used to re-format the maxquant results (evidence.txt file), to make them compatible with SAINTexpress program. SAINT protein interaction probability scoring was done through (http://saint-apms.sourceforge.net/Main.html).

RNA-seq: Differential gene expression and enrichment analysis used R package limmaVoom v 3.36 to process RNA counts (https://rdrr.io/bioc/limma/man/voom.html) and batch correction was done with R package ComBat v 3.46 (https://rdrr.io/bioc/sva/man/ComBat.html).
IHC, ICC, RNA-scope, proximity ligation assay, western blots: Microscopy data and western blot data was imported and analyzed on FIJI (ImageJ v 2.1) using the BioFormats importer for Olympus FV3000 acquired images.
Behavior, IHC, ICC, RNA-scope, proximity ligase assay and Western data were plotted with OriginPro 2018 (v 9.6.5) and statistical analysis was conducted with GraphPad Instat 3.

For manuscripts utilizing custom algorithms or software that are central to the research but not yet described in published literature, software must be made available to editors and reviewers. We strongly encourage code deposition in a community repository (e.g. GitHub). See the Nature Portfolio <u>guidelines for submitting code & software</u> for further information.

## Data

Policy information about <u>availability of data</u>

All manuscripts must include a <u>data availability statement</u>. This statement should provide the following information, where applicable:
- Accession codes, unique identifiers, or web links for publicly available datasets
- A description of any restrictions on data availability
- For clinical datasets or third party data, please ensure that the statement adheres to our <u>policy</u>

All the proteomic data are available at PRIDE with accession IDs PXD029257. The UniProt reference proteome used was UniProt UP000000589 9606 for Mus musculus. The RNA-seq data are available at GEO with accession ID GSE184773. All proteomic data are provided as supplemental information data tables 1-5. The analyzed RNA-seq data is provided as supplemental information table 6. All raw replicate data values used to generate the figures and the associated statistical tests are provided in supplemental information tables 7 and 8.

# Field-specific reporting

Please select the one below that is the best fit for your research. If you are not sure, read the appropriate sections before making your selection.

☒ Life sciences  ☐ Behavioural & social sciences  ☐ Ecological, evolutionary & environmental sciences

For a reference copy of the document with all sections, see <u>nature.com/documents/nr-reporting-summary-flat.pdf</u>

# Life sciences study design

All studies must disclose on these points even when the disclosure is negative.

| Sample size | Power analysis was conducted using values for power of 0.8 or higher and alpha of 0.1 or lower and an estimated effect size based on pilot data. Furthermore, group sizes were selected based on data from the use of similar models by our laboratory. |
|---|---|
| Data exclusions | No data was excluded from this manuscript |
| Replication | To verify the reproducibility of the experimental findings, all data collection was done in multiple batches comprising at least four replicates. The proteomic data analyses was conducted from 4 independently processed batches that each contained 8 mice  for each experimental group (in all cases). Behavioral data was conducted as the mice became available from the breeding colony and each experiment/recording was done in 2-3 batches containing between 3-6 mice per group. All experiments were successfully replicated. |
| Randomization | For proteomic experiments, mice were purchased from the supplier and each cage was randomly allocated to a group ad priori to AAV injections. For behavioral experiments, the mice were randomly allocated to a group as they became available and of age from the breeding colony in alternation. |
| Blinding | For the behavioral analyses, the investigators were blinded to group allocation during data collection, as numerical mouse IDs were the only identifier used. For analyses, the investigators were blinded to groups and mouse IDs by using the ImageJ/FIJI FIle Name Encrypter / Blind Analysis Tools plugin. Furthermore, behavioral experiments were run or analyzed by different people when possible.<br>For IHC, ICC,  Proximity ligase assay analyses: all file names were randomized/encripted using the ImageJ/FIJI FIle Name Encrypter / Blind Analysis Tools plugin so investigators were blinded to all groups and identifiers. Because some of these experiments required use of mice used for behavior experiments, investigators were also blinded during data collection by sole use of mouse ID as identifier. |

# Reporting for specific materials, systems and methods

We require information from authors about some types of materials, experimental systems and methods used in many studies. Here, indicate whether each material, system or method listed is relevant to your study. If you are not sure if a list item applies to your research, read the appropriate section before selecting a response.

## Materials & experimental systems

| n/a | Involved in the study |
|:---:|---|
| ☐ | ☒ Antibodies |
| ☐ | ☒ Eukaryotic cell lines |
| ☒ | ☐ Palaeontology and archaeology |
| ☐ | ☒ Animals and other organisms |
| ☒ | ☐ Human research participants |
| ☒ | ☐ Clinical data |
| ☒ | ☐ Dual use research of concern |

## Methods

| n/a | Involved in the study |
|:---:|---|
| ☒ | ☐ ChIP-seq |
| ☒ | ☐ Flow cytometry |
| ☒ | ☐ MRI-based neuroimaging |

# Antibodies

| | |
|---|---|
| Antibodies used | Primaries:<br>mouse anti-HA  (Biolegend, 901514)<br>rabbit anti-HA (abcam ab9110)<br>rabbit anti-β-actin (Abcam, ab8227)<br>rabbit anti-S100β (Abcam, ab13970)<br>rabbit anti-NeuN (Cell Signaling, 12943S)<br>guinea pig anti-neuN (Synaptic Systems, 266004)<br>rabbit anti-DARPP32 (Abcam, ab40801)<br>guinea pig anti-DARPP32 (Frontier Institute, DARPP-Gp-A250)<br>chicken anti-GFP (Abcam, ab13970)<br>mouse anti-RFP (Rockland, 600906379)<br>rabbit anti-RFP (Rockland, 600401379)<br>guinea pig anti- RFP (Synaptic Systems, 390004)<br>rabbit anti-PAICS (Invitrogen, 92985)<br>mouse anti-Nebl (Santa Cruz Biotechnology, 393784)<br>rabbit anti-Slc4a4/NBC1 (Novus, NBP32020)<br>rabbit anti-Arpc1a (Invitrogen 102339)<br>rabbit anti-Faim2 (Origene, TP300196)<br>rabbit anti-Hepacam (Novus Biologicals, 04983)<br>mouse anti- APC (abcam, ab16794)<br>rabbit anti- Olig2 (Millipore, AB9610)<br>rabbit anti- ΔFosB (Cell Signaling Technology, 14695S)<br>mouse anti- Glt1 (Santa Cruz Biotechnology, sc-365634)<br>mouse anti- Ezrin (BioLegend, 866401)<br><br>Secondaries:<br>Streptavidin-HRP (Sigma, RABHRP3)<br>IR-dye 800CW anti-rabbit (Li-Cor, 929-08972)<br>Alexa Fluor 488 goat anti-chicken (Molecular probes, A11039)<br>Alexa Fluor 647 goat anti-rabbit (Molecular probes, A21244)<br>Alexa Fluor 546 goat anti-mouse (Molecular probes, A11003)<br>Alexa Fluor 488 goat anti-rabbit (Molecular probes, A11008)<br>Alexa Fluor 546 goat anti-mouse (Molecular probes, A11003)<br>Streptavidin, Alexa Fluor 488 conjugate (Molecular probes, S11223)<br>Donkey anti-guinea pig Cy3 (Jackson ImmunoResearch, 706-165-148)<br>Goat anti-rabbit plus 647 (Invitrogen, A32733)<br>Goat anti-chicken plus 555 (Invitrogen, A32932) |
| Validation | Most of the antibodies used in this manuscript have been validated and reproduced by our lab across at least 7 manuscripts by checking cell specificity, background signal, and noting antigen specificity using western blot techniques (Srinivasan et al., 2016; Chai et al., 2017, Nagai et al., 2019; Yu et al., 2020; Diaz-Castro et al., 2019; Endo et al., 2022, Gangwani et al., 2023). All Khakh lab manuscripts. The SAPAP3 antibody was validated extensively by Dr. Guoping Feng's group by IHC in mouse brain and co-corresponding in situ hybridization (Welch et al., 2005).<br><br>Anti-Nebl, anti-NBC1/Slc4a4, anti-Faim2, anti-Arpc1a, anti-Hepacam, and anti-PAICS were stated to be validated by the supplier.<br><br>Anti-Nebl was validated by Santa Cruz biotechnologies by western blot and immunofluorescence. Furthermore this antibody has been used in mouse tissue IHC as referenced by Rudolph et al., 2020.<br>Anti NBC/Slc4a4 was validated by Novus Biologicals by "orthogonal" strategies in which the antibody was tested by IHC in different types of tissue including a negative control tissue where there is no positivity as expected (lymph node) in correlation with RNA-seq to predict where positivity can occur.<br>Anti-Faim2 was validated by Origen in mouse tissue by both western blot and IHC techniques in mouse brain.<br>Anti-Arpc1a was validated by Invitrogen with western blot and immunofluorescence techniques in tissue.<br>Anti-HEPCAM was validated by Novus biologicals with western blots across different types of lysates including mouse brain.<br>Anti-PAICS was validated by Invitrogen with IHC, ICC, and western blots in rats, which have the same PAICS sequence as mouse. |

# Eukaryotic cell lines

Policy information about cell lines

| | |
|---|---|
| Cell line source(s) | HEK 293T cells from ATCC (sex: female, RRID: CVCL_0045) |
| Authentication | The cells were authenticated by the supplier (ATCC) and by cell morphology. |
| Mycoplasma contamination | Cell line was not tested for mycoplasma |
| Commonly misidentified lines (See ICLAC register) | No misidentified cell lines were used in this study |

# Animals and other organisms

Policy information about studies involving animals; ARRIVE guidelines recommended for reporting animal research

| | |
|---|---|
| Laboratory animals | Proteomics experiments for Figs 1-3 were conducted with 7 week old wild-type C57BL/6NTac mice from Taconic Biosciences. Both male and female mice were used in alternating batches. Behavioral experiments were conducted with 5-6 month old targeted knockout mice for Dlgap3 from Jackson Laboratories (B6.129-Dlgap3tm1Gfng/J). Both males and females were used in alternating batches. |
| Wild animals | The study did not involve wild animals |
| Field-collected samples | This study did not use field-collected samples |
| Ethics oversight | All experiments were conducted in accordance with the National Institutes of Health (NIH) Guide for the Care and Use of Laboratory Animals and were approved and overseen by the Chancellor's Animal Research Committee (ARC) at the University of California, Los Angeles (UCLA) |

Note that full information on the approval of the study protocol must also be provided in the manuscript.

