## [Peer Review File · Nature]

Manuscript Title: Astrocyte-neuron subproteomes and obsessive-compulsive disorder mechanisms

Reviewer Comments & Author Rebuttals

Reviewer Reports on the Initial Version:

Referees' comments:

Referee #2 (Remarks to the Author):

Here Soto and colleagues present an exhaustive dataset generated using a wide array of AAV-targeted labeling techniques to purify proteins from astrocyte and neuron cell bodies and finer processes. This builds from the excellent previous work from this group highlighting brain-regions specific astrocyte protein and gene expression datasets in a number of control and mouse disease models.

Overall the data (which runs over 30+ figures) is massive in scope, and should provide interesting insights for a number of groups. Added with the application of the cell sub-type and cub-compartment targeting tools (BioID2), this study does represent a fantastic resource. I have some reservations about the validity of some clarifying statements (particularly of protein-protein interactions/predictions - see below), and some of the more compelling mouse model work should be included in the main manuscript rather than the extended data. Overall however this is a well-prepared manuscript, and the authors should be commended for comparing gene expression and protein levels in the same cells/cellular compartments - this is an interesting set of analyses that is not often completed.

I have a few statements that would benefit from clarification, and one or two questions that arose during the review of the manuscript.

Major points.

1. while I appreciated the authors excitement at their novel datasets, at times they are overly ambitious and their statements of fact, when they have only provided putative interactions/data. This is most evident in Fig 5

2. validation of cell type protein levels/gene expression (particularly astrocytes, see e.g Extended Data Fig 5a,b) - validation of non-neuronal targeting using Gfap-targeted AAVs by way of S100b labeling for astrocytes was a nice control. The authors should also check for incorrect oligodendrocyte lineage targeting as well as Gfap is expressed in some precursor cells (albeit there may be few of them at the ages used in this study, but they can expand rapidly in response to injury - like the AAV injection used here), and S100b staining may also capture some oligodendrocyte lineage cells.

Extended Data Fig 7 does go some way towards rectifying this - though the markers do not cover OPCs.

There remains some concern (or perhaps interesting underlying biology that the authors do not discuss) with the astrocyte-specific proteins defined here. Many of these are reported in other cells (e.g. neurons - Mapt; endothelial cells - Tjp1; among others). It would behoove the authors, especially for the larger node size/enriched proteins listed here, to comment on the specificity of these approaches, or at least validate with some in situ (were the proteins made by astrocytes, or engulfed after secretion from other cells? - both are exciting prospects), or is this a non-specific capture of these proteins that occurs during isolation (which does not discount the other exciting discoveries, but would be helpful for the unassuming reader to know of)

3. lack of clarity of animal numbers/biological replicates (e.g Extended Data Fig 7) - 'n=4

experimental runs' is listed in the figure legend - are these biological replicates or technical replicates? If the latter, this is analysis of only n=1.

Similarly, other n values should be more clearly stated - e.g. are the n=16/40/164 neurons/astrocytes in Extended Data Fig 1 from a single or multiple animals? Animal numbers should be clearly stated. The same is true for Ext Data Fig 4 (biological replicates of HEK cells? - there is confusion between 'cells' and 'batches'). The batch label appears to be from 8 (panel d)/11(panel h) coverslips collected from only 2 transfections - again effectively only a biological n of 2. T-testing on an n of two seems statistically inappropriate.

4. 'SAPAP3 in MSNs and in specific astrocyte subcompartments' section - while I appreciate this additional validation step, there should be some additional validation of markers that appear to be present in multiple cell types - this would more clearly highlight the specificity of the BioID2 method. It is not surprising that enriched proteins present in astrocytes but not other cells are then shown to be localized to astrocytes - but what of proteins that have been previously described in other cell types (e.g. MAPT, TJP1, as described above) - are these truly found in astrocytes, or is their inclusion here an artifact of the targeting/purification process for proteins? Unfortunately SAPAP3 is also one of these proteins - with many reports of it being present in nearly all CNS cells except astrocytes (OPC presence could overlap with the reported astrocyte location - given the expression of S100 by OPCs - the authors should clarify that these cells are not PDGFRa+ (or other OPC marker) just to really reiterate the specificity of their exciting methods.

(small editorial point) - all references to Fig 3 in this section are incorrect. The authors might have meant Fig 4 panels, but it is difficult to be sure.

5. comparison of gene expression and protein data - as mentioned above, the authors are to be commended for generating comparable datasets from the same cells/sub compartments, however there are numerous reasons that gene expression and protein levels may not correlate exactly. While I appreciate the authors argument that this should be considered, it is not simply that this is a lack of reliability of either method, but that there are real biological reasons why this would occur. For example, it could be a difference in protein v transcript half-life - a similar difference is seen in GFAP protein (long half life) and Gfap gene (only produced when required for robust morphological changes). Can the authors comment on the developmental changes associated with Dlgap3, or half-life of SAPAP3?

6. 'Distinct and convergent mechanisms for SAPAP3 in astrocytes and MSNs' section - this is the weakest section of the manuscript overall, simply because the reported protein-protein interactions for SAPAP3 are putative at best. If the authors would like to retain such strong statements they really should complete pull-down assays to clearly show interaction with SAPAP3 and the putative binding partners detected in silico and shown in this figure. If these experiments are not possible, the authors should clearly state that these are predicted putative interactions, and perhaps move these data to the extended data or remove them altogether

7. Extended Data Fig 26 - these data are great! As these are some of the more compelling data in this manuscript it was a shame to see them tucked away in the extended data. Personally I find the fluoxetine data more compelling than the protein-protein interaction in silico prediction data (Fig 5) and more easily interpretable than the heatmaps in panel k/l of Fig 6.

Minor points.

1. the authors have, for valid reasons, focused on the striatum in their investigations - however given they have completed some in situ and IF validation of a small number of proteins/genes, how broad are the assumptions that these findings are true for other subtypes of astrocytes, neurons, and astrocyte-neuron interactions? Where similar localizations seen in other brain regions?

2. Main text section, start of second paragraph - the statement 'Physiological signaling between

astrocytes and neurons also occurs between astrocyte processes and axon terminals and dendrites' is unclear, do the authors mean that the same astrocyte-neuron (cell body?) signaling occurs astrocyte-astrocyte and astrocyte-neuron dendrites/axons?

3. 'Brain region, cell and subcompartment specific proteomes' section (reference to Extended Data Fig 1a,b) - this figure does not refer to interactions between neurons and astrocytes, or astrocytes and astrocytes

4. Fig 3 - in its current format this figure is not helpful. The Astrocyte card is a bit small to be effectively read - it would serve a more useful place simply in the extended data along with the other 'cards'. GO term and venn diagram analyses of proteins identified are non-validating ways of interrogating the data. Instead, validating some of these hits (e.g. like those in Fig 2) with some IF/IHC/in situ and visualizing the cells in a WT mouse brain would be more helpful in validating the narrative. References to Fig 3c are not helpful due to it being too small, and the information contained in 6 separate sub-panels. Please refer to the Extended Data Fig 15 with exact panels under discussion

5. throughout - SAPAP3^{-/-} mice should be Dlgap3^{-/-} mouse (the gene is knocked out)

6. throughout - please check for small errors of gene/protein IDs and formatting - some are attributed incorrectly and can lead to confusion (given the impressive data from this group here and previously highlighting the importance of measuring both gene expression and protein levels)

6. confusion over SAPAP3 protein and Dlgap3 gene localization (referring to Fig 4) - Dlgap3 is clearly expressed in both the cell body and perhaps finer processes (hard to ascertain from Fig 4, panel H as the low mag (50uM scale bar larger image on the left) shows protein in all regions of the astrocyte, but the higher mag (20uM scale bar smaller image) shows only small dots of protein in the cell soma. There seems to be a disconnect between these two images, which may be reader error, but some clarification is required as this is an important part for the interpretation of these data. (particularly given the Dlgap3 mRNA is present only in the finer processes (panels in Fig 4g))

7. Figure 1, panel A - the coloration of somatic/membrane-bound protein is not as clear for neurons. Perhaps a small label at the top of these panels stating 'cell body' v 'cell membrane' would be an additional clarifying cue for readers

Referee #4 (Remarks to the Author):

A. Summary of the key results

1. Soto et al describe new findings on distinct proteome signatures of striatal neurons and astrocytes, including the identification of novel astrocytic expression and function of the protein Sapap3, which to date has been characterized as a post-synaptic scaffolding protein in glutamatergic synapses. The authors highlight that while there has been a rapid expansion in cell-specific gene expression analysis in brain tissue, less attention has been paid to cell-specific proteomic analysis which may better inform functional specialization of different cell types in the brain in health and disease. To address this gap, the authors leverage existing genetically targeted biotin ligase (BioID2) methods to develop a novel strategy to compare the proteome between cell types (neurons vs astrocytes) and cellular sub-compartments (membrane vs cytosol and 5 further astrocytic compartments), by using different AAV serotypes, gene promoters and compartment tags to achieve selective targeting for proteomic analysis. This provides a rich dataset that describes specific proteins, biological processes and protein networks that are differentially expressed between neurons and astrocytes and within different subcellular compartments. Surprisingly, this revealed astrocytic expression of the synaptic scaffolding protein Sapap3, which has been associated with obsessive compulsive disorder in animal and clinical studies.

B. Originality and significance: if not novel, please include reference

2. While the finding that a synaptic protein is expressed in astrocytes, and that its selective rescue in astrocytes in KO mice improves disease relevant phenotypes is novel and surprising, the impact would be significantly enhanced if the authors discovered what Sapap3 is doing in astrocytes that may contribute to OCD pathophysiology. Further, the significance and value of the proteomics results are somewhat lost in the extensive details about the method validation and long lists of differentially expressed proteins, enriched biological process and interaction map analyse, which feel like something you might expect in a "Resource" paper. Does this data provide new insight about neuron or astrocyte biology, or the specific changes in striatal neurons and astrocytes that contribute to OCD that could be targeted by new treatments? In its current state, this paper contains valuable descriptive results on proteomics, and causal manipulations demonstrating astrocyte and neuronal roles of Sapap3, however the lack of new data to understand differences in neuron and astrocyte biology or their contribution to disease limits the significance of the findings.

C. Data & methodology: validity of approach, quality of data, quality of presentation

Main points

See E point 2 on grooming bout measure.

3. Extensive space and detail is dedicated to the description of the cell type and compartment specific proteome results (about 2.5/6.5 pages in main text and 3/6 figures), similar to what might be expected in a "Resource" style paper. In many places the novelty and value of the findings is not clear, besides that they provide a resource for data mining and developing new hypotheses to test (as was done for Sapap3). Given that the journal typically allows for only 3-4 modest figures and 2000-2500 words for the main text, this is probably going to need to be cut substantially for the final version, if the manuscript is published in this journal. Suggestions to achieve this include:

a. The description of figure 1 in the text focussing on an explanation of how the methods were validated and how neurons and astrocytes were found to show expected differences in protein expression using the new methodological approach could be moved to a supplement. Instead, it would be valuable for the authors to more thoroughly explain how their data and interpretation of figure 1 provide new insight on neuron and astrocyte function.

b. Could the first paragraph describing results from extended figures 3-6 validating the specificity of the approach be moved to a discussion in the supplement?

c. Could figure 3a and b be integrated into figure 2, so that all "astrocyte cards" are included as supplemental figures?

Minor points:

4. Grooming in Sapap3 KO mice relative to WT mice (extended data figure 25) is much higher and more uniform than previous reports in similar aged mice (e.g. <https://doi.org/10.1038/s42003-020-01611-y>, DOI: 10.1111/gbb.12557, <https://doi.org/10.1038/s41386-018-0307-2>, <https://doi.org/10.1016/j.jneumeth.2017.05.026>, <https://doi.org/10.1038/s41598-021-88769-5> all show heterogenous grooming phenotype in similar aged mice). Can you provide any explanation as to why your colony would have a more robust/intense/penetrant phenotype (or a published example with similar levels of behavioural severity)? Do you select heterozygous breeders based on susceptibility to grooming phenotype? Do you house animals individually or in group cages during studies? It is important to recognize these differences, and look for possible explanations to enhance reproducibility of these effects.

5. Can you please include details of whether or not Sapap3 KO littermates were used in active and control groups for studies (vs non littermate controls). i.e. were litters with multiple KO mice split between experimental groups?

D. Appropriate use of statistics and treatment of uncertainties

Main points:

6. For figure 6, Mann-Whitney Test is used on grooming bout data for one experiment (astro-Sapap3), whereas Unpaired t test with Welch was used for the other (neuro-Sapap3) plotted in the same figure (grooming bouts). By eye, the astro-sapap3 data appear more normally distributed

and the neuro-Sapap3 look more skewed (which would require the non parametric Mann-Whitney Test). Were these tests used correctly? Given that significant weight was placed on the idea that astro-Sapap3 has an effect on grooming bouts but neuro-Sapap3 does not, it doesn't seem appropriate to use different types of statistical tests in the two datasets (to address whether these have different effects they should really be tested together in a two way ANOVA).

7. "equivalent sapap3 expression in neurons and astrocytes" (Fig 4a-b). What statistics were performed to analyse this data (were any statistical analyses performed)? These data look like they would show statistical difference in expression (especially 4b and 4a for "MSN"-cyto vs Astrocyte-cyto).

Minor points:

8. The methods section says that "exact p-values are shown in the figure panels or in the figure legends". Although these are available in the supplement they are not available in the figure panel/legend for figure 6, please add them.

E. Conclusions: robustness, validity, reliability

Main points:

9. I strongly recommend that the title "reveal mechanisms of obsessive-compulsive disorder" should be adjusted. This is misleading, given that the majority of studies were conducted in an OCD-relevant mouse model, and studies on human RNA seq data do not "reveal mechanisms of OCD". It is important to make clear distinctions, and not over state what can be inferred from animal research and descriptive post mortem analysis, particularly in an influential journal (and in the title of article). "Astrocyte and neuron subcompartment proteomes reveal mechanisms of Sapap3 function relevant to obsessive-compulsive disorder" may be more appropriate for the title. Other areas where the authors should be careful with language around clinical relevance include "how astrocyte and neuronal molecular mechanisms contribute to a psychiatric disease" (abstract) and "We focused on this discovery because of the potential clinical relevance to understanding how astrocytes and neurons may both contribute to OCD-like behaviors in mice." (end page 4).

10. The statements "astrocytic SAPAP3 and neuronal SAPAP3 preferentially rescued repetitive and anxiety like behaviors, respectively", "the effect of astrocytic GFP-SAPAP3 was significantly greater for rescue of the number of self-grooming bouts when compared to neuronal GFP-SAPAP3" and "our experiments show that astrocyte and neuron mechanisms converge to drive dysfunction in OCD, and that targeting both is necessary to be therapeutically effective" are not supported by the experimental data, and all mention of separate effects of these interventions should be removed. While it is true that the authors provide evidence that only neuronal, and not astrocytic, SAPAP3 can rescue anxiety-like behaviour (on a single measure of anxiety, which is not ideal and should be mentioned as a limitation), there is strong evidence that both neuronal and astrocytic SAPAP3 rescue grooming on all measures except # of grooming bouts. Importantly, the definition of grooming bouts was courser than what is typically used in the literature, which questions the sensitivity of this bout measure. Specifically, "Separate grooming bouts were considered when the pause was more than 5 sec or if behaviors other than self-grooming occurred" was used to define grooming bouts, which is much longer than what is typically used to define new grooming bouts in mice (e.g. papers using 2 second pause: <https://doi.org/10.1016/j.jneumeth.2017.05.026>, <https://doi.org/10.1038/s42003-020-01611-y>, doi:10.1038/nn.4395 and 3 seconds pause used by Welch et al 2007 Nature doi:10.1038/nature06104). In addition to this, less animals were included in the neuro-SAPAP3 rescue experiment compared to the astrocyte-SAPAP3 experiment, and it appears that this analysis may be underpowered (the absolute difference between means is similar between astro-sapap3 and neuro-sapap3 studies, but SEM error bars are much larger due to the smaller sample size for neuro-sapap3 group). To say that astro-Sapap3 had a great effect on grooming than neuronal-Sapap3, it would be more accurate to use a two way ANOVA comparing these groups. Given these limitations of grooming bout analysis, and effects of both neuro- and astro-Sapap3 on grooming time and lesions, these effects on grooming should not be described as different.

Minor points:

11. "astrocytic rescue of GFP-SAPAP3 resulted in beneficial effects comparable to fluoxetine". This

is misleading given that Fluoxetine improves anxiety but astrocytic Sapap3 does not. Please remove or adjust this statement.

F. Suggested improvements: experiments, data for possible revision

Main points:

12. The discovery that Sapap3 is expressed in astrocytes is very significant, however the impact of this finding is substantially reduced without some understanding of the biological function of Sapap3 in astrocytes. Does Sapap3 act as a scaffold in astrocytes like it does in neurons? Does the presence vs absence of Sapap3 have some functional significance (e.g. effects on glutamate uptake/homeostasis and the actin cytoskeleton)? Are there any simple functional assays that can be performed in WT vs Sapap3 KO vs Astro-Sapap3 rescue, to begin to demonstrate whether Sapap3 does anything to the function of its binding partners described in figure 5?

G. References: appropriate credit to previous work?

Minor points:

13. "A circuit based understanding of OCD is slowly emerging 30,60,61, but striatal molecular and cellular mechanisms giving rise to OCD-like phenotypes remain to be understood". It would be appropriate to reference papers that have begun to examine striatal cellular mechanisms of OCD-relevant phenotypes, including the contribution of PV+ interneurons (DOI: 10.1126/science.1232380), D2 MSNs (DOI: 10.1038/s41386-021-01161-9) and astrocytes (<https://doi.org/10.1038/npp.2015.26> and your own studies <https://doi.org/10.1016/j.neuron.2018.08.015>). A discussion of how the current findings relate to this existing work would also be valuable for readers.

H. Clarity and context: lucidity of abstract/summary, appropriateness of abstract, introduction and conclusions

Writing was very clear, although the context related to OCD was at times overstated (see)

Main point:

14. "MSN" should be replaced with "neuron" in all instances describing studies that rely on viruses using hsyn promotor, given that hsyn promotor is "pan-neuronal" and is expected to transfect striatal interneurons. Although interneurons constitute only a small proportion of striatal neurons, they are functionally significant in the context of this work, with parvalbumin interneurons providing strong inhibitory control over MSNs, with evidence that this is capable of normalizing compulsive grooming in Sapap3 KO mice (doi: 10.1126/science.1232380. and more recently <https://doi.org/10.1101/2022.01.10.475745>).

I. Other minor points

15. "Relative to the AAV fluorescent protein controls, we detected ~800-1600 proteins enriched within each of the Astro BioID2, Astro Lck-BioID2, Neuro BioID2 or Neuro Lck-BioID2 groups following in vivo expression." Where does this value come from? Is it associated with any panel in figure 1? For example Fig1e shows 1640 +262 enriched proteins in Neuro-Lck-BioID2 group, whereas other groups have ~500-600 based on the ven diagram values.

16. In the text, figure 4a is mislabelled figure 3A (lines 1-2 of page 5), and figures 5c-d are mislabelled (d and e respectively, page 5 late paragraph 2).

17. For open field diagrams, can the square within the arena be re-sized so that it positioned at the centre/periphery boundary that is used to quantify anxiety? This would be more informative, and without checking the legend carefully its easy to assume that that is what the inner line represents this boundary. The methods say that they chamber was 28cm wide, and the figure 6 diagrams have an inner boundary ~1/6 of the total width (~4.67cm; the extended figure 25 looks even larger and is ~1/4 the chamber width or 7cm wide). The methods say the actual boundary was 2.5cm from the chamber walls, so this line in the figure panel is misleading. Please check which is correct and update the figure panels to reflect the actual boundary, and include this in the figure legend.

18. Ext fig 24: please show higher resolution images so that it is clear whether cells co-express markers (similar to Ext Fig 21). Neuro-Sapap3 expression is very difficult to see. For all

colocalization studies to examine cell type selectivity of viral constructs (Ext Fig 5, 21, 24), did any GFP or HA+ cells not express S100 β or DARPP-32/NeuN? Please include this data in the supplement (or make it clearer if already there).

Author Rebuttals to Initial Comments:

Dear Dr. Rowland,

Thank you for the careful review of our manuscript and for the opportunity to revise it. You will see in the point-by-point responses below that we have addressed every single Reviewer comment with new data, analyses, and revisions. The revisions are marked in blue text.

In summary guidance you wrote “...While they find your work of potential interest, as do we, they have raised important concerns that in our view need to be addressed before we can consider publication in Nature. In particular, the reviewers felt that the paper serves as an excellent resource, but were less convinced about the novel biological insights. Do you think you will be able to address these concerns?”. First, we are pleased that the Reviewers appreciated that the study represents an excellent resource, which is critical for the field. Second, with new experiments over the last six months, we were able to address all the concerns and now provide further new biological insight. The new data are described in detail in the sections that follow and in the revised manuscript. Briefly, we show

- (i) How the astrocyte plasma membrane proteome is altered in SAPAP3 KO mice (Fig 4f)
- (ii) How SAPAP3 affects astrocyte morphology and the actin cytoskeleton (Fig 4f-j, Fig 6c-d)
- (iii) How validated SAPAP3-Ezrin and SAPAP3-Glt1 protein-protein interactions are reduced in the context of SAPAP3 KO mice relative to controls (Fig 4c-e)
- (iv) How astrocyte morphology, the actin cytoskeleton, and SAPAP3-Ezrin and SAPAP3-Glt1 protein-protein interactions are restored following SAPAP3 rescue (Fig 4c-e; Fig 6c,d)
- (v) How cell-specific rescue of SAPAP3 restores Δ FosB levels in striatal neurons of SAPAP3 KO mice (Fig 6a).

Furthermore, as requested by Reviewer 4, we have reduced the length of the methodological sections (by moving data to Extended data) and revised some of the language we previously used - in order to tone down some aspects (as advised).

In addition, we have formatted the revised manuscript according to your detailed guidelines, and include all the requested sections (statistics, reproducibility, data and code availability, source data, and raw data (gels), raw replicate values). The figures are formatted according to guidelines.

We recognize that we have much Extended data (noted positively by Reviewer 1), but we included all the results to be sure the Reviewers see that we have addressed their comments in full.

Please let us know if there is something we have missed. With these changes, we hope the manuscript will be considered suitable for publication.

{REDACTED}

Reviewer #1:

“Here Soto and colleagues present an exhaustive dataset generated using a wide array of AAV-targeted labeling techniques to purify proteins from astrocyte and neuron cell bodies and finer processes. This

builds from the excellent previous work from this group highlighting brain-regions specific astrocyte protein and gene expression datasets in a number of control and mouse disease models.

Overall the data (which runs over 30+ figures) is massive in scope, and should provide interesting insights for a number of groups. Added with the application of the cell sub-type and sub-compartment targeting tools (BioID2), this study does represent a fantastic resource. I have some reservations about the validity of some clarifying statements (particularly of protein-protein interactions/predictions - see below), and some of the more compelling mouse model work should be included in the main manuscript rather than the extended data. Overall however this is a well-prepared manuscript, and the authors should be commended for comparing gene expression and protein levels in the same cells/cellular compartments - this is an interesting set of analyses that is not often completed.

I have a few statements that would benefit from clarification, and one or two questions that arose during the review of the manuscript.”

Thank you for these supportive comments and for your enthusiasm for the study. We have addressed all your comments with point-by-point responses below. These have made the study stronger.

Major points.

1. *“While I appreciated the authors excitement at their novel datasets, at times they are overly ambitious and their statements of fact, when they have only provided putative interactions/data. This is most evident in Fig 5”*

Thank you for this guidance. We have now toned down the language in regards to the putative interactions, called them “putative” as appropriate in the text, and have validated the key ones in relation to our study with co-immunoprecipitation and with the proximity ligation assay. These new data are included in the revised Fig 4 and the accompanying Extended data (see specific comments below).

We also take this opportunity to clarify that STRING database protein-protein interaction maps shown in the astrocyte subproteome cards can be filtered based on published affinity purification mass spectrometry (AP-MS experiments). We have done this for all the interaction maps shown in the astrocyte subcompartment section (Fig 2 and Extended Data figs 14-19) and have now added this clarification to the methods section called “Protein networks and protein-protein interaction analysis” on page 23.

2. *“Validation of cell type protein levels/gene expression (particularly astrocytes, see e.g Extended Data Fig 5a,b) - validation of non-neuronal targeting using Gfap-targeted AAVs by way of S100b labeling for astrocytes was a nice control. The authors should also check for incorrect oligodendrocyte lineage targeting as well as Gfap is expressed in some precursor cells (albeit there may be few of them at the ages used in this study, but they can expand rapidly in response to injury - like the AAV injection used here), and S100b staining may also capture some oligodendrocyte lineage cells. Extended Data Fig 7 does go some way towards rectifying this - though the markers do not cover OPCs.*

There remains some concern (or perhaps interesting underlying biology that the authors do not discuss) with the astrocyte-specific proteins defined here. Many of these are reported in other cells (e.g. neurons - Mapt; endothelial cells - Tjp1; among others). It would behoove the authors, especially for the larger node size/enriched proteins listed here, to comment on the specificity of these approaches, or at least validate with some in situ (were the proteins made by astrocytes, or engulfed after secretion from other cells? - both are exciting prospects), or is this a non-specific capture of these proteins that occurs during isolation (which does not discount the other exciting discoveries, but would be helpful for the unassuming reader to know of)”

In regards to the first paragraph concerning oligodendrocytes and OPCs; thank you for these comments. We have now re-analyzed the proteomic data to cover OPCs in order to assess OPC protein detection in our datasets. We found no OPC proteins such as *Olig2*, *Tnr*, *Pdgfra*, or *Cacng4* in our datasets (Extended data Fig 6). We have also now performed immunostaining of HA-BioID2 and of the streptavidin-labelled proteins with oligodendrocyte and OPC markers APC (CC1) and Olig2 to assess any off-target oligodendrocyte and OPC expression of BioID2 or of streptavidin labeling. We found no detectable HA-BioID2 or biotinylated proteins within oligodendrocytes and OPCs. These data are shown below in Reviewer Fig 1. In Reviewer Fig 1, panels a and b show cytosolic Astro BioID2 and biotinylated proteins co-stained with S100 β and with either APC (CC1) or Olig2 oligodendrocyte and OPC markers. Panel c shows the quantification as a percent of cells with either HA-BioID2 or streptavidin (biotinylated protein) expression. There was no detectable expression of either HA or biotinylated proteins in APC or Olig2 cells. Reviewer Fig 1d-f show similar analysis, but with the membrane targeted Astro-Lck-BioID2. Here again, no expression was seen in CC1 or Olig2 positive cells. These data clearly show that neither the astrocyte targeted constructs nor the biotinylated proteins are found in oligodendrocytes or OPCs.

Reviewer Fig 1. BioID2 immunostaining with myelinating oligodendrocyte and OPC markers. a. Representative images of immunostained mouse striatum injected with astrocyte-specific cytosolic BioID2 and treated with biotin for 7 days. Top panels show 40x magnified images of the staining pattern with S100 β as an astrocyte cell marker or adenomatous polyposis coli (APC) [aka CC1] as a myelinating oligodendrocyte cell marker¹. Bottom panels show 40x magnified images. **b.** Representative images of immunostained mouse striatum injected with astrocyte-specific cytosolic BioID2 and treated with biotin for 7 days. Top panels show 40x magnified images of the staining pattern with S100 β as an astrocyte cell marker or Olig2 as an oligodendrocyte precursor cell (OPC) marker. Bottom panels show 40x magnified images. **c.** Bar graphs depict the percent of S100 β +, APC+, or Olig2+ cells with cytosolic BioID2-HA or biotinylated protein expression. 8 FOVs from 4 mice. **d.** Representative images of immunostained mouse striatum injected with astrocyte-specific plasma membrane Lck-BioID2 and treated with biotin for 7 days. Top panels show 40x magnified images of the staining pattern with S100 β as an astrocyte cell marker or APC as a myelinating oligodendrocyte cell marker. Bottom panels show 40x magnified images. **e.** Representative images of immunostained mouse striatum injected with astrocyte-specific Lck-BioID2 and treated with biotin for 7 days. Top panels show 40x magnified images of the staining pattern with S100 β as an astrocyte cell marker or Olig2 as an oligodendrocyte precursor cell (OPC) marker. Bottom panels show 40x magnified image. **f.** Bar graphs depict the percent of S100 β +, APC+, or Olig2+ cells with cytosolic BioID2-HA or biotinylated protein expression. 8 FOVs from 4 mice.

We would also like to clarify that we do not disagree with the Reviewer's comment that some S100 β positive oligodendrocytes and OPCs exist in the striatum. Our own analyses of *Aldh1l1*-tdTomato mice shows that 92% of S100 β positive cells are astrocytes and ~8% are oligodendrocytes/OPCs. However, the key point here is that the astrocyte AAV vectors we use to target astrocytes do not target oligodendrocytes or OPCs and that is clear from Reviewer Fig 1 and from previous work with these AAVs that includes RNA-seq, which showed highly selective astrocytic gene expression²⁻⁹.

In regards to the second paragraph about validating some of the proteins, which also comes up again later, we have now done so for *Crym*, *Mapt*, and *Tjp1* as suggested. These new data are reported in Extended data Fig 10 and clearly show expression within astrocytes by RNAscope of these mRNAs. That figure also includes quantification of the RNAscope puncta in *Aldh1l1* tdTomato and S100 β positive cells. We also include plots from Dropviz.org that confirms the Reviewer's comment that *Mapt* and *Tjp1* are found in other cells. However, our evaluations and the Dropviz.org analyses shown in Extended data Fig 10 clearly show they also exist in astrocytes. We thank the reviewer for this comment, which we have addressed and it has made our study stronger.

3. *“Lack of clarity of animal numbers/biological replicates (e.g Extended Data Fig 7) - 'n=4 experimental runs' is listed in the figure legend - are these biological replicates or technical replicates? If the latter, this is analysis of only n=1. Similarly, other n values should be more clearly stated - e.g. are the n=16/40/164 neurons/astrocytes in Extended Data Fig 1 from a single or multiple animals? Animal numbers should be clearly stated. The same is true for Ext Data Fig 4 (biological replicates of HEK cells? - there is confusion between 'cells' and 'batches'). The batch label appears to be from 8 (panel d)/11(panel h) coverslips collected from only 2 transfections - again effectively only a biological n of 2. T-testing on an n of two seems statistically inappropriate.”*

In retrospect we did a poor job of reporting the n numbers. We have fixed this and have added all the n numbers accurately in the figure legends for every figure. For example, each proteomic experiment consists of 4 experiments with 8 mice in each. Furthermore, the HEK-293 cells are from 3-4 transfections. All other data sets are at least n = 4 independent experiments. We did not, and do not, perform any statistical tests on n = 2. Note also that the raw replicate values and the statistical tests are all provided in Extended data tables 7 and 8.

4. *“SAPAP3 in MSNs and in specific astrocyte subcompartments' section - while I appreciate this additional validation step, there should be some additional validation of markers that appear to be present in multiple cell types - this would more clearly highlight the specificity of the BioID2 method. It is not surprising that enriched proteins present in astrocytes but not other cells are then shown to be localized to astrocytes - but what of proteins that have been previously described in other cells types (e.g. MAPT, TJP1, as described above) - are these truly found in astrocytes, or is there inclusion here an artifact of the targeting/purification process for proteins?*

Unfortunately SAPAP3 is also one of these proteins - with many reports of it being present in nearly all CNS cells except astrocytes (OPC presence could overlap with the reported astrocyte location - given the expression of S100 by OPCs - the authors should clarify that these cells are not PDGFR α + (or other OPC marker) just to really reiterate the specificity of their exciting methods. (small editorial point) - all references to Fig 3 in this section are incorrect. The authors might have meant Fig 4 panels, but it is difficult to be sure.”

This comment relates to comment 2 above and makes a similar point about validation and about oligodendrocytes and OPCs in a different way. In terms of validation of *Mapt* and *Tjp1*,

we now include new data in Extended data Fig 10 on this. These data show clear evidence by RNAscope for the expression of these within astrocytes that were genetically labelled with *Aldh1l1* tdTomato or by S100 β immunostaining, which is also supported in that figure by plots from a published scRNA-seq dataset¹⁰ (dropviz.org). Furthermore, *Tjp1* has been described in astrocytes¹¹ and its mRNA was detected as highly expressed within astrocytes in several astrocyte gene expression studies (e.g. brainrnaseq.org and liddelowlab.com/gliaseq). This is also true for *Mapt*. These RNA databases are independent of our lab databases and support our findings. We also clarify that “highly enriched” does not mean “astrocyte unique”. We have made many edits to make this distinction clear. Irrespectively, Extended data Fig 10 addresses the reviewer’s comment about *Tjp1* and *Mapt*. We have added clarifying language throughout the revised manuscript.

We think there is a slight misunderstanding and we don’t understand the comment on SAPAP3 in this context. We are excited by this discovery, because our study identifies it as being expressed in astrocytes for the first time. So the Reviewer is correct that this is new and has not been previously reported! Furthermore, in terms of possible expression of SAPAP3 in S100 β positive cells that are not astrocytes, in Fig 3 we have now replaced the S100 β immunostaining with assessments of SAPAP3 and *Dlgap3* expression in astrocytes that have been genetically labelled with tdTomato. Our data provide compelling evidence across multiple independent approaches for astrocytic SAPAP3 expression. In summarizing our data, on page 4 we write

“The proteomic findings were supported by neuron and astrocyte-specific RNA-seq, which showed similar *Dlgap3* expression (Fig 3b; FDR < 0.05). In accord, single cell RNA-seq⁸ analyses of *Dlgap3* showed similar expression in neurons and astrocytes (Fig 3c). To further validate the astrocytic data, we performed RNAscope fluorescence *in situ* hybridization and detected abundant *Dlgap3* mRNA in genetically labelled tdTomato astrocytes and this was abolished in SAPAP3 KO mice (Fig 3d,f; P < 0.01). We also detected abundant SAPAP3 protein in tdTomato labeled astrocytes and the immunostaining was significantly reduced in SAPAP3 KO mice (Fig 3e,g; P < 0.01). Together, data from proteomics, cell specific RNA-seq, scRNA-seq, RNAscope, and IHC in WT and SAPAP3 KO mice provide strong evidence that astrocytes express SAPAP3 (Figs 1e, 2d, 2e, 3a-e).”

In relation to the callouts to Fig 3, we have now corrected those typos. Thank you!

5. “*Comparison of gene expression and protein data - as mentioned above, the authors are to be commended for generating comparable datasets from the same cells/sub compartments, however there are numerous reasons that gene expression and protein levels may not correlate exactly. While I appreciate the authors argument that this should be considered, it is not simply that this is a lack of reliability of either method, but that there are real biological reasons why this would occur. For example, it could be a difference in protein v transcript halflife - a similar difference is seen in GFAP protein (long half life) and Gfap gene (only produced when required for robust morphological changes). Can the authors comment on the developmental changes associated with Dlgap3, or halflife of SAPAP3?*”

This is an important comment and we are pleased that Reviewer 1 has asked us to clarify it. We do not think RNA-sequencing is unreliable; our lab has and continues to use this powerful approach. Our data show that the lack of correlation between RNA and protein should be considered and that such analyses could provide a clue to biological mechanisms that may drive that disconnect. To clarify on page 3 we write

“Although many of the top astrocyte and neuron enriched genes identified by RNA-seq^{2,3} were detected in the proteomes (Extended data Fig 9a,b), the relationship between protein abundance and

RNA expression was weak (Extended data Fig 9c-j), indicating that RNA levels do not accurately reflect protein abundance¹². This is not a critique of RNAseq, but likely reflects meaningful biology related to differences in transcript and protein turnover, as is known for other cells¹².”

Please note that SAPAP3 is not an example of a molecule that was detected by proteomics and not by RNAseq: it was detected reliably by both methods as being similarly and highly expressed in striatal astrocytes and neurons (see above text). In terms of the half-life of SAPAP3, it has recently been measured as ~11 days in 5 month old mice¹³. In relation to the question about development, we could not find any data on the developmental profile of *Dlgap3* and so we performed a specific set of experiments to evaluate this. These new data are reported in Extended data Fig 30 and are mentioned on page 7 with the following sentence

“Interestingly, postnatal *Dlgap3* expression in astrocytes and neurons differed in mice (Extended data Fig 30), portending future exploration of how SAPAP3 expression may relate to the emergence of OCD-phenotypes during development and adolescence.”

6. *“Distinct and convergent mechanisms for SAPAP3 in astrocytes and MSNs’ section - this is the weakest section of the manuscript overall, simply because the reported protein-protein interactions for SAPAP3 are putative at best. If the authors would like to retain such strong statements they really should complete pull-down assays to clearly show interaction with SAPAP3 and the putative binding partners detected in silico and shown in this figure. If these experiments are not possible, the authors should clearly state that these are predicted putative interactions, and perhaps move these data to the extended data or remove them altogether.”*

Thank you for this wise guidance. We have now toned down the language throughout the manuscript and use the word “putative” as guided. And, as suggested, we have also strengthened the study and validated interactions that are germane to this manuscript. On page 5 we now write

“The major interactions of SAPAP3 within astrocytes related to glutamate uptake and the actin cytoskeleton (Fig 4a). We therefore sought to validate key protein-protein interactions between SAPAP3 and Glt1 (*Slc1a2*) and of SAPAP3 with Ezrin (*Ezr*). Since SAPAP3, Ezrin, and Glt1 are expressed in other cells as well as astrocytes¹⁰ co-immunoprecipitation (co-ip) of endogenous proteins would not inform if they associate in astrocytes. Thus, we first used recombinant proteins expressed in striatal astrocytes *in vivo* for co-ip. We found HA-tagged SAPAP3 to co-ip with Ezrin-GFP and with Glt1-GFP, and conversely Ezrin-GFP and Glt1-GFP to co-ip with HA-SAPAP3 (Extended data Fig 21d). Second, to explore associations between endogenous proteins, we used a proximity ligation assay (PLA; Fig 4c) and found clear associations between SAPAP3 + Glt1 and between SAPAP3 + Ezrin in tdTomato positive astrocytes (Fig 4d,e). The PLA signals were absent in SAPAP3 KO mice (Fig 4c-d).”

7. *“Extended Data Fig 26 - these data are great! As these are some of the more compelling data in this manuscript it was a shame to see them tucked away in the extended data. Personally I find the fluoxetine data more compelling than the protein-protein interaction in silico prediction data (Fig 5) and more easily interpretable than the heatmaps in panel k/l of Fig 6.”*

Thank you! The dataset in question is now Extended data Fig 27. We too like the fluoxetine data, but in truth those controls are well established in the field and we performed those experiments largely as controls for our own studies. Given the limits on space, we have opted to keep those data in the Extended data. However, we do not dismiss the Reviewer’s point and have now re-designed Fig 5k to more clearly show the fluoxetine data on their own.

We hope this presentation meets the Reviewer's guidance to show these control data more clearly in the main manuscript (by modifying Fig 5k) and by keeping all the data in Extended data Fig 27 so anyone who wants to see them can do so.

Minor points.

1. *“the authors have, for valid reasons, focused on the striatum in their investigations - however given they have completed some in situ and IF validation of a small number of proteins/genes, how broad are the assumptions that these findings are true for other subtypes of astrocytes, neurons, and astrocyte-neuron interactions? Where similar localizations seen in other brain regions?”*

The constructs we report do work in other brain regions and we have started performing similar proteomic studies in other brain areas and hope to report those in a follow up manuscript in a year or so. The current manuscript already includes 30 Extended figures and adding more would make it dense and unfocused. Furthermore, if buried in the Extended data, those data may never get noticed or used. Finally, Reviewer 4 asked us to reduce the “resource” aspects of our manuscript. We would be discordant with that strong guidance if we added further brain regions that are not related to the main point of this study that is focused on the striatum and SAPAP3. For all these reasons, we hope to report additional brain regions in a follow up study.

We also point out that Supplementary information Table 1 lists the Addgene IDs for the 20 new AAV constructs we have made as part of our study: we are making them all openly available so others can use them in any brain area they are interested in without delay.

2. *“Main text section, start of second paragraph - the statement 'Physiological signaling between astrocytes and neurons also occurs between astrocyte processes and axon terminals and dendrites' is unclear, do the authors mean that the same astrocyte-neuron (cell body?) signaling occurs astrocyte-astrocyte and astrocyte-neuron dendrites/axons?”*

Yes we agree; that sentence was confusing. We have deleted that sentence and replaced it with a simpler one that makes sense on page 1.

3. *“Brain region, cell and subcompartment specific proteomes' section (reference to Extended Data Fig 1a,b) - this figure does not refer to interactions between neurons and astrocytes, or astrocytes and astrocytes”*

Yes, that figure call out was a typo. We have fixed it with a revised sentence on page 2.

4. *“Fig 3 - in it's current format this figure is not helpful. The Astrocyte card is a bit small to be effectively read - it would serve a more useful place simply in the extended data along with the other 'cards'. GO term and venn diagram analyses of proteins identified are non-validating ways of interrogating the data. Instead, validating some of these hits (e.g. like those in Fig 2) with some IF/IHC/in situ and visualizing the cells in a WT mouse brain would be more helpful in validating the narrative. References to Fig 3c are not helpful due to it being too small, and the information contained in 6 separate sub-panels. Please refer to the Extended Data Fig 15 with exact panels under discussion”*

As suggested, we have removed the previous version of Fig 3 and added all the astrocyte subproteomes cards as Extended data Figures 14-19 (as supplement to Fig 2). We have validated a protein in each astrocyte subcompartment and its co-localization score with the GFP-fusion of that subcompartment in each astrocyte card to assess not only expression within astrocytes, but also whether that protein was more likely present in its detected

subcompartment. Furthermore, we have reduced the length of the section related to the astrocyte cards, because Reviewer 4 asked us to reduce the resource aspects of the manuscript.

5. *“throughout - SAPAP3^{-/-} mice should be Dlgap3^{-/-} mouse (the gene is knocked out)”*

Although we agree with Reviewer 1, the SAPAP3 knockout mouse is widely used, well-characterized, and is an established mouse model that was named in 2007¹⁴. We have thus used accepted nomenclature to refer to the mouse line in our study as SAPAP3 KO with the appropriate references.

6. *“throughout - please check for small errors of gene/protein IDs and formatting - some are attributed incorrectly and can lead to confusion (given the impressive data from this group here and previously highlighting the importance of measuring both gene expression and protein levels)”*

Thank you. We have checked and reformatted the gene names.

6. *“confusion over SAPAP3 protein and Dlgap3 gene localization (referring to Fig 4) - Dlgap3 is clearly expressed in both the cell body and perhaps finer processes (hard to ascertain from Fig 4, panel H as the low mag (50uM scale bar larger image on the left) shows protein in all regions of the astrocyte, but the higher mag (20uM scale bar smaller image) shows only small dots of protein in the cell soma. There seems to be a disconnect between these two images, which may be reader error, but some clarification is required as this is an important part for the interpretation of these data. (particularly given the Dlgap3 mRNA is present only in the finer processes (panels in Fig 4g))”*

We think there was a misunderstanding here caused by our previous presentation. The 50 um scale bar image shows SAPAP3 protein expression in the entire dorsal striatum. This expression is consistent with prior work and shows that SAPAP3 is highly expressed in striatum and expressed in multiple cell types. As we zoom in to an astrocyte, the expression of SAPAP3 within the astrocyte is clustered (higher magnification image) to distinct compartments. We made this clearer in the figure legend to avoid confusion.

7. *“Figure 1, panel A - the coloration of somatic/membrane-bound protein is not as clear for neurons. Perhaps a small label at the top of these panels stating 'cell body' v 'cell membrane' would be an additional clarifying cue for readers”*

Thank you for the comment, we have added labels in Fig 1a to show the cytosol and the cell membrane so that there is a clarifying cue for readers.

Reviewer #4:

A. Summary of the key results

1. *‘Soto et al describe new findings on distinct proteome signatures of striatal neurons and astrocytes, including the identification of novel astrocytic expression and function of the protein Sapap3, which to date has been characterized as a post-synaptic scaffolding protein in glutamatergic synapses. The authors highlight that while there has been a rapid expansion in cell-specific gene expression analysis in brain tissue, less attention has been paid to cell-specific proteomic analysis which may better inform functional specialization of different cell types in the brain in health and disease. To address this gap, the authors leverage existing genetically targeted biotin ligase (BioID2) methods to develop a novel strategy to compare the proteome between cell types (neurons vs astrocytes) and cellular sub-*

compartments (membrane vs cytosol and 5 further astrocytic compartments), by using different AAV serotypes, gene promoters and compartment tags to achieve selective targeting for proteomic analysis. This provides a rich dataset that describes specific proteins, biological processes and protein networks that are differentially expressed between neurons and astrocytes and within different subcellular compartments. Surprisingly, this revealed astrocytic expression of the synaptic scaffolding protein Sapap3, which has been associated with obsessive compulsive disorder in animal and clinical studies.’

Thank you for these comments that highlight the relevance and importance of our study. We have addressed all your other points with revisions, analyses, and new data.

B. Originality and significance: if not novel, please include reference

2. “While the finding that a synaptic protein is expressed in astrocytes, and that its selective rescue in astrocytes in KO mice improves disease relevant phenotypes is novel and surprising, the impact would be significantly enhanced if the authors discovered what Sapap3 is doing in astrocytes that may contribute to OCD pathophysiology. Further, the significance and value of the proteomics results are somewhat lost in the extensive details about the method validation and long lists of differentially expressed proteins, enriched biological process and interaction map analyse, which feel like something you might expect in a “Resource” paper. Does this data provide new insight about neuron or astrocyte biology, or the specific changes in striatal neurons and astrocytes that contribute to OCD that could be targeted by new treatments? In its current state, this paper contains valuable descriptive results on proteomics, and causal manipulations demonstrating astrocyte and neuronal roles of Sapap3, however the lack of new data to understand differences in neuron and astrocyte biology or their contribution to disease limits the significance of the findings.”

We thank Reviewer 4 for these comments and guidance. The revised Fig 4 reports new data showing that SAPAP3 affects the actin cytoskeleton in astrocytes. Furthermore, the disrupted astrocyte cytoskeleton in astrocytes was recovered by astrocytic rescue of SAPAP3 in SAPAP3 KO mice, as was a key interaction between SAPAP3 and the actin cytoskeletal protein, Ezrin (Fig 6c,d; Extended data Fig 27). We believe these new data address the comment of Reviewer 4, namely “the impact would be significantly enhanced if the authors discovered what Sapap3 is doing in astrocytes that may contribute to OCD pathophysiology’. The new data show SAPAP3 in astrocytes affects the actin cytoskeleton in the context of OCD pathophysiology.

In terms of the second point about the manuscript being “resource” like, Reviewer 1 and the editor valued this aspect and we also believe that for the field these data are critical and new. Nonetheless, we took the comments from Reviewer 4 seriously and have reduced the length of methodological and resource-like sections significantly.

The revised manuscript is focused on the new insights related to astrocytic and neuronal contributions to OCD-phenotypes in mice. This has been achieved by including new data sets showing

- (i) How the astrocyte plasma membrane proteome is altered in SAPAP3 KO mice (Fig 4f)
- (ii) How SAPAP3 affects astrocyte morphology and the actin cytoskeleton (Fig 4f-j, Fig 6b,c)
- (iii) How SAPAP3-Ezrin and SAPAP3-Glt1 protein-protein interactions are altered in the context of SAPAP3 KO mice relative to controls (Fig c-e)
- (iv) How astrocyte morphology, the actin cytoskeleton, and SAPAP3-Ezrin and SAPAP3-Glt1 protein-protein interactions are restored following SAPAP3 rescue (Fig 4c-e; Fig 6b,c; Extended data Fig 27)
- (iv) How cell-specific rescue of SAPAP3 restores Δ FosB levels in neurons of SAPAP3 KO mice (Fig 6a; Extended data Fig 26).

As requested, the resource aspects have now been reduced substantially, e.g. by shifting to Extended data and by trimming the text.

C. Data & methodology: validity of approach, quality of data, quality of presentation
Main points

See E point 2 on grooming bout measure.

3. *“Extensive space and detail is dedicated to the description of the cell type and compartment specific proteome results (about 2.5/6.5 pages in main text and 3/6 figures), similar to what might be expected in a “Resource” style paper. In many places the novelty and value of the findings is not clear, besides that they provide a resource for data mining and developing new hypotheses to test (as was done for Sapap3). Given that the journal typically allows for only 3-4 modest figures and 2000-2500 words for the main text, this is probably going to need to be cut substantially for the final version, if the manuscript is published in this journal. Suggestions to achieve this include:*

a. The description of figure 1 in the text focussing on an explanation of how the methods were validated and how neurons and astrocytes were found to show expected differences in protein expression using the new methodological approach could be moved to a supplement. Instead, it would be valuable for the authors to more thoroughly explain how their data and interpretation of figure 1 provide new insight on neuron and astrocyte function.”

We have significantly reduced the length of the methodological aspects in relation to Fig 1. However, some explanation is critical for the rest of the manuscript to make sense – otherwise the discovery of astrocytic expression of SAPAP3 would be lost and lead to a disjointed and confusing manuscript. The methodology enabled the discovery and has to be presented before the OCD-related studies, albeit in shorter format as in the revised manuscript. Furthermore, for the glial biology field, the proteomic data are critical and thus, we have retained a smaller section on them in the text.

We have also addressed Reviewer 4’s comment “*to more thoroughly explain how their data and interpretation of figure 1 provide new insight on neuron and astrocyte function*”, by including in Fig 1h-K detailed analyses for the major differences between astrocytes and neurons at the protein level *in vivo* (a first for the field we believe). These data reveal fundamental differences between astrocytes and neurons with regards to Ca²⁺-dependent vesicular release and lipid homeostasis, and their molecular basis. They illustrate the power of the larger proteomic datasets we report.

In terms of the comment about the number of figures and length, we have followed the detailed revision instructions provided to us by the editor to revise the manuscript and believe the revised manuscript is consistent with those editorial guidelines.

b. “Could the first paragraph describing results from extended figures 3-6 validating the specificity of the approach be moved to a discussion in the supplement?”

The entire paragraph cannot be moved to the Extended data as this would break the logic and flow of the manuscript and would make the proteomic findings confusing. However, we have reduced this section substantially to one small paragraph in order to reduce the methodological aspects as suggested.

c. “Could figure 3a and b be integrated into figure 2, so that all “astrocyte cards” are included as supplemental figures?”

Thank you for this suggestion. We followed this guidance and have removed the previous version of Fig 3 and added all the astrocyte cards as Extended data Figs 14-19. We have also integrated panels from previous Fig 3a and b into a revised Figure 2 as suggested.

Minor points:

4. *“Grooming in Sapap3 KO mice relative to WT mice (extended data figure 25) is much higher and more uniform than previous reports in similar aged mice (e.g. <https://doi.org/10.1038/s42003-020-01611-y>, DOI: 10.1111/gbb.12557, <https://doi.org/10.1038/s41386-018-0307-2>, <https://doi.org/10.1016/j.jneumeth.2017.05.026>, <https://doi.org/10.1038/s41598-021-88769-5> all show heterogenous grooming phenotype in similar aged mice). Can you provide any explanation as to why your colony would have a more robust/intense/penetrant phenotype (or a published example with similar levels of behavioural severity)? Do you select heterozygous breeders based on susceptibility to grooming phenotype? Do you house animals individually or in group cages during studies? It is important to recognize these differences, and look for possible explanations to enhance reproducibility of these effects.”*

We have gone back and looked at our data carefully and do not believe anything is remiss. As stated in the methods, the behavioral evaluations were performed during the light cycle between 10 am and 2 pm in order to capture the more extensive grooming episodes (as described in Welch et al 2007 in Ref ¹⁴). Additionally, when compared to other studies, the phenotype we report is not more severe: our SAPAP3 KO mice spend about 30-50% of the time grooming, which is similar to the data from Welch et al ¹⁴ and others). In terms of the other clarifications, we do not select heterozygous breeders based on susceptibility to grooming phenotype, we always house animals in groups, and avoid housing animals individually (as required by local animal welfare rules). We agree with the referee about reproducibility, and as such we have stated all methods clearly in the methods section.

5. *“Can you please include details of whether or not Sapap3 KO littermates were used in active and control groups for studies (vs non littermate controls). i.e. were litters with multiple KO mice split between experimental groups?”*

Yes, litters with multiple SAPAP3 KO mice were split between experimental groups. For example, if a litter had four SAPAP3 KO mice, two mice were injected with eGFP-SAPAP3 and two with GFP control. The mice were kept with their littermates in the cage according to capacity and local UCLA mouse housing rules. We have added this note to the methods section.

D. Appropriate use of statistics and treatment of uncertainties

Main points:

6. *“For figure 6, Mann-Whitney Test is used on grooming bout data for one experiment (astro-Sapap3), whereas Unpaired t test with Welch was used for the other (neuro-Sapap3) plotted in the same figure (grooming bouts). By eye, the astro-sapap3 data appear more normally distributed and the neuro-Sapap3 look more skewed (which would require the non parametric Mann-Whitney Test). Were these tests used correctly? Given that significant weight was placed on the idea that astro-Sapap3 has an effect on grooming bouts but neuro-Sapap3 does not, it doesn't seem appropriate to use different types of statistical tests in the two datasets (to address whether these have different effects they should really be tested together in a two way ANOVA).”*

Fig 6 is now Fig 5 in the revised manuscript. We have checked all the statistics in this figure. Because most of the data were not normally distributed, we could not perform a two-way

ANOVA on these data. Therefore, we have applied Kruskal-Wallis tests with Dunn post-hoc test when significant to compare the means of all four groups together (Astro GFP, Astro SAPAP3, Neuro GFP, Neuro SAPAP3). This is made clear in the data analysis section and in the figure legends. Also, the raw replicate values and the details of the statistical tests are provided in the Excel spreadsheets 7 and 8. A related comment is also addressed below.

7. “*equivalent sapap3 expression in neurons and astrocytes*” (Fig 4a-b). What statistics were performed to analyse this data (were any statistical analyses performed)? These data look like they would show statistical difference in expression (especially 4b and 4a for “MSN”-cyto vs Astrocyte-cyto).”

Fig 4 is now Fig 3 in the revised manuscript. In order to obtain the values in Figure 3a-b, linear models using empirical Bayes methods using R package limma were used to identify the proteins and genes with a false discovery rate (FDR) < 0.05 when compared to GFP controls. These are the appropriate statistical methods for large multiomic datasets and this is accurately reported in the methods and the FDR values are stated in the revised Fig 3a,b. However, it is not appropriate to pick any protein or gene and compare its protein or RNA expression between two data sets using a simple t test for such multiomic data. This is why we stated that the levels were “equivalent” and did not perform any statistical tests to support the word “equivalent” in a paired way like the reviewer suggests. We are confident the data have been analyzed correctly, but we have removed the word “equivalent” and replaced it with “similar” throughout the manuscript, which is supported by the data.

Minor points:

8. “*The methods section says that “exact p-values are shown in the figure panels or in the figure legends”. Although these are available in the supplement they are not available in the figure panel/legend for figure 6, please add them.*”

We edited the methods and removed the sentence that stated “exact p-values are shown in the figure panels or in the figure legends.” We have added the following sentences to the methods section, and this is reflected in all the figures and legends:

“If the data were normally distributed we employed parametric tests, while if they were not normally distributed we used non parametric tests. Paired and unpaired Student’s two-tailed t tests (as appropriate) two tailed Mann-Whitney tests, and one and two-way ANOVAs were used for most statistical analyses with significance declared at $P < 0.05$. When P values were greater than 0.05, they are stated as non-significant (n.s.). When the P value was less than 0.01, it is stated as < 0.01 . All proteomic and transcriptomic analyses used a statistical FDR value < 0.05 unless otherwise stated. All mice were assigned to particular experimental groups at random. No data points were excluded from any experiment.”

E. Conclusions: robustness, validity, reliability

Main points:

9. “*I strongly recommend that the title “reveal mechanisms of obsessive-compulsive disorder” should be adjusted. This is misleading, given that the majority of studies were conducted in an OCD-relevant mouse model, and studies on human RNA seq data do not “reveal mechanisms of OCD”. It is important to make clear distinctions, and not over state what can be inferred from animal research and descriptive post mortem analysis, particularly in an influential journal (and in the title of article). “Astrocyte and neuron subcompartment proteomes reveal mechanisms of Sapap3 function relevant to obsessive-compulsive disorder” may be more appropriate for the title. Other areas where the authors should be*

careful with language around clinical relevance include “how astrocyte and neuronal molecular mechanisms contribute to a psychiatric disease” (abstract) and “We focused on this discovery because of the potential clinical relevance to understanding how astrocytes and neurons may both contribute to OCD-like behaviors in mice.” (end page 4).”

We regret our excitement for the work has caused the Reviewer alarm. This was not the intended outcome. Nonetheless, we changed the title to read “Astrocyte and neuron subproteomes reveal mechanisms relevant to obsessive-compulsive disorder” to address the reviewer. We have also deleted all the flagged sentences and stated we have worked with “OCD-like phenotypes in mice” in the abstract and in the text. We hope this is now acceptable.

10. *“The statements “astrocytic SAPAP3 and neuronal SAPAP3 preferentially rescued repetitive and anxiety like behaviors, respectively”, “the effect of astrocytic GFP-SAPAP3 was significantly greater for rescue of the number of self-grooming bouts when compared to neuronal GFP-SAPAP3” and “our experiments show that astrocyte and neuron mechanisms converge to drive dysfunction in OCD, and that targeting both is necessary to be therapeutically effective” are not supported by the experimental data, and all mention of separate effects of these interventions should be removed. While it is true that the authors provide evidence that only neuronal, and not astrocytic, SAPAP3 can rescue anxiety-like behaviour (on a single measure of anxiety, which is not ideal and should be mentioned as a limitation), there is strong evidence that both neuronal and astrocytic SAPAP3 rescue grooming on all measures except # of grooming bouts. Importantly, the definition of grooming bouts was coarser than what is typically used in the literature, which questions the sensitivity of this bout measure. Specifically, “Separate grooming bouts were considered when the pause was more than 5 sec or if behaviors other than self-grooming occurred” was used to define grooming bouts, which is much longer than what is typically used to define new grooming bouts in mice (e.g. papers using 2 second pause: <https://doi.org/10.1016/j.jneumeth.2017.05.026>, <https://doi.org/10.1038/s42003-020-01611-y>, doi:10.1038/nn.4395 and 3 seconds pause used by Welch et al 2007 Nature doi:10.1038/nature06104). In addition to this, less animals were included in the neuro-SAPAP3 rescue experiment compared to the astrocyte-SAPAP3 experiment, and it appears that this analysis may be underpowered (the absolute difference between means is similar between astro-sapap3 and neuro-sapap3 studies, but SEM error bars are much larger due to the smaller sample size for neuro-sapap3 group). To say that astro-Sapap3 had a great effect on grooming than neuronal-Sapap3, it would be more accurate to use a two way ANOVA comparing these groups. Given these limitations of grooming bout analysis, and effects of both neuro- and astro-Sapap3 on grooming time and lesions, these effects on grooming should not be described as different.”*

We have gone back and checked the analyses we performed. We conducted our grooming bout analysis with the 5 s pause following a Nature Protocols Resource ¹⁵ and in order to keep consistent with our lab’s prior grooming analysis ⁹. This is stated clearly in the methods. Because some of the data were non-parametric and the standard deviations differed markedly even when the data were normally distributed using the Bartlett’s test method, we could not use a Two-way ANOVA. Therefore, in order to address the Reviewer’s comment we conducted Kruskal-Wallis tests for the both the grooming and anxiety behaviors. There was no significant difference between astrocyte and neuron rescue for the grooming behaviors when the means were compared. Therefore, we have removed the statement “astrocytic SAPAP3 and neuronal SAPAP3 preferentially rescued repetitive and anxiety like behaviors, respectively”. Thank you for this guidance!

However, our data still show that only neuronal rescue can restore anxiety-like behavior. Furthermore, we have added elevated plus maze (EPM) data in Figs 5e, 5d, and 5i to

support our open field data. We have also added EPM data to our WT vs SAPAP3 KO and fluoxetine experiments. When we conducted Kruskal-Wallis test with a subsequent Dunn post-hoc test on the anxiety behavior of the cell-type specific rescue data, this showed that there was a significant difference between cell type when comparing Astro-SAPAP3 and Neuro-SAPAP3 means. All these data are added to the statistical supplement and to the figure legends. Therefore, on two independent measures of anxiety, only neuronal intervention rescues the high anxiety phenotype of SAPAP3 mice.

We have made many revisions to make these distinctions clear and to address the Reviewer's comment.

Minor points:

11. *“astrocytic rescue of GFP-SAPAP3 resulted in beneficial effects comparable to fluoxetine”. This is misleading given that Fluoxetine improves anxiety but astrocytic Sapap3 does not. Please remove or adjust this statement.”*

We have adjusted this statement on page 6 with the following words

“By this metric, astrocytic rescue by GFP-SAPAP3 resulted in beneficial effects comparable to fluoxetine in the self-grooming measurements (Extended data Fig 25).”

F. Suggested improvements: experiments, data for possible revision

Main points:

12. *“The discovery that Sapap3 is expressed in astrocytes is very significant, however the impact of this finding is substantially reduced without some understanding of the biological function of Sapap3 in astrocytes. Does Sapap3 act as a scaffold in astrocytes like it does in neurons? Does the presence vs absence of Sapap3 have some functional significance (e.g. effects on glutamate uptake/homeostasis and the actin cytoskeleton)? Are there any simple functional assays that can be performed in WT vs Sapap3 KO vs Astro-Sapap3 rescue, to begin to demonstrate whether Sapap3 does anything to the function of its binding partners described in figure 5?”*

We thank the Reviewer for appreciating the high significance of our discovery that SAPAP3 is expressed in astrocytes. We have addressed the Reviewer's comment to provide additional data concerning how SAPAP3 works in astrocytes. In the revised version, Fig 5 is now Fig 4 and reports new data showing that SAPAP3 affects the actin cytoskeleton in astrocytes. Furthermore, the disrupted astrocyte cytoskeleton in astrocytes was recovered by astrocytic rescue of SAPAP3 in SAPAP3 KO mice (Fig 6b,c; Extended data Fig 27). Moreover, the interaction of SAPAP3 with Ezrin (an actin cytoskeletal stabilizing protein) was disrupted in the SAPAP3 KO mice and restored by SAPAP3 rescue. We believe these new data address the overall comment of Reviewer 4 to provide an understanding of the biological function of SAPAP3 in astrocytes. These new data are reported between pages 5-7.

G. References: appropriate credit to previous work?

Minor points:

13. *“A circuit based understanding of OCD is slowly emerging 30,60,61, but striatal molecular and cellular mechanisms giving rise to OCD-like phenotypes remain to be understood”. It would be appropriate to reference papers that have begun to examine striatal cellular mechanisms of OCD-relevant phenotypes, including the contribution of PV+ interneurons (DOI: 10.1126/science.1232380), D2 MSNs (DOI: 10.1038/s41386-021-01161-9) and astrocytes (<https://doi.org/10.1038/npp.2015.26> and your own*

studies <https://doi.org/10.1016/j.neuron.2018.08.015>). A discussion of how the current findings relate to this existing work would also be valuable for readers.

We have revised that section of the discussion and include the following sentences to cite the suggested papers and to place our work in context. Given the space constraints, a longer discussion was not possible, but could be done with a follow up review or something similar.

“Classically considered a neuronal disease, OCD involves striatal circuit malfunction¹⁶⁻¹⁸, but the molecular and cellular basis of the disorder has remained unclear¹⁹. However, it is emerging that diverse cell types can contribute to OCD-phenotypes^{9,20-24}. Building on recent work with depression²⁵ and degeneration^{8,26}, our experiments show that astrocyte and neuron SAPAP3 mechanisms are of relevance to OCD phenotypes in mice. Our proteomic experiments demonstrate how SAPAP3, a protein shared by astrocytes and neurons and involved in human OCD²⁷⁻³⁰, produces effects on complex OCD-related behavioural and neuronal phenotypes via distinct astrocyte and neuron molecular interactions that affect the astrocytic actin cytoskeleton.”

H. Clarity and context: lucidity of abstract/summary, appropriateness of abstract, introduction and conclusions

“Writing was very clear, although the context related to OCD was at times overstated (see)”

Thank you! As suggested in preceding sections, we have followed the Reviewer’s guidance to tone down some of the assertions we previously made. The revised manuscript is more balanced as a result.

Main point:

14. “MSN” should be replaced with “neuron” in all instances describing studies that rely on viruses using *hSyn* promoter, given that *hSyn* promoter is “pan-neuronal” and is expected to transfect striatal interneurons. Although interneurons constitute only a small proportion of striatal neurons, they are functionally significant in the context of this work, with parvalbumin interneurons providing strong inhibitory control over MSNs, with evidence that this is capable of normalizing compulsive grooming in *Sapap3* KO mice (doi: 10.1126/science.1232380. and more recently <https://doi.org/10.1101/2022.01.10.475745>).”

We have now used the general term “neuron” as suggested when referring to our studies that use the *hSyn* promoter. We also agree with the reviewer that 95% of the neurons are MSNs based on our analyses (See Reviewer Fig 2). On page 2 we write

The striatum contains extensive contacts between astrocytes and neurons, 95% of which are DARPP32 positive medium spiny neurons (MSNs)^{2,5,31}.”

Reviewer Fig 2. Assessment of MSNs in the striatum. Representative 40x striatal images of NeuN (yellow) pan-neuronal marker with DARPP32 (blue) medium spiny neuron (MSN) marker. Bar graph shows that 95% of NeuN+ neurons are DARPP32+ MSNs.

I. Other minor points

15. “Relative to the AAV fluorescent protein controls, we detected ~800-1600 proteins enriched within each of the Astro BioID2, Astro Lck-BioID2, Neuro BioID2 or Neuro Lck-BioID2 groups following in vivo expression.” Where does this value come from? Is it associated with any panel in figure 1? For example Fig 1e shows 1640 +262 enriched proteins in Neuro-Lck-BioID2 group, whereas other groups have ~500-600 based on the ven diagram values.”

We agree that section was not clear. We have now revised the volcano plots in Figs 1 and 2 to more clearly indicate the numbers of proteins that were unique to each group and those that were shared, but enriched in each group. The numbers are provided on the volcano plots. The reviewer is correct, the Neuro-Lck-BioID2 group contains 1640 unique proteins and 262 shared ones. The Astro-Lck-BioID2 group contains 246 unique proteins and 262 shared ones. These distinctions are now clear in all the revised volcano plots in Figs 1 and 2, and in the Extended data.

16. “In the text, figure 4a is mislabelled figure 3A (lines 1-2 of page 5), and figures 5c-d are mislabelled (d and e respectively, page 5 late paragraph 2).”

We have fixed these typos.

17. “For open field diagrams, can the square within the arena be re-sized so that it positioned at the centre/periphery boundary that is used to quantify anxiety? This would be more informative, and without checking the legend carefully its easy to assume that that is what the inner line represents this boundary. The methods say that the chamber was 28cm wide, and the figure 6 diagrams have an inner boundary ~1/6 of the total width (~4.67cm; the extended figure 25 looks even larger and is ~1/4 the chamber width or 7cm wide). The methods say the actual boundary was 2.5cm from the chamber walls, so this line in the figure panel is misleading. Please check which is correct and update the figure panels to reflect the actual boundary, and include this in the figure legend.”

The correct value was that reported in the methods section: the actual boundary was 2.5 cm from the boundary wall. The line in the previous Figure panel was misleading and we have removed the inner line for clarity in the figure panels and to avoid confusion. The boundary values are clearly stated in the figure legends.

18. “Ext fig 24: please show higher resolution images so that it is clear whether cells co-express markers (similar to Ext Fig 21). Neuro-Sapap3 expression is very difficult to see. For all colocalization studies to examine cell type selectivity of viral constructs (Ext Fig 5, 21, 24), did any GFP or HA+ cells not express S100 β or DARPP-32/NeuN? Please include this data in the supplement (or make it clearer if already there).”

As suggested, we have added higher magnification images to Fig 5b and to Extended data Fig 22. Neuro-SAPAP3 expression may have been difficult to see in the previous lower magnification images, because of its highly synaptic localization. The cellular expression is now clearer in the higher magnification images in the revised Fig 5b. For all co-localization studies we have included the mean, SD, and SEM of GFP or HA+ cells that do not express S100 β or DARPP-32/NeuN+ in the appropriate figure panels (e.g. Extended Figure 22). There was negligible co-localization in the potentially off target cell, and high co-localization in the target cell. The AAV constructs were cell selective and this has been reported previously several times²⁻⁹.

Overall, thanks to all the Reviewers for their comments, which have improved the study.

References cited here

- 1 Bhat, R. V. *et al.* Expression of the APC tumor suppressor protein in oligodendroglia. *Glia* **17**, 169-174, doi:10.1002/(sici)1098-1136(199606)17:2<169::Aid-glia8>3.0.Co;2-y (1996).
- 2 Chai, H. *et al.* Neural circuit-specialized astrocytes: transcriptomic, proteomic, morphological, and functional evidence. *Neuron* **95**, 531-549 (2017).
- 3 Diaz-Castro, B., Gangwani, M. R., Yu, X., Coppola, G. & Khakh, B. S. Astrocyte molecular signatures in Huntington's disease. *Sci Transl Med* **11**, eaaw8546 (2019).
- 4 Nagai, J. *et al.* Hyperactivity with Disrupted Attention by Activation of an Astrocyte Synaptogenic Cue. *Cell* **177**, 1280-1292 e1220 (2019).
- 5 Oceau, J. C. *et al.* An Optical Neuron-Astrocyte Proximity Assay at Synaptic Distance Scales. *Neuron* **98**, 49-66 (2018).
- 6 Shigetomi, E. *et al.* Imaging calcium microdomains within entire astrocyte territories and endfeet with GCaMPs expressed using adeno-associated viruses. *The Journal of general physiology* **141**, 633-647, doi:10.1085/jgp.201210949 (2013).
- 7 Tong, X. *et al.* Astrocyte Kir4.1 ion channel deficits contribute to neuronal dysfunction in Huntington's disease model mice. *Nat Neurosci* **17**, 694-703, doi:10.1038/nn.3691 (2014).
- 8 Yu, X. *et al.* Context-Specific Striatal Astrocyte Molecular Responses Are Phenotypically Exploitable. *Neuron* **108**, 1146-1162.e1110, doi:10.1016/j.neuron.2020.09.021 (2020).
- 9 Yu, X. *et al.* Reducing Astrocyte Calcium Signaling In Vivo Alters Striatal Microcircuits and Causes Repetitive Behavior. *Neuron* **99**, 1170-1187 e1179 (2018).
- 10 Saunders, A. *et al.* Molecular Diversity and Specializations among the Cells of the Adult Mouse Brain. *Cell* **174**, 1015-1030.e1016, doi:10.1016/j.cell.2018.07.028 (2018).
- 11 Howarth, A. G., Hughes, M. R. & Stevenson, B. R. Detection of the tight junction-associated protein ZO-1 in astrocytes and other nonepithelial cell types. *Am J Physiol* **262**, C461-469, doi:10.1152/ajpcell.1992.262.2.C461 (1992).
- 12 Liu, Y., Beyer, A. & Aebersold, R. On the Dependency of Cellular Protein Levels on mRNA Abundance. *Cell* **165**, 535-550, doi:10.1016/j.cell.2016.03.014 (2016).
- 13 Kluever, V. *et al.* Protein lifetimes in aged brains reveal a proteostatic adaptation linking physiological aging to neurodegeneration. *Sci Adv* **8**, eabn4437, doi:10.1126/sciadv.abn4437 (2022).
- 14 Welch, J. M. *et al.* Cortico-striatal synaptic defects and OCD-like behaviours in Sapap3-mutant mice. *Nature* **448**, 894-900, doi:10.1038/nature06104 (2007).
- 15 Kalueff, A. V., Aldridge, J. W., LaPorte, J. L., Murphy, D. L. & Tuohimaa, P. Analyzing grooming microstructure in neurobehavioral experiments. *Nature protocols* **2**, 2538-2544, doi:10.1038/nprot.2007.367 (2007).
- 16 Burguière, E., Monteiro, P., Mallet, L., Feng, G. & Graybiel, A. M. Striatal circuits, habits, and implications for obsessive-compulsive disorder. *Curr Opin Neurobiol* **30**, 59-65, doi:10.1016/j.conb.2014.08.008 (2015).
- 17 Graybiel, A. M. & Grafton, S. T. The striatum: where skills and habits meet. *Cold Spring Harb Perspect Biol* **7**, a021691, doi:10.1101/cshperspect.a021691 (2015).
- 18 Kalueff, A. V. *et al.* Neurobiology of rodent self-grooming and its value for translational neuroscience. *Nat Rev Neurosci* **17**, 45-59, doi:10.1038/nrn.2015.8 (2016).
- 19 Stein, D. J. *et al.* Obsessive-compulsive disorder. *Nature reviews. Disease primers* **5**, 52, doi:10.1038/s41572-019-0102-3 (2019).
- 20 Burguière, E., Monteiro, P., Feng, G. & Graybiel, A. M. Optogenetic stimulation of lateral orbitofronto-striatal pathway suppresses compulsive behaviors. *Science* **340**, 1243-1246, doi:10.1126/science.1232380 (2013).

- 21 Ahmari, S. E. *et al.* Repeated cortico-striatal stimulation generates persistent OCD-like behavior. *Science* **340**, 1234-1239, doi:10.1126/science.1234733 (2013).
- 22 Ramírez-Armenta, K. I. *et al.* Optogenetic inhibition of indirect pathway neurons in the dorsomedial striatum reduces excessive grooming in Sapap3-knockout mice. *Neuropsychopharmacology* **47**, 477-487, doi:10.1038/s41386-021-01161-9 (2022).
- 23 Aida, T. *et al.* Astroglial glutamate transporter deficiency increases synaptic excitability and leads to pathological repetitive behaviors in mice. *Neuropsychopharmacology* **40**, 1569-1579, doi:10.1038/npp.2015.26 (2015).
- 24 Chen, S. K. *et al.* Hematopoietic origin of pathological grooming in Hoxb8 mutant mice. *Cell* **141**, 775-785, doi:10.1016/j.cell.2010.03.055 (2010).
- 25 Cui, Y. *et al.* Astroglial Kir4.1 in the lateral habenula drives neuronal bursts in depression. *Nature* **554**, 323-327, doi:10.1038/nature25752 (2018).
- 26 Guttenplan, K. A. *et al.* Neurotoxic reactive astrocytes induce cell death via saturated lipids. *Nature*, doi:10.1038/s41586-021-03960-y (2021).
- 27 Züchner, S. *et al.* Multiple rare SAPAP3 missense variants in trichotillomania and OCD. *Mol Psychiatry* **14**, 6-9, doi:10.1038/mp.2008.83 (2009).
- 28 Bienvenu, O. J. *et al.* Sapap3 and pathological grooming in humans: Results from the OCD collaborative genetics study. *American journal of medical genetics. Part B, Neuropsychiatric genetics : the official publication of the International Society of Psychiatric Genetics* **150b**, 710-720, doi:10.1002/ajmg.b.30897 (2009).
- 29 Boardman, L. *et al.* Investigating SAPAP3 variants in the etiology of obsessive-compulsive disorder and trichotillomania in the South African white population. *Comprehensive psychiatry* **52**, 181-187, doi:10.1016/j.comppsy.2010.05.007 (2011).
- 30 Crane, J. *et al.* Family-based genetic association study of DLGAP3 in Tourette Syndrome. *American journal of medical genetics. Part B, Neuropsychiatric genetics : the official publication of the International Society of Psychiatric Genetics* **156b**, 108-114, doi:10.1002/ajmg.b.31134 (2011).
- 31 Khakh, B. S. Astrocyte-Neuron Interactions in the Striatum: Insights on Identity, Form, and Function. *Trends Neurosci* **42**, 617-630, doi:10.1016/j.tins.2019.06.003 (2019).

Reviewer Reports on the First Revision:

Referees' comments:

Referee #2 (Remarks to the Author):

The authors have addressed the concerns raised in the original round of review, and the manuscript presents with a clearer narrative and more appropriately concludes from the data as presented.

Additional points to be edited/for clarification:

1. Inclusion of clarifying statement at line 1086 that STRING output was only of validated connections is appreciated. It would be nice to also see this statement included in the text lest readers are unaware that the putative/predicted connections have been removed. Unfortunately the data does seem to include some predicted interactions still. For example, Fig 2, panel h shows interactions between Slc1a6 (bottom left) with Slc1a2 and Slc32a1 – on the STRING online database, these are predicted interactions from 'gene neighbourhood' 'text mining' and 'co expression' studies. Other spot checks of 'known' connections in the many STRING diagrams provided in the manuscript produce similar concerns. Can the authors comment on this irregularity.
2. Full color blots are provided in Extended Data Fig 1, however monochrome blots are not provided in full with only cropped images in the Extended Data. E.g. Ext Data Figs 3, 4, 12, among others.
>> please amend Extended Date Fig 1 to state that the blot is cropped for Extended Data Fig 21d

Referee #4 (Remarks to the Author):

Thank you to the authors for your thorough and well organized response to the reviewers. I agree with the authors that the revisions have substantially enhanced the impact of the manuscript and interest to the broad readership of Nature.

Specifically, new data enhances the understanding of how astrocytic sapap3 may be involved in astrocyte function and provide clues about astrocytic involvement in OCD pathophysiology.

I also appreciate the revised statistics which are much clearer. I have two comments relating to this and a few other minor points:

1) For statistical analysis included in an extended data table, for 2 way ANOVAs it would be important to report main effects and interactions. For example, data from figure 5J is analysed with a 2 way ANOVA, but only post hoc tests are reported. There is a significant rescue x cell type interaction, suggesting that neuro-Sapap3 rescues distance travelled more than Astro-Sapap3 (this conclusion is supported by a Bonferroni posthoc test). There may be other examples where interactions from a two-way ANOVA provide additional insight to the data.

2) Replicates within an individual animal (multiple measures with an animal) are often included as individual data points on graphs. I assume that this is typical of how this is done in the field for most measures (e.g. astrocyte area graphs in the extended data figure 27) but for the new delta FosB analysis in figure 6 I think this is inappropriate. Typically in the field if multiple levels/sections of the same brain region are quantified these values are averaged and one value is plotted and analysed per animal. The approach to take multiple measures from some but not all animals is artificially inflating the power of the study (see <https://doi.org/10.1016/j.neuron.2014.10.042> for more discussion on this issue). Please reconsider whether this approach is appropriate for delta Fos B data (and other areas where it is used), and reanalyse data with individual data point per animal as needed.

3) I appreciate the authors revisions around the relevance to OCD. One suggestion related to extended figure 29 data is to include brief methods commenting on how the 61 OCD and TS genes were selected? E.g. Was this based on a threshold for # total publications in Phenopedia? How many came from OCD vs Tourette Syndrome lists?

4) Figure 5b and c shows spread of GFP labelled sapap3 above the corpus callosum in rescue experiments. Would it be possible to quantify the spread of the rescue construct to clearly demonstrate the degree of extra-striatal sapap3 expression that was produced, as well as the degree of spread to ventral striatum which is now highlighted as relevant in figure 6a and b?

Reviewer Reports on the First Revision:

Dear Dr. Rowland,

Thank you for the careful review of our revised manuscript and for the opportunity to revise it to address the remaining comments from the reviewers. You will see in the point-by-point responses below that we have fully addressed the Reviewer comments with clarifications, edits, inclusion of further statistical analyses, and with the provision of raw blots as requested. The additions to the text are shown in blue font.

In addition, we have carefully checked all the figures for errors, made some of the lines thicker and increased spacing between the graphs to make the layout sharper. We also removed redundant words, and moved the previous Extended data Fig 1 into Supplementary file 1 along with the other raw western blots.

Please let us know if there is something we have missed. With these changes, we hope the manuscript will be considered suitable for publication.

{REDACTED}

Reviewer #1:

“The authors have addressed the concerns raised in the original round of review, and the manuscript presents with a clearer narrative and more appropriately concludes from the data as presented.”

Thank you for these supportive comments for the revised manuscript, and thank you for your previous comments that made it stronger.

“Additional points to be edited/for clarification:

1. Inclusion of clarifying statement at line 1086 that STRING output was only of validated connections is appreciated. It would be nice to also see this statement included in the text lest readers are unaware that the putative/predicted connections have been removed. Unfortunately the data does seem to include some predicted interactions still. For example, Fig 2, panel h shows interactions between Slc1a6 (bottom left) with Slc1a2 and Slc32a1 – on the STRING online database, these are predicted interactions from ‘gene neighbourhood’ ‘text mining’ and ‘co expression’ studies. Other spot checks of ‘known’ connections in the many STRING diagrams provided in the manuscript produce similar concerns. Can the authors comment on this irregularity.”

Thank you for pointing this out. We want to emphasize that the STRING database is valuable to predict interactions and that further experimental evidence is required to solidify *bona fide* protein-protein interactions, as we have shown in several cases in the last round of revisions for the key interactions we report. Although we filtered the larger STRING analysis interactions for the AP-MS experiments, we cannot discount that the probability scores for

“gene neighborhood”, “text mining” and “co-expression studies” were higher than AP-MS and thus, were captured in the maps. Picking and choosing what we showed in the maps also seemed inappropriate to us, and could be arbitrary and potentially misleading. Therefore, in order to avoid confusion and present the data as transparently as possible, we have now referred to all STRING network figures as protein-protein association maps throughout the manuscript text and in the figures. Additionally, all network edges are now referred to as “putative” associations or “putative” interactions in both the text, the figures, and the figure legends. This includes Figure 1, Figure 2, Extended data figure 7, and all the astrocyte cards (Extended data figures 13-18). Furthermore, we have also clarified the methods section to include the following on page 23

“A list of protein-protein associations and putative interactions from published datasets was assembled using STRING. STRING database interactions were filtered to include affinity purification mass spectrometry validations (AP-MS). We caution that such interactions are putative and have been labelled as such, and further validations are necessary on a case-by-case basis, as we have done for the key interactions reported herein.”

“2. Full color blots are provided in Extended Data Fig 1, however monochrome blots are not provided in full with only cropped images in the Extended Data. E.g. Ext Data Figs 3, 4, 12, among others. >> please amend Extended Date Fig 1 to state that the blot is cropped for Extended Data Fig 21d”

The full uncropped blots for every Western blot experiment conducted in the manuscript, including the full blot corresponding to figure 20d, are now all provided as Supplemental file 1. Not including them was an oversight on our part, which is now corrected. Thank you!

Reviewer #2:

“Thank you to the authors for your thorough and well organized response to the reviewers. I agree with the authors that the revisions have substantially enhanced the impact of the manuscript and interest to the broad readership of Nature.

Specifically, new data enhances the understanding of how astrocytic Sapap3 may be involved in astrocyte function and provide clues about astrocytic involvement in OCD pathophysiology. I also appreciate the revised statistics which are much clearer. I have two comments relating to this and a few other minor points:”

Thank you to the reviewer for their supportive comments. The additional comments are addressed below.

“1) For statistical analysis included in an extended data table, for 2 way ANOVAs it would be important to report main effects and interactions. For example, data from figure 5J is analysed with a 2 way ANOVA, but only post hoc tests are reported. There is a significant rescue x cell type interaction, suggesting that neuro-Sapap3 rescues distance travelled more than Astro-Sapap3 (this conclusion is supported by a Bonferroni posthoc test). There may be other examples where interactions from a two-way ANOVA provide additional insight to the data.”

Thank you for this comment, which applied to Fig 5j and Extended data Fig 23. This was an oversight from our part and we have now double checked the statistics as suggested. For Figure 5j and Extended data Fig 23, where we used two-way ANOVA with Bonferroni post-hoc test, we have added the appropriate “overall ANOVA” statistics for rescue x cell-type interaction. These can be found in the revised Extended data table 7 and 8. There, we report the degrees of freedom, F-statistic, and P-value for the interaction before application of the post-hoc test. As

the reviewer noted, there is a significant interaction between rescue x cell-type that suggests Neuro SAPAP3 rescues both locomotion behaviors more than Astro SAPAP3 (total distance traveled and total speed). This is in accord with our findings as Neuro SAPAP3 treated mice display significantly less anxiety phenotypes, and thus travel more at faster rates.

We have also now checked all the statistics files; the data are reported correctly in the revised files.

“2) Replicates within an individual animal (multiple measures with an animal) are often included as individual data points on graphs. I assume that this is typical of how this is done in the field for most measures (e.g. astrocyte area graphs in the extended data figure 27) but for the new delta FosB analysis in figure 6 I think this is inappropriate. Typically in the field if multiple levels/sections of the same brain region are quantified these values are averaged and one value is plotted and analysed per animal. The approach to take multiple measures from some but not all animals is artificially inflating the power of the study (see <https://doi.org/10.1016/j.neuron.2014.10.042> for more discussion on this issue). Please reconsider whether this approach is appropriate for delta Fos B data (and other areas where it is used), and reanalyse data with individual data point per animal as needed.”

Many thanks to the reviewer for this guidance. Yes, we have performed the experiments and analyses according to past studies, which are all cited in the manuscript. We had not performed the Δ FosB experiments/analysis before and appreciate the guidance provided by the Reviewer. We have now reanalyzed the Δ FosB data and show the results as the average of all fields of view for each mouse. We also conducted the statistical tests based on mouse numbers rather than field of view, as suggested. The new analyses are still significant and do not change our results or conclusions and we have reported all the values in Extended data files 7 and 8 for the graphs shown in Figure 6 and Extended data figure 25. Furthermore, we have changed the graphs in Figure 6 and Extended data figure 25 to reflect the mouse numbers rather than the number of sections.

“3) I appreciate the authors revisions around the relevance to OCD. One suggestion related to extended figure 29 data is to include brief methods commenting on how the 61 OCD and TS genes were selected? E.g. Was this based on a threshold for # total publications in Phenopedia? How many came from OCD vs Tourette Syndrome lists?”

Thank you for this suggestion. The requested methods are now provided on page 24 with the following sentences:

“Comparison of human and mouse datasets in OCD

The 61 genes associated with human OCD and Tourette’s syndrome were obtained from Phenopedia (phgkb.cdc.gov; accessed Jan. 2021). The genes were chosen on a threshold of at least two publications. A total of 63 OCD and 23 Tourette’s syndrome genes were obtained. When compared, 15 genes overlapped between the OCD and Tourette’s syndrome lists. The 61 genes plotted represent genes that have homologs in mice and were detected at any quantity (FPKM > 0) in our mouse RNA-seq studies.”

“4) Figure 5b and c shows spread of GFP labelled sapap3 above the corpus callosum in rescue experiments. Would it be possible to quantify the spread of the rescue construct to clearly demonstrate the degree of extra-striatal sapap3 expression that was produced, as well as the degree of spread to ventral striatum which is now highlighted as relevant in figure 6a and b?”

We think there is a slight misunderstanding here in regards to the “spread” to the ventral striatum. The spread to the ventral striatum in these experiments was planned for and results

from the AAV microinjection procedure that was designed to cover most of the striatum. These methods are accurately described on page 18 and were almost identical to those used previously by others. Thus, the spread to the ventral striatum is extensive, expected and desired. Indeed, we suspect we have covered all the striatum. We also revised the methods on page 18 and now write

“Briefly, GFP or GFP-SAPAP3 AAVs was bilaterally injected into the striatum of 3-4 week old mice through 2 sites at 3 locations per hemisphere. At each of the injection sites, the microinjection needle was advanced to the deepest (ventral) position for the first injection while the additional injections were made every 0.3 mm while withdrawing the injection needle. The coordinates from bregma were: injection site 1: anterior 0.5 mm, mediolateral 1.5 mm, dorsoventral 2.9 mm, 2.6 mm, 2.3 mm from pial surface; injection site 2: anterior 0.5 mm, mediolateral -1.5mm, dorsoventral 2.9 mm, 2.6 mm, 2.3 mm from pial surface. For each injection location, 150 nl of virus was injected and the needle was left in place for 5 mins after each injection. These injection procedures were chosen to cover most of the striatum (dorsal and ventral).”

The spread to the corpus callosum is accurately reported here and in all our past studies with local AAV microinjections into the striatum ¹⁻⁵. This expression results from the fact that the needle tract has to penetrate the cortex and the corpus callosum in order to reach the striatum. Although we wait for the virus to be expelled from the microinjection needle into the striatum before we slowly withdraw the needle, this procedure is imperfect and a little AAV gets sucked up the needle tract and results in some expression in the corpus callosum. Over a period of nearly 10 years we have found no way to avoid this and believe this occurs to some degree in all studies that have used AAV microinjections in the striatum (although it is possible past researchers underreported it). In terms of quantifying the spread, the expression is restricted to the corpus callosum and sometimes the overlying cortex for about 100 um above the striatum. We finish by emphasizing that this spread to the corpus callosum is accurately reported in this study and in our past papers ¹⁻⁵. Furthermore, this small off-striatum expression cannot explain the results we report, which are based on extensive and multiple lines of evidence as reported in the manuscript. Finally, SAPAP3 is not found in the corpus callosum or the immediate cortex overlying the striatum, but is highly enriched in the striatum that was specifically targeted by our microinjections. To make this clear for the reader, on page 18 of the methods we write

“As shown in the figures, we confirmed that intrastriatal microinjection of AAV2/5-delivered cargo was cell selective and restricted to the striatum, although there was a little expression proximal to the needle tract in cells of the cortex and sometimes of the corpus callosum. We suspect such expression occurred in all past studies employing viruses, as it is impossible to reach subcortical brain structures without advancing and withdrawing the needle through the overlying tissue: all studies employing microinjections (including ours) need to be interpreted with this anatomical caveat in mind. We have previously reported this in regards to our surgical procedures and discussed it ¹⁸.”

(Note: Ref 18 in the manuscript is Ref 2 below)

References cited here

- 1 Chai, H. *et al.* Neural circuit-specialized astrocytes: transcriptomic, proteomic, morphological, and functional evidence. *Neuron* **95**, 531-549 (2017).
- 2 Nagai, J. *et al.* Hyperactivity with Disrupted Attention by Activation of an Astrocyte Synaptogenic Cue. *Cell* **177**, 1280-1292 e1220 (2019).
- 3 Oceau, J. C. *et al.* An Optical Neuron-Astrocyte Proximity Assay at Synaptic Distance Scales. *Neuron* **98**, 49-66 (2018).

- 4 Yu, X. *et al.* Context-Specific Striatal Astrocyte Molecular Responses Are Phenotypically Exploitable. *Neuron* **108**, 1146-1162.e1110, doi:10.1016/j.neuron.2020.09.021 (2020).
- 5 Yu, X. *et al.* Reducing Astrocyte Calcium Signaling In Vivo Alters Striatal Microcircuits and Causes Repetitive Behavior. *Neuron* **99**, 1170-1187 e1179 (2018).

Reviewer Reports on the Second Revision:

Referees' comments:

Referee #2 (Remarks to the Author):

The compartment-specific targeting tools on display here remain the highlight of the manuscript. This is an exciting approach, that has been used to good effect here to produce a nice resource set of proteome data.

The authors have addressed the concerns raised in the original rounds of review. I note two points below that should only be text edits, though the first raises some concern if the comparison is not as region-specific, or if the methods applied are not as fair a comparison as the authors suggest in their comparison.

Major.

1. missing data/methods - the authors spend some time discussing the importance of in vivo methodology, and compare in vitro astrocytes showing deficits in culture systems, however no information is provided about purification and culture, or purity measurements. This seems disingenuous, as many culture methods have been shown to provide poor replicates of in vivo astrocytes, while others have been shown to more adequately recapitulate some aspects (while still not recapitulating others). A lack of methods is a major oversight. Are these isolated striatal astrocytes (presumably a requirement for comparison with in vivo lucifer yellow filled striatal astrocytes?), at what age are these astrocytes isolated (from single brains? - hard to tell from the figure legend).

The authors have great data, the tools are impressive, and it is well-reported that the complex bushy nature of astrocytes is not recapitulated in vitro. But their focus here is not on morphology, why not compare proteomes? Can these tools be used in this system? This would be incredibly powerful for the development/screening of new therapies - for instance for OCD. The authors really missed an opportunity here to make some fundamental discoveries that could help patients in a big way.

Minor.

1. the authors should edit the title to state 'in mice' as they have no evidence that this pathway, or astrocytes, are involved in human patients with obsessive compulsive disorder. (it should be noted that the authors make this admission in the abstract)

Referee #4 (Remarks to the Author):

I'm satisfied with the authors response, and feel the manuscript is ready for publication.

Author Rebuttals to Second Revision:

Dear Dr. Rowland,

Thank you for the review of our revised manuscript (2021-11-17741B) and for the opportunity to revise it to address the remaining two comments from Reviewer #2. We have highlighted the changes with blue font in the manuscript file.

Please let us know if there is something we have missed.

{REDACTED}

Referee #2:

“The compartment-specific targeting tools on display here remain the highlight of the manuscript. This is an exciting approach, that has been used to good effect here to produce a nice resource set of proteome data. The authors have addressed the concerns raised in the original rounds of review. I note two points below that should only be text edits, though the first raises some concern if the comparison is not as region-specific, or if the methods applied are not as fair a comparison as the authors suggest in their comparison.”

Thank you for these supportive comments and for recognizing the value of the approach and data. We address your comment below and have made the requested text edits in blue font.

1. *“missing data/methods - the authors spend some time discussing the importance of in vivo methodology, and compare in vitro astrocytes showing deficits in culture systems, however no information is provided about purification and culture, or purity measurements. This seems disingenuous, as many culture methods have been shown to provide poor replicates of in vivo astrocytes, while others have been shown to more adequately recapitulate some aspects (while still not recapitulating others). A lack of methods is a major oversight. Are these isolated striatal astrocytes (presumably a requirement for comparison with in vivo lucifer yellow filled striatal astrocytes?), at what age are these astrocyte isolated (from single brains? - hard to tell from the figure legend).”*

Thank you for these clarifying questions, which we did not make clear in the previous version. We now clarify that we did not culture astrocytes for any experiment in this manuscript. We assessed astrocytes *in vivo* under identical conditions for all our experiments.

In the comment above, we believe the Reviewer is referring to Extended data figure 1c, which shows GFP+ striatal astrocytes that were dissociated for FACS and compared to astrocytes with full morphology from lightly fixed whole striata of 8 week old mice. These cells were thus from the same brain

region (striatum) and the same age (8 weeks) as the Lucifer yellow dye filled astrocytes shown next to them in that figure (from 15 and 8 mice, respectively). These details were stated in the legend for Extended data figure 1, but we were remiss in not including detailed methods for this aspect. This was an innocent oversight, which we have now fixed on pages 15-16 by adding the requested methods. In addition, we added to the Extended data figure 1 legend that the cells were from 8 week old mice, again as requested.

As part of the same comment, the Reviewer wrote, “*The authors have great data, the tools are impressive, and it is well-reported that the complex bushy nature of astrocytes is not recapitulated in vitro. But their focus here is not on morphology, why not compare proteomes?*”

We agree with the reviewer. To clarify and explain, in Extended data figure 1c-d we compared the full morphology of astrocytes filled with Lucifer yellow to those that were dissociated to show that the latter method is invalid since dissociated astrocytes lose all their branches and processes (as the Reviewer points out). Since the cells were destroyed by dissociation, we considered it inappropriate to use them for proteomics and this is why we developed approaches to determine their proteomes *in vivo*. In a past study¹, we did use dissociated astrocytes. In that study, a total of 1,378 proteins were detected. With our new methods reported in the current manuscript, 3,274 proteins across all compartments were identified. These comparisons further confirm that analyzing dissociated astrocytes is misleading, as the reviewer states, and support our focus in the current manuscript on reporting astrocyte proteomes determined *in vivo* with new approaches. We clarify this on page 4 with the following sentence

With our *in vivo* methods, 3,274 astrocyte subcompartment proteins were identified, whereas for dissociated astrocytes¹ only 1,378 were detected, underscoring that dissociated cells lose most of their processes and associated proteomes (Extended data Fig 1).

Later, as part of the same comment the Reviewer wrote, “*Can these tools be used in this system? This would be incredibly powerful for the development/screening of new therapies - for instance for OCD. The authors really missed an opportunity here to make some fundamental discoveries that could help patients in a big way.*”

We believe the Reviewer is referring to the deployment of the proteomic methods we report in OCD model mice. Yes; that is exactly what we did and Figure 4 of the manuscript is dedicated to this. Those studies were performed in response to the comments from the first round of revisions requested in February 2022. We are happy that the Reviewer feels our approach is incredibly powerful.

Thus, we used our tools to determine the astrocyte proteomes of control and OCD model mice. These data comprise Figure 4 in which we identified and compared the interactors of SAPAP3 in astrocytes and neurons (Fig 4a and b). Additionally, in Fig 4f we determined the plasma membrane subproteome of astrocytes in the OCD mouse model, compared it to the wild type, and identified novel mechanisms that contribute to OCD. The validation and testing of these actin cytoskeleton mechanisms is reported Figures 4 and 6.

Furthermore, we also compared astrocyte and neuron proteomic data to published data from humans (Extended data figure 28). These data showed that there exist astrocyte and neuron-specific as well as shared astrocyte/neuron proteins detected in the OCD mouse model and also associated with human OCD, which speaks to the Reviewer's comment about discoveries related to human OCD. Our approaches and data will inspire further studies, by us and others, aimed at fundamental mechanisms in OCD.

Minor.

1. *“the authors should edit the title to state 'in mice' as they have no evidence that this pathway, or astrocytes, are involved in human patients with obsessive compulsive disorder. (it should be noted that the authors make this admission in the abstract)”*

We chose the title based on specific guidance provided to us during the first review. We would be happy to add the words “in mice” to the title if the Editor thinks it is appropriate to do so. We ask because the current title was requested by another reviewer.

Referee #4:

“I'm satisfied with the authors response, and feel the manuscript is ready for publication.”

Thank you to Reviewer #4 for their thorough review of our manuscript. The comments provided have made our manuscript stronger.

References cited here

1. Chai, H., B. Diaz-Castro, E. Shigetomi, *et al.* 2017. Neural circuit-specialized astrocytes: transcriptomic, proteomic, morphological, and functional evidence. *Neuron*. **95**: 531-549.